# Conformal Risk Training:
# End-to-End Optimization of Conformal Risk Control

**Christopher Yeh, Nicolas Christianson, Adam Wierman, Yisong Yue**
Department of Computing and Mathematical Sciences
California Institute of Technology
Pasadena, CA 91125
{cyeh,nchristi,adamw,yyue}@caltech.edu

## Abstract

While deep learning models often achieve high predictive accuracy, their predictions typically do not come with any provable guarantees on risk or reliability, which are critical for deployment in high-stakes applications. The framework of conformal risk control (CRC) provides a distribution-free, finite-sample method for controlling the expected value of any bounded monotone loss function and can be conveniently applied post-hoc to any pre-trained deep learning model. However, many real-world applications are sensitive to tail risks, as opposed to just expected loss. In this work, we develop a method for controlling the general class of Optimized Certainty-Equivalent (OCE) risks, a broad class of risk measures which includes as special cases the expected loss (generalizing the original CRC method) and common tail risks like the conditional value-at-risk (CVaR). Furthermore, standard post-hoc CRC can degrade average-case performance due to its lack of feedback to the model. To address this, we introduce "conformal risk training," an end-to-end approach that differentiates through conformal OCE risk control *during model training or fine-tuning*. Our method achieves provable risk guarantees while demonstrating significantly improved average-case performance over post-hoc approaches on applications to controlling classifiers' false negative rate and controlling financial risk in battery storage operation.

## 1  Introduction

We study the problem of training deep learning models that are used for potentially risky downstream decision making. For example, in high-stakes tasks such as tumor classification, doctors need models that achieve both good overall classification accuracy and *provably* bounded false negative rate, to ensure that the health risk of a false negative prediction (i.e., misclassifying a tumor as benign) is sufficiently prioritized in model predictions. In such settings, it is important to design a unified approach (trained model and decision-making policy) that simultaneously controls for the desired level of risk while maximizing the utility of downstream decisions.

One promising paradigm is *risk control*. Given a (pre-trained) model whose predictions are used by a decision policy parameterized by $\lambda \in \Lambda$, the goal is to choose $\lambda$ such that $\mathbb{E}[L(\lambda)] \leq \alpha$ for some loss function $L$ and risk level $\alpha$. Many common goals can be framed as risk control problems: bounding the false negative rate of a classifier, producing predictive uncertainty sets that satisfy a target coverage level, ensuring factuality of large language model outputs, etc. [6, 29].

The conformal risk control (CRC) method [6] introduces a sufficient criterion for solving the risk control problem when the loss function $L$ is monotone. While CRC is elegant and simple, the original formulation has several limitations. First, CRC is limited to controlling the *expected* loss of $L$, while more general notions of risk are critical for high-stakes problems in the real world. Notably, the

39th Conference on Neural Information Processing Systems (NeurIPS 2025).

original paper on CRC [6] poses an open question on how to tackle more general notions of risk beyond expected loss, such as the conditional value-at-risk (CVaR). Second, because CRC is purely applied post-hoc (i.e., given a pre-trained model) to model outputs without providing any feedback to the model, it may significantly degrade model performance.

In this paper, we propose a theoretical and algorithmic framework, called *conformal risk training*, that extends CRC to enable end-to-end training and is applicable to tail risk formulations such as CVaR. Our key contributions are as follows:

1. First, we develop a risk control method for controlling the general class of *optimized certainty-equivalent (OCE)* risks [10, 11], a broad class of risk measures that includes as special cases the expected loss (generalizing the original CRC method) and the conditional value-at-risk (CVaR) [50]. In particular, the CVaR formulation partially answers an open question posed by the original CRC paper [6]. The key insight is that any OCE risk can be bounded by a monotone transformation of the loss, thereby preserving the monotonicity required for CRC-style methods.

2. Second, we propose *conformal risk training*, a method that trains a model and the conformal risk control procedure end-to-end. Our method substantially generalizes the method of conformal training of uncertainty sets [47, 51] to the conformal risk control setting. This trains the model to be "risk aware"—i.e., it learns to produce predictions that maximize performance while minimizing downstream risk.

3. Finally, we demonstrate empirically that fine-tuning models using conformal risk training leads to significant performance improvements while guaranteeing satisfaction of risk constraints. We present results for maximizing model specificity while controlling false negative rate on a tumor image segmentation task, as well as maximizing average profit while controlling tail-risk of losses in a battery storage operation task.

**Related work.** Previous works that explore post-hoc conformal risk control either bound the expectation of a monotone loss function, or provide a high-probability bound for (possibly) more general losses. We use the term "post-hoc" to refer to procedures that are applied to the outputs of a pretrained prediction model without further fine-tuning of the model. Within the class of bounds on expected losses, conformal prediction (CP) [49, 45, 4] bounds expected miscoverage loss for set-valued predictions, whereas conformal risk control (CRC) [6] generalizes CP to bound the expectation of any bounded monotone loss function. For high-probability risk bounds, Risk-Controlling Prediction Sets (RCPS) [9] bound the expected loss of set-valued predictors for monotone losses, whereas Learn Then Test (LTT) [5] bounds more general risks and non-monotone losses at the cost of typically looser bounds from having to apply a family-wise error correction procedure. Recently, [16] developed a high-probability bound for the family of distortion risk measures which includes CVaR. While [6] shows how to convert a high-probability risk bound to an expected risk bound, it is not clear that their methodology can be extended to directly bound more general risks like CVaR. As such, to the best of our knowledge, our work is the first conformal-style approach that provides a *certain* (as opposed to high-probability) bound on risk measures beyond expected loss for monotone loss functions.

Several prior works within the CP literature have also explored calibrating model uncertainty during training. Conformal training [47] and related works [21, 17, 39] introduced methods for incorporating CP differentiably during model training by treating part of each minibatch as a pseudo-calibration set. These works primarily focus on reducing the size of calibrated prediction sets. In contrast, our *conformal risk training* method is the first to incorporate conformal risk control differentiably during model training and is compatible with more general performance objectives such as reducing the false positive rate of a classifier or minimizing an expected decision loss. We show in Appendix E that conformal training is a special case of our method.

Beyond the conformal prediction literature, a number of works in the machine learning literature (e.g., [33, 20, 35]) have introduced risk-sensitive objectives into learning; while these methods may, for example, reduce CVaR risk, they do not come with risk control guarantees.

Finally, our work is related to methods from "predict-then-optimize" [22] and decision-focused learning [37, 44], especially the growing literature on decision-focused uncertainty quantification (UQ). These methods aim to generate prediction sets that optimize some downstream decision-making objective while still maintaining calibration. Several of these methods are applied post-hoc to (potentially) decision-agnostic models [48, 18, 31], whereas others have combined conformal training with decision-focused learning [25, 51]. Our conformal risk training method builds upon

this decision-focused UQ literature: whereas these prior works have largely focused on set-valued predictions and their associated risks, our method allows for more general risk measures.

**Outline.** This paper is structured as follows. Section 2 introduces our overall problem setting and reviews the standard CRC result. Section 3 introduces our broad generalization to the conformal control of OCE risks, of which standard CRC is a special case, and shows that this framework can be extended to more general assumptions in the specific case of CVaR risks. Section 4 introduces our conformal risk training procedure and discusses how the gradient of the risk-controlling parameter can be computed. Section 5 highlights key experimental results, and Section 6 concludes. Additional experimental results are presented in Appendix A, and all proofs are deferred to Appendix C.

**Notation.** $[N]$ denotes the set $\{1, 2, \ldots, N\}$. $[x]_+$ denotes the function $\max\{0, x\}$. $\mathbf{1}[\cdot]$ is the indicator function, while $\mathbf{1}_d$ is a length-$d$ vector of ones. $\frac{\partial}{\partial x}$ denotes a partial derivative; $\frac{d}{dx}$ denotes the total derivative. $\partial f(x)$ is the subdifferential of a function $f$ at a point $x$.

## 2  Preliminaries and Background on Conformal Risk Control

A primary goal in the training and deployment of machine learning (ML) models is to achieve both good overall performance and to ensure *reliable* deployment through the control of some notion of risk. To achieve this dual goal, existing approaches follow a two-stage, "Pretrain, then Risk Control" approach. First, an ML model with parameters $\theta \in \Theta \subseteq \mathbb{R}^D$ is trained to minimize a standard training objective, such as cross-entropy for classification. Then, after training, the decision-maker applies a post-hoc *risk control* procedure to the model to guarantee provably bounded risk. Formally, let $L : \Theta \times \Lambda \to \mathbb{R}$ denote a (random) mapping from model parameters $\theta \in \Theta$ and an "aggressiveness" parameter $\lambda \in \Lambda$ to some loss. The decision-maker seeks to choose the parameter $\lambda$ to ensure that risk–the expectation of the loss $L$–is bounded at a chosen level $\alpha$:

$$\mathbb{E}[L(\theta, \lambda)] \leq \alpha. \tag{1}$$

For instance, in a tumor image segmentation problem, $L$ may denote the fraction of false negative pixels on a randomly drawn image, and $\alpha$ is a desired upper bound on the false negative rate (FNR). The aggressiveness parameter $\lambda \in [0, 1]$ can be chosen as the threshold used to distinguish positive predictions from negative ones. Thus, a smaller $\lambda$ will yield more positive predictions and a lower false negative rate, and a greater $\lambda$ will yield more negative predictions and a higher false negative rate. See Example 1 for a more detailed description of this task.

A number of post-hoc approaches to control the risk of pretrained machine learning models have been proposed in the literature. Most notable is the approach of *conformal risk control (CRC)* [6], which gives a distribution-free, finite-sample methodology to provably enforce the risk bound (1). In the remainder of this section, we will omit the model parameters $\theta$ as an input to the loss function $L$, since the results hold beyond the case of controlling the risk of machine learning model decisions.

CRC assumes that the decision-maker has a dataset $\{L_1, \ldots, L_N\}$ of previous loss functions, and that the goal is to control the marginal loss of the next instance: $\mathbb{E}[L_{n+1}(\lambda)] \leq \alpha$. This can be done under several mild assumptions.

**Assumption 1.** $\Lambda \subset \mathbb{R}$ *is a closed and bounded set with minimum value* $\lambda_{\min} := \min \Lambda$. *The set* $\{L_i : \Lambda \to \mathbb{R}\}_{i=1}^{N+1}$ *is a set of exchangeable, left-continuous random functions. Furthermore,* $B : \Lambda \to \mathbb{R}$ *is a left-continuous function that almost surely upper bounds each* $L_i$ *pointwise, so that for all* $i \in [N+1]$, $\Pr(\forall \lambda \in \Lambda : L_i(\lambda) \leq B(\lambda)) = 1$.

**Assumption 2.** *The functions* $\{L_i\}_{i=1}^{N+1}$ *are almost-surely nondecreasing, and* $B$ *is nondecreasing.*

**Assumption 3.** *The risk control problem is* feasible. *That is, the desired risk level* $\alpha \in \mathbb{R}$ *is chosen such that* $\mathbb{E}[L_{N+1}(\lambda_{\min})] \leq \alpha$.

Under these assumptions, CRC gives the following approach for selecting $\lambda$ to control risk.

**Proposition 1.** *Under Assumptions 1 and 2, let* $\alpha \in \mathbb{R}$ *be a desired risk level and define the set* $\hat{\Lambda}$ *as*

$$\hat{\Lambda} := \{\lambda \in \Lambda \mid h(\lambda) \leq \alpha\}, \quad where \quad h(\lambda) := \frac{1}{N+1}\left(B(\lambda) + \sum_{i=1}^{N} L_i(\lambda)\right). \tag{2}$$

*Then, for any* $\lambda \in \hat{\Lambda}$, *we have* $\mathbb{E}[L_{N+1}(\lambda)] \leq \alpha$. *Furthermore, if Assumption 3 holds, then choosing* $\hat{\lambda} := \max\{\lambda_{\min}, \sup \hat{\Lambda}\}$ *ensures risk control:* $\mathbb{E}[L_{N+1}(\hat{\lambda})] \leq \alpha$.

We make three brief remarks. First, the CRC procedure we describe above in Proposition 1 is actually a mild generalization of the original method in [6]; in particular, we allow the risk upper bound $B$ to be a function of $\lambda$, whereas prior work assumes $B$ to be a constant. In practice, this allows us to achieve tighter bounds on the risk. Second, we assume that the functions $L_i$ are left-continuous and nondecreasing and we choose $\hat{\lambda}$ via the supremum, following the convention of [34]; on the other hand, the original CRC paper [6] assumes right-continuous and nonincreasing functions $L_i$ and chooses a conservativeness parameter $\hat{\lambda}$ via the infimum. These approaches are equivalent up to reflection of the parameter $\lambda$. Finally, observe that due to monotonicity of the function $h(\lambda)$, the risk controlling parameter $\hat{\lambda}$ can be computed in practice through bisection search; see Algorithm 3 in Appendix B for a full description of this approach.

Standard CRC (Proposition 1) provides a sufficient criterion for choosing the parameter $\hat{\lambda}$ to ensure bounded *marginal* (*i.e.*, expected) loss. However, it does not address the control of more general notions of risk; in the next section, we will show that this framework can be broadly extended to the control of the general family of *optimized certainty equivalent* risks.

## 3   Conformal Risk Control for Optimized Certainty Equivalents

The previous section describes the control of *expected* loss to achieve the risk bound (1). However, in real-world, high-stakes applications, decision-makers may seek to control their risk beyond just the expected loss, especially if they are sensitive to losses of particular magnitudes. In general, they are faced with the problem of controlling their loss $L$ under some chosen risk measure $R$, which maps from random variables to $\mathbb{R}$:

$$R[L(\lambda)] \leq \alpha. \tag{3}$$

While the CRC methodology described in the previous section can be extended to achieve the more general risk control (3) in certain special cases, such as when $R$ is a quantile (see [6, Section 4.2]), there are many other important notions of risk which this approach cannot directly accommodate. For instance, in [6, Section 4.2], the authors pose the question of whether CRC or some other approach can be extended to enforce the risk control bound (3) when $R$ is the *conditional value-at-risk* (CVaR), a common risk measure in financial and energy systems applications [42, 32, 38, 36].

In this section, we answer this question in the affirmative, proving that in fact, CRC can be extended to control any risk in the broad family of *optimized certainty equivalent* risks, defined as follows.

**Definition 1** ([10, 11])**.** *A risk measure $R$ mapping a real-valued random variable $X$ to $\mathbb{R}$ is an* **optimized certainty equivalent (OCE)** *risk measure if $R[X]$ can be expressed as*

$$R[X] = \inf_{t \in \mathbb{R}} t + \mathbb{E}[\phi(X - t)],$$

*where $\phi : \mathbb{R} \to \mathbb{R} \cup \{+\infty\}$ is a disutility function that is nondecreasing, closed, and convex with $\phi(0) = 0$ and $1 \in \partial\phi(0)$.*

The family of OCE risks includes a number of practical and popular risk measures, including mean-variance and entropic risks [33]. Moreover, the *conditional value-at-risk* can also be shown to be an OCE risk measure with the disutility function $\phi_{\mathrm{CVaR}^\delta}(x) = \frac{1}{1-\delta}[x]_+$.

**Definition 2** ([41, 42])**.** *Let $X$ be a real-valued random variable, and let $F$ be its cumulative distribution function. The* **conditional value-at-risk** *of $X$ at level $\delta \in [0, 1)$, denoted $\mathrm{CVaR}^\delta[X]$, is the average value of $X$ on its $\delta$-tail, or the $1 - \delta$ fraction of its largest outcomes. If $X$ has a density, then $\mathrm{CVaR}^\delta[X] = \mathbb{E}[X \mid F(X) \geq \delta]$. For general random variables $X$, $\mathrm{CVaR}^\delta[X]$ can be expressed via the variational formula*

$$\mathrm{CVaR}^\delta[X] = \inf_{t \in \mathbb{R}} t + \frac{1}{1-\delta}\, \mathbb{E}[X - t]_+.$$

In Theorem 1, we show that the CRC methodology described in Proposition 1 can be broadly generalized to accommodate *any OCE risk measure*.

**Assumption 4.** *The OCE risk control problem is* feasible. *That is, the desired risk level $\alpha$ is chosen such that $R[L_{N+1}(\lambda_{\min})] \leq \alpha$.*

---

**Algorithm 1** (Post-hoc) Conformal OCE Risk Control (CORC)

---

**Require:** parameter space $\Lambda = [\lambda_{\min}, \lambda_{\max}]$, functions $\{L_i\}_{i=1}^N$, upper-bound $B$, risk threshold $\alpha \in \mathbb{R}$,
  numerical tolerance $\epsilon > 0$, hyperparameter $t \in \mathbb{R}$

  **function** CORC($\Lambda, \{L_i\}_{i=1}^N, B, \alpha, \epsilon, t$, disutility $\phi : \mathbb{R} \to \mathbb{R}$)
    $\underline{\lambda} \leftarrow \lambda_{\min}, \overline{\lambda} \leftarrow \lambda_{\max}$
    **while** $\overline{\lambda} - \underline{\lambda} > \epsilon$ **do**
      $\lambda \leftarrow (\underline{\lambda} + \overline{\lambda})/2$
      **if** $\tilde{h}(\lambda, t) \leq \alpha$ **then** $\underline{\lambda} \leftarrow \lambda$ **else** $\overline{\lambda} \leftarrow \lambda$               ▷ $\tilde{h}$ *is defined in* (4)
    **return** $\lambda$
  **function** CONFORMALCVARCONTROL($\Lambda, \{L_i\}_{i=1}^N, B, \alpha, \epsilon, t$, quantile level $\delta \in [0, 1)$)
    **if** $\alpha < B(\lambda_{\min})$ or $t \notin [B(\lambda_{\min}), \alpha]$ **then return** $\lambda_{\min}$
    **return** CORC($\Lambda, \{L_i\}_{i=1}^N, B, \alpha, \epsilon, t, \phi_{\mathrm{CVaR}^\delta}$)

---

**Theorem 1.** *Suppose Assumptions 1 and 2 hold, and fix $\alpha, t \in \mathbb{R}$. Let $R$ be an OCE risk measure with disutility function $\phi$. Define the exchangeable random functions $\{\tilde{L}_{i,t} : \Lambda \to \mathbb{R}\}_{i=1}^{N+1}$ by $\tilde{L}_{i,t}(\lambda) := t + \phi(L_i(\lambda) - t)$, and let $\tilde{B}_t(\lambda) := t + \phi(B(\lambda) - t)$. Define*

$$\tilde{h}_t(\lambda) := \frac{1}{N+1}\left(\tilde{B}_t(\lambda) + \sum_{i=1}^N \tilde{L}_{i,t}(\lambda)\right), \tag{4}$$

*and let $\hat{\Lambda}_t := \{\lambda \in \Lambda \mid \tilde{h}_t(\lambda) \leq \alpha\}$. For every $\lambda \in \hat{\Lambda}_t$, $R[L_{N+1}(\lambda)] \leq \alpha$. Furthermore, if Assumption 4 holds, then choosing $\hat{\lambda} := \max\{\lambda_{\min}, \sup \hat{\Lambda}_t\}$ ensures risk control: $R[L_{N+1}(\hat{\lambda})] \leq \alpha$.*

The key insight underlying this theorem is that for any OCE risk, $\tilde{h}_t$ is nondecreasing in $\lambda$. This structure gives rise to the conformal OCE risk control (CORC) algorithm shown in Algorithm 1, which computes $\hat{\lambda}$ using bisection search.

Theorem 1 is a strict generalization of the original CRC methodology in Proposition 1. By choosing the disutility $\phi(x) = x$, the risk measure $R$ is simply the expectation, and we recover the setting of controlling expected risk. In this special case, for all $t \in \mathbb{R}$, we have

$$\forall i \in [N] : \ \tilde{L}_{i,t} = L_i, \quad \tilde{B}_t = B, \quad \tilde{h}_t(\lambda) = h(\lambda), \quad \hat{\Lambda}_t = \hat{\Lambda},$$

thus recovering Proposition 1 exactly. Moreover, Theorem 1 answers positively the question from [6] of whether CRC can be generalized to control the CVaR, since the CVaR is an OCE risk measure. In fact, it is possible to obtain an even more general result in the specific case of controlling the CVaR. Specifically, we may relax the condition in Assumption 2 that the losses $L_i$ are nondecreasing.

**Assumption 5.** *All $\{L_i\}_{i=1}^{N+1}$ and $B$ are monotonic in $\lambda$. Note that within the set of functions $\{B\} \cup \{L_i\}_{i=1}^{N+1}$, we allow for some functions to be monotonically nondecreasing (e.g., $L_1, L_3$), while others may be monotonically nonincreasing (e.g., $L_2, B$).*

Even under this milder assumption on the structure of the losses $L_i$, we can obtain the following risk control guarantee for the CVaR.

**Theorem 2.** *Suppose that Assumptions 1 and 5 hold. Fix any $\delta \in [0, 1)$ and $\alpha \in \mathbb{R}$. Define $\hat{\Lambda}_t$ as in Theorem 1 using the disutility function $\phi_{\mathrm{CVaR}^\delta}(x) = \frac{1}{1-\delta}[x]_+$. Then, for every $t \in [B(\lambda_{\min}), \alpha]$ and $\lambda \in \hat{\Lambda}_t$, we have the risk control bound $\mathrm{CVaR}^\delta\left[L_{N+1}(\lambda)\right] \leq \alpha$.*

While the above result is written specifically for case of CVaR, we note that the result can be extended to some, but not all, other OCE risk measures; see Appendix C.3 for further details. Theorem 2 motivates the conformal CVaR control algorithm, which we present in Algorithm 1. Note that Theorem 2 is only non-vacuous when $B(\lambda_{\min}) \leq \alpha$; otherwise, assuming feasibility of the risk control problem (Assumption 4), we can still use $\lambda_{\min}$ to obtain the desired bound.

Compared to the standard CRC result (Proposition 1), Theorems 1 and 2 both involve an additional hyperparameter $t$. For the risk bounds to hold, $t$ should not depend on the calibration data $\{L_1, \ldots, L_N\}$ used to select the risk control parameter $\hat{\lambda}$. In practice, we recommend using an additional held-out set of losses $\{L'_1, \ldots, L'_k\}$ (e.g., the training set), and picking the $t$ that yields the largest risk control parameter on this set.

**Algorithm 2** Conformal Risk Training

---

**function** CONFORMALRISKTRAINING(training set $\{(x_i, y_i)\}_{i=1}^M$, parameter space $\Lambda = [\lambda_{\min}, \lambda_{\max}]$, upper-bound $B : \Lambda \to \mathbb{R}$, disutility $\phi : \mathbb{R} \to \mathbb{R}$, risk threshold $\alpha \in \mathbb{R}$, numerical tolerance $\epsilon > 0$, initial model parameters $\theta$)

    **for** mini-batch $D \subseteq [M]$ **do**

        Randomly split batch: $(D_{\text{cal}}, D_{\text{pred}}) \leftarrow D$

        Define loss functions $L_i(\theta, \lambda) := L(x_i, y_i, \theta, \lambda)$ for $i \in D_{\text{cal}}$

        Compute $\lambda(\theta) = \text{CORC}(\Lambda, \{L_i(\theta, \cdot)\}_{i \in D_{\text{cal}}}, B, \alpha, \epsilon, \phi)$        ▷ *CORC is defined in Algorithm 1*

        **for** $i \in D_{\text{pred}}$ **do**

            Define cost functions $\ell_i(\theta, \lambda) := \ell(x_i, y_i, \theta, \lambda)$

            Compute (sub)gradient of objective $\mathrm{d}\theta_i := \mathrm{d}\ell_i(\theta, \lambda(\theta)) / \mathrm{d}\theta$

        Update $\theta$ using (sub)gradients: $\sum_{i \in D_{\text{pred}}} \mathrm{d}\theta_i$

---

While the conformal OCE and CVaR risk control procedures we have proposed in this section enable controlling substantially more general risks than the original CRC framework, such a post-hoc risk control procedure may still come with a substantial cost upon ML model deployment. For example, applying CRC to control the false negative rate of a model for tumor segmentation may come at the cost of a large false positive rate (see Figure 1 in our experiments). Because CRC is designed to be applied post-hoc to a pretrained model, there is no existing approach to remedy this degradation in performance. To improve performance while guaranteeing controlled risk, a better approach would *train* or *fine-tune* models subject to a constraint enforcing risk control. Designing a methodology to accomplish this goal is the focus of the next section.

## 4 Conformal Risk Training

Returning to the setting described at the start of Section 2, the loss function $L$ typically depends on the output of an ML model with parameters $\theta \in \Theta$. To explicitly denote this relationship, we write $L(\theta, \lambda)$, and we define $h(\theta, \lambda)$ (equation 2) and $\tilde{h}_t(\theta, \lambda)$ (equation 4) accordingly in terms of $\{L_i(\theta, \lambda)\}_{i=1}^N$. The CRC (Section 2) and CORC (Section 3) procedures for picking $\lambda$ treat the model parameters $\theta$ as fixed. Thus, we call them "post-hoc" procedures.

However, as we show in our experiments, this separation of pre-training model parameters $\theta$ and performing risk control post-hoc leaves substantial room for improvement. Instead, in this section, we consider the problem of jointly optimizing the model parameters $\theta$ and the risk control parameter $\lambda$, which we call the *end-to-end optimal risk control problem*:

$$\min_{\theta \in \Theta, \, \lambda \in \Lambda} \mathbb{E}[\ell(\theta, \lambda)] \quad \text{s.t. } R[L(\theta, \lambda)] \leq \alpha, \tag{5}$$

where $\ell : \Theta \times \Lambda \to \mathbb{R}$ is a cost function that measures model performance and is differentiable almost everywhere. Note that we are specifically concerned with settings where the cost function $\ell$ depends on the risk control parameter $\lambda$. For a standard training objective such as cross-entropy where $\ell$ only depends on the model parameters $\theta$ but not on $\lambda$, the end-to-end optimal risk control problem (5) reduces to standard empirical risk minimization.

The following example concretely illustrates the roles of $\ell$ and $L$ for the problem of optimizing specificity in tumor image segmentation while controlling false negative rate.

**Example 1.** *In tumor image segmentation, an input image $X \in \mathcal{X} = \mathbb{R}^{d \times 3}$ (represented as a flattened array of RGB pixels) has a corresponding binary label $Y \in \mathcal{Y} = \{0, 1\}^d$, where 1 indicates the presence of a tumor at a given pixel location. Let $f_\theta : \mathcal{X} \to [0, 1]^d$ denote a segmentation model parameterized by $\theta$ that outputs a predicted probability that each pixel is tumorous. If $\lambda \in [0, 1]$ is the decision threshold, let $\hat{Y}(\theta, \lambda) \in \mathcal{Y}$ denote the binarized prediction, i.e., for all $j \in [d]$, $\hat{Y}(\theta, \lambda)_j := \mathbf{1}[f_\theta(X)_j \geq \lambda]$. Then, the fraction of false negative predictions is*

$$L(\theta, \lambda) := 1 - \frac{|Y \wedge \hat{Y}(\theta, \lambda)|}{|Y|} = 1 - \frac{1}{|Y|} \sum_{j : Y_j = 1} \mathbf{1}[f_\theta(X)_j \geq \lambda] = \frac{1}{|Y|} \sum_{j : Y_j = 1} \mathbf{1}[f_\theta(X)_j < \lambda], \tag{6}$$

*where $|\cdot|$ counts the number of ones in a binary vector. Clearly, $L$ is nondecreasing in $\lambda$. The expected loss $\mathbb{E}[L(\theta, \lambda)]$ gives the FNR of the model $f_\theta$, and we can pick $\lambda$ to control the FNR to be less than some target threshold $\alpha$.*

*However, in addition to controlling FNR, doctors and patients may also want to reduce false positives (i.e., optimize **specificity**). We can approximate the fraction of false positives in an image with*

$$\ell(\theta, \lambda) = \frac{1}{|\mathbf{1}_d - Y|} \sum_{j : Y_j = 0} \sigma\left(\frac{f_\theta(X)_j - \lambda}{T}\right),$$

*where $T$ is a temperature hyperparameter.*

As in the post-hoc CRC setting, we usually do not have access to the exact distribution of $L$ (nor the distribution of $\ell$). Instead, we only have a dataset of i.i.d. (or exchangeable) samples $\ell_i$ and $L_i$ for $i \in [N]$, from which we can form the *end-to-end optimal CORC problem*:

$$\min_{\theta \in \Theta, \lambda \in \Lambda} \frac{1}{N} \sum_{i=1}^{N} \ell_i(\theta, \lambda) \quad \text{s.t. } \tilde{h}_t(\theta, \lambda) \leq \alpha. \tag{7}$$

We can equivalently express (7) as a bi-level optimization problem. The outer problem minimizes over $\theta$, whereas the inner level chooses a risk-controlling $\lambda$:

$$\min_{\theta \in \Theta} \frac{1}{N} \sum_{i=1}^{N} \ell_i(\theta, \lambda(\theta)), \quad \text{where} \quad \lambda(\theta) := \arg\min_{\lambda \in \Lambda} \frac{1}{N} \sum_{i=1}^{N} \ell_i(\theta, \lambda) \ \text{s.t. } \tilde{h}_t(\theta, \lambda) \leq \alpha. \tag{8}$$

We shall assume that Assumption 4 holds, so in case the inner optimization problem in (8) is infeasible, we set $\lambda(\theta) = \lambda_{\min}$ to ensure risk control.

To solve (8) via gradient descent, we need to compute the gradient

$$\frac{d\ell_i}{d\theta}(\theta, \lambda(\theta)) = \frac{\partial \ell_i}{\partial \theta}(\theta, \lambda(\theta)) + \frac{\partial \ell_i}{\partial \lambda}(\theta, \lambda(\theta)) \cdot \frac{d\lambda}{d\theta}(\theta). \tag{9}$$

The key challenge lies in computing the derivative $\frac{d\lambda}{d\theta}(\theta)$. Fortunately, as we will show in Section 4.1, this derivative can be computed exactly in many common settings.

Assuming for now that we can compute $\frac{d\lambda}{d\theta}(\theta)$, we propose the *conformal risk training* method (Algorithm 2) for solving the general end-to-end CORC problem.[1] Inspired by the conformal training method [47], conformal risk training splits each minibatch of training data $D$ into two halves, $D_{\text{cal}}$ and $D_{\text{pred}}$. We use the pseudo-calibration set $D_{\text{cal}}$ to form the loss functions $L_i$ and compute the risk control parameter $\lambda(\theta)$, while we use $D_{\text{pred}}$ to form the cost functions $\ell_i$. Note that after training a model with conformal risk training, we use a fresh calibration dataset (and thus a fresh set of $L_1, \ldots, L_N$) to compute $\lambda(\theta)$ for use on a test input.

## 4.1 Computing the gradient in conformal risk training

This section discusses the computation of the gradient $\frac{d\lambda}{d\theta}(\theta)$. This is challenging in general, as it requires differentiating through the optimal solution of the lower-level CORC problem in (8), which may in general be nonconvex. Fortunately, it is possible to compute this gradient in many practical settings. In the following (informal) theorem, we give a high-level description of two cases in which we can compute this gradient; see Appendix C.4 for a formal statement of the theorem and its proof.

**Theorem** (Informal version of Theorem 3, Appendix C.4)**.** *Suppose Assumptions 1 and 2 hold and the inner problem in (8) is feasible. Under mild differentiability conditions, we may obtain a closed-form expression for $\frac{d\lambda}{d\theta}(\theta)$ in the following two cases:*

*(i) if $\{\ell_i\}_{i=1}^{N}$ are strictly decreasing in $\lambda$, and $\{B\} \cup \{L_i\}_{i=1}^{N}$ are piecewise constant in $\lambda$;*
*(ii) if $\{\ell_i\}_{i=1}^{N}$ are strictly convex or strictly monotone in $\lambda$, and $\{B\} \cup \{L_i\}_{i=1}^{N}$ are convex in $\lambda$.*

Note that when considering certain OCE risks like the CVaR in Theorem 3, the nondecreasing assumption (Assumption 2) can be relaxed to monotonicity (Assumption 5) in a similar manner as done in Theorem 2.

The gradient for conformal training [47] follows as a special case of (i) above (see Appendix E). We now show that case (i) also applies immediately to the FNR loss considered in Example 1.

---

[1]The name "conformal risk training" was coined by David Stutz on his blog. However, to the best of our knowledge, we are the first to actually develop a concrete methodology for end-to-end conformal risk control.

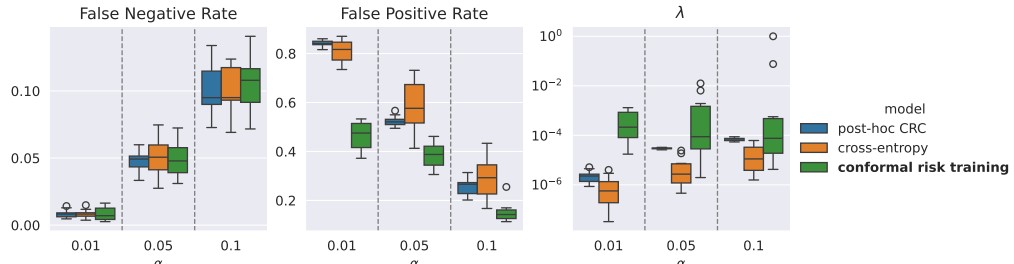

Figure 1: Results for the tumor image segmentation problem (Section 5.1) across different FNR thresholds $\alpha$ with 10 random seeds. **(left)** All three methods show FNR controlled at level $\alpha$. **(middle)** Whereas only applying CRC post-hoc results in very large FPR, our method (conformal risk training, in green) is able to significantly reduce FPR without sacrificing FNR. **(right)** Our method generally picks a higher classification threshold $\lambda$ than the baselines, suggesting that it is less conservative.

**Example 2.** *Recall the setting of Example 1. Given a dataset $D_{cal} = \{(X_i, Y_i)\}_{i=1}^N$ drawn exchangeably from $\mathcal{P}$, let $L_i(\theta, \lambda) := 1 - |Y_i \wedge \hat{Y}_i(\theta, \lambda)|/|Y_i|$. Clearly, every $L_i$ is bounded by $B = 1$. Following the conformal risk control procedure, using the form (6) of the loss, and noting that each $\ell_i$ is strictly decreasing in $\lambda$, we may pick*

$$\lambda(\theta) := \max \left\{ \lambda \in [0, 1] \left| \frac{1}{N+1} \left( 1 + \sum_{i=1}^N \sum_{j: (Y_i)_j = 1} \frac{1}{|Y_i|} \mathbf{1}[f_\theta(X_i)_j < \lambda] \right) \leq \alpha \right. \right\}. \quad (10)$$

*Assuming that all values $f_\theta(X_i)_j$ are unique, then according to the proof of Theorem 3(i) (see Appendix C.4), the optimal value $\lambda(\theta) = f_\theta(X_i)_j$ for some specific $(i, j)$. Thus, the gradient is*

$$\frac{\mathrm{d}\lambda(\theta)}{\mathrm{d}\theta} = \frac{\mathrm{d}}{\mathrm{d}\theta} f_\theta(X_i)_j.$$

## 5 Experiments

In this section, we present experimental results for our conformal risk training method on two problems: (1) controlling false negative rate in tumor image segmentation [6], and (2) controlling CVaR of losses in grid-scale battery storage operation [19]. Code to reproduce our results are available on GitHub,[2] and additional experimental results and details are reported in Appendices A and D.

### 5.1 Controlling false negative rate in tumor image segmentation

We adopt the colonoscopy gut polyp image segmentation problem setup explored in [6, Section 3.1] and described in Example 1. We use a pre-trained PraNet [23] as our model $f_\theta$, and we split images from 4 public datasets (CVC-ClinicDB [13], CVC-ColonDB [12], ETIS-LaribPolypDB [46], Kvasir-SEG [28]) into training, calibration, and test splits. In Figure 1, we compare the FNR and false positive rate (FPR) on the test set across three different models: (1) "post-hoc CRC" applied directly to the pre-trained PraNet; (2) "cross-entropy" refers to the fine-tuning PraNet using cross-entropy classification loss and then applying CRC; and (3) "conformal risk training" refers to fine-tuning PraNet using our method described in Section 4. For each model, we try 10 different random seeds for dividing the calibration and test splits, and we vary the target FNR $\alpha$ across three different values (0.01, 0.05, 0.1).

As Figure 1 shows, all three models have their expected FNR controlled at the target level $\alpha$. However, for the "post-hoc CRC" and "cross-entropy" baselines, applying post-hoc CRC comes at a significant cost to the FPR, reaching as high as 80% FPR when the target FNR is 1%. In contrast, our conformal risk training method reduces FPR on average by 23-42% across the $\alpha$ levels. Furthermore, our method yields larger average values of $\lambda$ than the baselines, suggesting that our method reduces conservativeness while maintaining the risk guarantee.

---

[2]https://github.com/chrisyeh96/conformal-risk-training

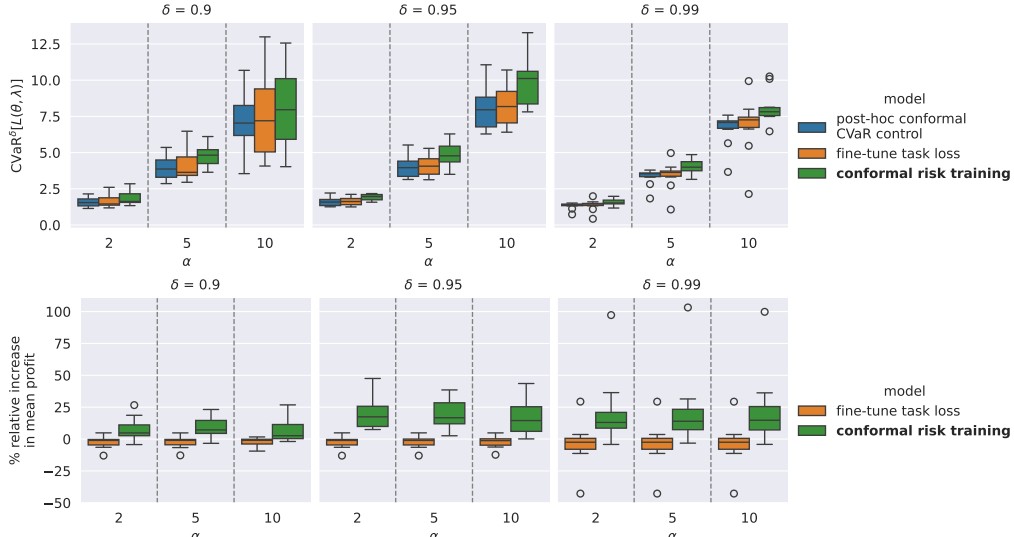

Figure 2: Results from the battery storage problem (Section 5.2) across different CVaR quantile levels $\delta$ and risk control thresholds $\alpha$, with 10 random seeds. **(top)** All three methods show CVaR risk controlled at the target level $\alpha$. **(bottom)** Comparison of the relative increase in profit (*i.e., negative task loss*) achieved by different methods over the post-hoc CRC baseline. Higher values are better.

## 5.2 Controlling CVaR tail risk from battery storage operations

Energy resource operators seek electricity price forecasting models which deliver optimal expected profit while ensuring that the risk of losses due to poor predictions is adequately controlled. We adopt a grid-scale battery operation problem [19], where a battery operator predicts electricity prices $Y \in \mathbb{R}^T$ over a $T$-step horizon and uses the predicted prices to decide how much to charge ($z^{\text{in}} \in \mathbb{R}^T$) or discharge ($z^{\text{out}} \in \mathbb{R}^T$) the battery, subject to constraints on the battery's state of charge expressed in terms of the net charge $z^{\text{net}} \in \mathbb{R}^T$. The input features $X$ include the past day's prices and temperature, the current day's energy load forecast, and other date-related features. The battery has capacity $C$, charging efficiency $\gamma$, and maximum charge/discharge rates $c^{\text{in}}$ and $c^{\text{out}}$. The task loss function $f$ represents the multiple objectives of maximizing profit, battery health by discouraging rapid charging/discharging (with weight $\epsilon^{\text{ramp}}$), and flexibility to participate in other markets by keeping the battery near half its capacity (with weight $\epsilon^{\text{flex}}$):

$$f(y, z) = (z^{\text{in}} - z^{\text{out}})^\top y + \epsilon^{\text{ramp}} \left( \left\| z^{\text{in}} \right\|^2 + \left\| z^{\text{out}} \right\|^2 \right) + \epsilon^{\text{flex}} \left\| z^{\text{net}} \right\|^2,$$

with the constraint set $\mathcal{Z}$ containing all $z := (z^{\text{in}}, z^{\text{out}}, z^{\text{net}}) \in \mathbb{R}^{3T}$ that satisfy the constraints

$$0 \leq z^{\text{in}} \leq c^{\text{in}}, \quad 0 \leq z^{\text{out}} \leq c^{\text{out}}, \quad -\frac{C}{2} \leq z^{\text{net}} \leq \frac{C}{2}, \quad \forall t \in [T] : z_t^{\text{net}} := \sum_{\tau=1}^{t} \gamma z_\tau^{\text{in}} - z_\tau^{\text{out}}.$$

Following [19], we set $T = 24$, $C = 1$, $\gamma = 0.9$, $c^{\text{in}} = 0.5$, $c^{\text{out}} = 0.2$, $\epsilon^{\text{flex}} = 0.1$, and $\epsilon^{\text{ramp}} = 0.05$.

If $z \in \mathcal{Z}$, then clearly $\lambda z \in \mathcal{Z}$ for all $\lambda \in \Lambda := [0, 1]$. We can thus implement a $\lambda$-dependent decision-maker via a "decision-scaling" rule $z_\theta(x, \lambda) = \lambda \hat{z}_\theta(x)$, where $\hat{z}_\theta(x) = \arg\min_{z \in \mathcal{Z}} f(\hat{y}_\theta(x), z)$ and $\hat{y}_\theta : \mathcal{X} \to \mathcal{Y}$ is a neural network pre-trained to minimize mean squared error in predicting electricity prices.[3] We use the task loss $\ell(\theta, \lambda) = f(Y, z_\theta(X, \lambda))$ as our training objective, which is strictly convex in $\lambda$, and we apply CVaR control to the financial risk term $L(\theta, \lambda) := \lambda (\hat{z}_\theta^{\text{in}}(X) - \hat{z}_\theta^{\text{out}}(X))^\top Y$. We construct a monotone-increasing upper bound $B(\lambda) := 100\lambda$ based on the empirical observation that our decision-maker $\hat{z}_\theta$ satisfies $(\hat{z}_\theta^{\text{in}}(x) - \hat{z}_\theta^{\text{out}}(x))^\top y \leq 100$ on our training and validation sets. Note that Theorem 2 enables us to control the CVaR of $L(\theta, \lambda)$, which may either be monotone nondecreasing or nonincreasing in $\lambda$; in contrast, the traditional CRC method (Proposition 1) does not apply, as it only allows for controlling the expectation of $L$ when it is nondecreasing.

---

[3]We do not train a model to directly map features $x$ to decisions $z$ because $z$ must satisfy constraints. Instead, we train $\hat{y}_\theta$ to predict electricity prices, and then define the decision $\hat{z}_\theta(x)$ as a function of $\hat{y}_\theta(x)$.

Table 1: Sensitivity of task loss and tail risk to relative changes in hyperparameter $t$ on the battery storage problem. Values given show mean $\pm$ 1 standard deviation of the task loss $\mathbb{E}_{(X,Y)\sim D}[f(Y, \lambda \hat{z}_\theta(X))]$ and financial tail risk $\mathrm{CVaR}^\delta_{(X,Y)\sim D}[\lambda(\hat{z}^{\text{in}}_\theta(X) - \hat{z}^{\text{out}}_\theta(X))^\top Y]$ for 10 models $\hat{y}_\theta$ trained with different random seeds. $t$ is calculated as $t = (1 + \Delta t)t_0$, where $t_0$ denotes the optimal value of $t$ tuned on the training set.

| | $\Delta t$ | $\alpha = 2$ | | | $\alpha = 5$ | | | $\alpha = 10$ | | |
| --- | --- | --- | --- | --- | --- | --- | --- | --- | --- | --- |
| | | $\delta = 0.9$ | 0.95 | 0.99 | $\delta = 0.9$ | 0.95 | 0.99 | $\delta = 0.9$ | 0.95 | 0.99 |
| $\mathbb{E}[\ell]$ | $-0.5$ | $-8.6 \pm 3.2$ | $-4.3 \pm 1.6$ | $-1.5 \pm 0.5$ | $-21.3 \pm 8.0$ | $-10.7 \pm 4.0$ | $-3.9 \pm 1.2$ | $-35.6 \pm 6.3$ | $-21.4 \pm 7.9$ | $-7.7 \pm 2.5$ |
| | $-0.1$ | $-8.6 \pm 3.2$ | $-4.3 \pm 1.6$ | $-1.8 \pm 0.5$ | $-21.3 \pm 7.9$ | $-10.7 \pm 4.0$ | $-4.4 \pm 1.3$ | $-35.6 \pm 6.3$ | $-21.5 \pm 7.9$ | $-8.9 \pm 2.6$ |
| | $0$ | $-8.6 \pm 3.2$ | $-4.3 \pm 1.6$ | $-1.8 \pm 0.5$ | $-21.3 \pm 7.9$ | $-10.7 \pm 4.0$ | $-4.5 \pm 1.3$ | $-35.6 \pm 6.3$ | $-21.5 \pm 7.9$ | $-9.0 \pm 2.6$ |
| | $0.1$ | $-8.6 \pm 3.2$ | $-4.3 \pm 1.6$ | $-1.8 \pm 0.5$ | $-21.3 \pm 7.9$ | $-10.7 \pm 4.0$ | $-4.6 \pm 1.2$ | $-35.6 \pm 6.3$ | $-21.5 \pm 7.9$ | $-9.1 \pm 2.5$ |
| | $0.5$ | $-8.6 \pm 3.2$ | $-4.3 \pm 1.6$ | $-1.7 \pm 0.4$ | $-21.3 \pm 7.9$ | $-10.7 \pm 4.0$ | $-4.1 \pm 1.0$ | $-35.6 \pm 6.3$ | $-21.5 \pm 7.9$ | $-8.3 \pm 2.0$ |
| $\mathrm{CVaR}^\delta[L]$ | $-0.5$ | $1.6 \pm 0.3$ | $1.6 \pm 0.3$ | $1.1 \pm 0.2$ | $3.9 \pm 0.8$ | $4.0 \pm 0.8$ | $2.8 \pm 0.6$ | $7.1 \pm 2.2$ | $8.0 \pm 1.5$ | $5.6 \pm 1.1$ |
| | $-0.1$ | $1.6 \pm 0.3$ | $1.6 \pm 0.3$ | $1.3 \pm 0.2$ | $3.9 \pm 0.8$ | $4.0 \pm 0.8$ | $3.3 \pm 0.6$ | $7.0 \pm 2.2$ | $8.1 \pm 1.5$ | $6.5 \pm 1.2$ |
| | $0$ | $1.6 \pm 0.3$ | $1.6 \pm 0.3$ | $1.3 \pm 0.2$ | $3.9 \pm 0.8$ | $4.0 \pm 0.8$ | $3.3 \pm 0.6$ | $7.0 \pm 2.2$ | $8.1 \pm 1.5$ | $6.6 \pm 1.2$ |
| | $0.1$ | $1.6 \pm 0.3$ | $1.6 \pm 0.3$ | $1.3 \pm 0.2$ | $3.9 \pm 0.8$ | $4.0 \pm 0.8$ | $3.4 \pm 0.6$ | $7.0 \pm 2.2$ | $8.1 \pm 1.5$ | $6.7 \pm 1.2$ |
| | $0.5$ | $1.6 \pm 0.3$ | $1.6 \pm 0.3$ | $1.2 \pm 0.3$ | $3.9 \pm 0.8$ | $4.0 \pm 0.8$ | $3.1 \pm 0.7$ | $7.0 \pm 2.2$ | $8.1 \pm 1.5$ | $6.2 \pm 1.5$ |
| $t_0$ | - | $0.0 \pm 0.0$ | $0.0 \pm 0.0$ | $0.8 \pm 0.3$ | $0.0 \pm 0.0$ | $0.0 \pm 0.0$ | $2.0 \pm 0.7$ | $0.1 \pm 0.1$ | $0.1 \pm 0.1$ | $4.1 \pm 1.5$ |

Because $L$ and $B$ are both convex in $\lambda$ and differentiable almost everywhere in $(\theta, \lambda)$, $\lambda(\theta)$ is the solution to a convex optimization problem, and Theorem 3(ii) allows us to compute the derivative $\mathrm{d}\lambda(\theta)/\mathrm{d}\theta$ by differentiating through the KKT conditions of the convex optimization problem [1].

We compare test set tail risk and task loss across three different models: (1) "post-hoc conformal CVaR control" applied directly to the pretrained price prediction model $\hat{y}_\theta$; (2) "fine-tune task loss" refers to fine-tuning $\hat{y}_\theta$ using a decision-focused task-loss [19] and then applying conformal CVaR control; and (3) "conformal risk training" refers to fine-tuning $\hat{y}_\theta$ using the procedure described in Section 4. For each model, we try 10 different random seeds for dividing the validation and test splits, and we vary the target CVaR tail risk $\alpha \in (2, 5, 10)$ and the quantile level $\delta \in (0.9, 0.95, 0.99)$.

As Figure 2 (top) shows, all three models have their CVaR tail risk controlled at the target level $\alpha$. However, in Figure 2 (bottom), it is clear that whereas fine-tuning using the task loss does not improve average task loss when controlling for tail risk, our conformal risk training method achieves between 7.2% and 22.6% mean improvement in task loss (*i.e.*, higher average profit) compared to the "post-hoc conformal CVaR control" baseline at all tested risk thresholds $\alpha$ and quantile levels $\delta$.

**Sensitivity to $t$ hyperparameter.** As noted in Section 3, conformal CVaR control requires picking a hyperparameter $t \in [B(\lambda_{\min}), \alpha]$, which we set to the value $t_0$ that yields the largest risk control parameter $\hat{\lambda}$ on the training set. To understand the sensitivity of conformal CVaR control to the choice of $t$, we computed task loss and the empirical CVaR of financial risk under both relative (Table 1) and absolute (Table 3) perturbations of $t$ away from $t_0$. Empirically, we find that the task loss and CVaR are not very sensitive to changes in $t$; the task loss from perturbed $t$ tends to be close to (within 1 standard deviation of) the task loss from $t_0$.

## 6 Conclusion

We have developed the conformal OCE risk control method, which is a strict generalization of the original CRC procedure. In particular, this allows us to directly control the CVaR tail risk, unlike previous works which only control expected losses or provide high-probability bounds. We have also developed conformal risk training, which generalizes the conformal training procedure from conformal prediction to the setting of conformal OCE risk control, and our experiments show significant improvements in model performance over applying post-hoc CRC alone.

**Limitations and future directions.** The main limitations of conformal OCE risk control are the same limitations that apply to standard CRC: the risk control guarantee only applies to monotone and exchangeable losses. For conformal risk training, while we derive the exact gradient in some common cases, we do not provide a complete characterization of when the gradient exists.

Future work may examine the tightness of the conformal OCE risk control bound and consider generalizations of CRC to other families of risk measures such as distortion [16] or coherent risk measures [7]. We believe that conformal OCE risk control will be of particular interest to high-stakes applications in finance, robotics, and LLM alignment, where provable tail risk guarantees are critical.

## Acknowledgments and Disclosure of Funding

This work was supported by NSF grants (CCF-2326609, CNS-2146814, CPS-2136197, CNS-2106403, NGSDI-2105648), an NSF Graduate Research Fellowship (DGE-2139433), gifts from Amazon, OpenAI, and Latitude AI, and the Caltech Resnick Sustainability Institute.

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

Figure 3: Predictions on 8 randomly selected images from the test set for the tumor image segmentation problem (Section 5.1). Black and white pixels indicate correct outputs. False positives are shaded teal; false negatives are shaded red.

# A    Additional experimental results

**Tumor image segmentation.**    Figure 3 shows a random selection of 8 images from the test set, along with the corresponding predictions made by the baseline methods ("post-hoc CRC" and "cross-entropy") compared to our "conformal risk training" method. Evidently, our method significantly reduces false positives (shown in teal) without significantly increasing false negatives (shown in red), especially when the false negative rate is controlled at a very low $\alpha$. Note that the marginal risk control guarantee offered by the CRC procedure (Algorithm 3) ensures that false negative rate will be

Table 2: Comparison of cross-entropy loss, false negative rate (FNR), and false positive rate (FPR) across baseline methods ("post-hoc CRC" and "cross-entropy") and our conformal risk training (CRT) method. The $\alpha$ values in the different columns indicate the level at which FNR is controlled by CRC. The $\alpha$ value in parenthesis following "CRT" indicates the FNR threshold enforced during conformal risk training.

| | cross-entropy loss | $\alpha = 0.01$ | | $\alpha = 0.05$ | | $\alpha = 0.1$ | |
| --- | --- | --- | --- | --- | --- | --- | --- |
| | | FNR | FPR | FNR | FPR | FNR | FPR |
| post-hoc CRC | $2.8 \pm 0.1$ | $0.009 \pm 0.003$ | $0.841 \pm 0.014$ | $0.048 \pm 0.008$ | $0.524 \pm 0.022$ | $0.101 \pm 0.019$ | $0.256 \pm 0.035$ |
| cross-entropy | $2.9 \pm 0.2$ | $0.008 \pm 0.003$ | $0.811 \pm 0.049$ | $0.050 \pm 0.014$ | $0.580 \pm 0.106$ | $0.101 \pm 0.017$ | $0.293 \pm 0.092$ |
| CRT ($\alpha = .01$) | $11.9 \pm 0.4$ | $0.008 \pm 0.005$ | $0.465 \pm 0.057$ | - | - | - | - |
| CRT ($\alpha = .05$) | $5.8 \pm 2.5$ | - | - | $0.049 \pm 0.014$ | $0.385 \pm 0.054$ | - | - |
| CRT ($\alpha = .1$) | $7.8 \pm 6.1$ | - | - | - | - | $0.106 \pm 0.021$ | $0.152 \pm 0.041$ |

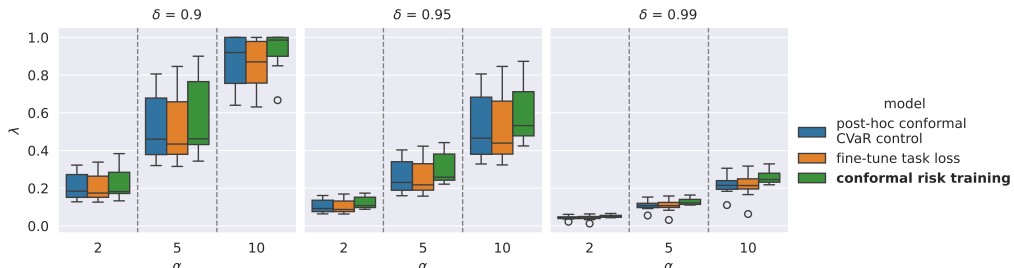

Figure 4: Values of the decision scaling factor $\lambda$ for the battery storage problem (Section 5.2) across different CVaR quantile levels $\delta$ and risk control thresholds $\alpha$, with 10 random seeds. Higher values indicate more aggressiveness and larger charge/discharge decisions.

controlled at the target level *on average* over test examples, while individual examples might exceed the target false negative rate. This is why in certain cases, our method ("conformal risk training") might exceed the false negative rate of other methods (e.g., in the 7th column of the $\alpha = 0.05$ case in Figure 3), while on average, it guarantees the target false negative rate.

We would also like to emphasize the decision-focused nature of our method. As shown in Figure 1, further training the pre-trained image segmentation model with the PraNet weighted cross-entropy loss [23] and then applying post-hoc CRC (shown in orange) does not improve model performance over directly applying post-hoc CRC to the pre-trained model (shown in blue). This observation is verified in Table 2, where we see no meaningful difference in performance between the "post-hoc CRC" and "cross-entropy" rows. In contrast, at a given false negative rate (FNR) risk level $\alpha$, our conformal risk training method achieves lower false positive rate (FPR) but incurs higher cross-entropy loss.

**Battery storage operations.** In Figure 4, we show values of the decision scaling factor $\lambda$ obtained through our conformal risk training approach compared with the baselines described in Section 5.2. In general across values of the CVaR quantile level $\delta$ and risk control threshold $\alpha$, it appears that our conformal risk training method yields values of $\lambda$ that are larger on average than the alternative methods. This implies that our method is able to learn to deploy *more aggressive* decisions, and employs a greater portion of the battery capacity when charging/discharging than the alternative methods. In other words, by learning to use a larger scaling factor $\lambda$ while preserving the risk constraint, the electricity price forecasting model trained via conformal risk training becomes more "calibrated" in regard to risk. In addition, as the value of $\delta$ increases, the scaling factor $\lambda$ decreases across all methods, reflecting the increased need to make more conservative decisions in settings where losses on the extreme tail must be controlled.

Conformal CVaR control performs better when the calibration set is larger. As shown in Figure 5, a larger calibration set generally reduces the task loss and achieves a tighter CVaR bound closer to $\alpha$ (i.e., not being overly conservative). The reason a larger calibration set size $N$ helps is because the term $\frac{1}{N+1}\tilde{B}_t(\lambda)$ appears in the expression for $\tilde{h}_t(\lambda)$. Increasing $N$ reduces the effect of the upper bound $B$ and enables choosing larger (less conservative) $\lambda$. However, there are diminishing gains as $N$ increases.

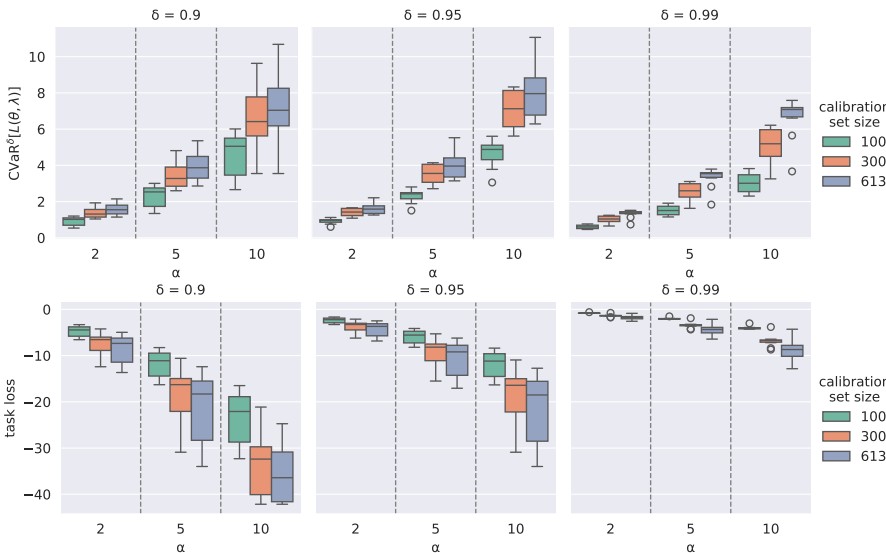

Figure 5: Comparison of financial tail risk (top) and task loss (bottom) as a function of calibration set size on the battery storage problem (Section 5.2), across different CVaR quantile levels $\delta$ and risk control thresholds $\alpha$, with 10 random seeds. Here, post-hoc conformal CVaR control is applied to the pre-trained prediction model $\hat{y}_\theta$.

Table 3: Sensitivity of task loss and tail risk to absolute changes in hyperparameter $t$ on the battery storage problem described in Section 5.2. Values given show mean $\pm$ 1 stddev of the task loss $\mathbb{E}_{(X,Y)\sim D}[f(Y, \lambda\hat{z}_\theta(X))]$ and the financial tail risk $\text{CVaR}^\delta_{(X,Y)\sim D}[\lambda(\hat{z}^{\text{in}}_\theta(X) - \hat{z}^{\text{out}}_\theta(X))^\top Y]$ for 10 models $\hat{y}_\theta$ trained with different random seeds. $t$ is calculated as $t = t_0 + \Delta t$, where $t_0$ denotes the optimal value of $t$ tuned on the training set. In the case that $t_0 + \Delta t \notin [B(\lambda_{\min}), \alpha]$, no values are given, as indicated by "-".

| | | $\alpha = 2$ | | | $\alpha = 5$ | | | $\alpha = 10$ | | |
| | $\Delta t$ | $\delta = 0.9$ | 0.95 | 0.99 | $\delta = 0.9$ | 0.95 | 0.99 | $\delta = 0.9$ | 0.95 | 0.99 |
|---|---|---|---|---|---|---|---|---|---|---|
| $\mathbb{E}[\ell]$ | $-5$ | - | - | - | - | - | - | - | - | $-6.5 \pm 0.9$ |
| | $-2$ | - | - | - | - | - | $-3.3 \pm 1.2$ | - | - | $-8.2 \pm 2.5$ |
| | $-1$ | - | - | $-1.3 \pm 0.2$ | - | - | $-4.1 \pm 1.2$ | - | - | $-9.1 \pm 2.3$ |
| | $0$ | $-8.6 \pm 3.2$ | $-4.3 \pm 1.6$ | $-1.8 \pm 0.5$ | $-21.3 \pm 7.9$ | $-10.7 \pm 4.0$ | $-4.5 \pm 1.3$ | $-35.6 \pm 6.3$ | $-21.5 \pm 7.9$ | $-9.0 \pm 2.6$ |
| | $1$ | $-6.8 \pm 1.9$ | $-4.4 \pm 1.1$ | $-1.0 \pm 0.7$ | $-19.9 \pm 6.7$ | $-11.3 \pm 3.6$ | $-4.5 \pm 0.8$ | $-35.4 \pm 6.3$ | $-22.2 \pm 7.5$ | $-9.3 \pm 2.1$ |
| | $2$ | $-0.7 \pm 0.3$ | $-0.8 \pm 0.0$ | $0.0 \pm 0.0$ | $-18.1 \pm 5.5$ | $-11.3 \pm 3.1$ | $-3.4 \pm 1.3$ | $-34.8 \pm 6.2$ | $-22.6 \pm 7.0$ | $-8.9 \pm 1.5$ |
| | $5$ | $0.0 \pm 0.0$ | $0.0 \pm 0.0$ | $0.0 \pm 0.0$ | $-0.4 \pm 0.9$ | $-1.9 \pm 0.7$ | $0.0 \pm 0.0$ | $-32.2 \pm 7.2$ | $-21.9 \pm 5.6$ | $-4.8 \pm 3.5$ |
| $\text{CVaR}^\delta[L]$ | $-5$ | - | - | - | - | - | - | - | - | $4.4 \pm 0.3$ |
| | $-2$ | - | - | - | - | - | $2.3 \pm 0.6$ | - | - | $5.9 \pm 1.0$ |
| | $-1$ | - | - | $0.9 \pm 0.1$ | - | - | $3.0 \pm 0.5$ | - | - | $6.6 \pm 0.7$ |
| | $0$ | $1.6 \pm 0.3$ | $1.6 \pm 0.3$ | $1.3 \pm 0.2$ | $3.9 \pm 0.8$ | $4.0 \pm 0.8$ | $3.3 \pm 0.6$ | $7.0 \pm 2.2$ | $8.1 \pm 1.5$ | $6.6 \pm 1.2$ |
| | $1$ | $1.3 \pm 0.3$ | $1.7 \pm 0.2$ | $0.8 \pm 0.6$ | $3.7 \pm 0.8$ | $4.3 \pm 0.6$ | $3.4 \pm 0.5$ | $7.0 \pm 2.1$ | $8.4 \pm 1.4$ | $6.9 \pm 1.0$ |
| | $2$ | $0.1 \pm 0.1$ | $0.3 \pm 0.1$ | $0.0 \pm 0.0$ | $3.4 \pm 0.8$ | $4.3 \pm 0.6$ | $2.6 \pm 1.2$ | $6.9 \pm 2.1$ | $8.6 \pm 1.3$ | $6.7 \pm 1.1$ |
| | $5$ | $0.0 \pm 0.0$ | $0.0 \pm 0.0$ | $0.0 \pm 0.0$ | $0.1 \pm 0.1$ | $0.7 \pm 0.3$ | $0.0 \pm 0.0$ | $6.2 \pm 1.7$ | $8.4 \pm 1.0$ | $3.8 \pm 3.0$ |
| $t_0$ | - | $0.0 \pm 0.0$ | $0.0 \pm 0.0$ | $0.8 \pm 0.3$ | $0.0 \pm 0.0$ | $0.0 \pm 0.0$ | $2.0 \pm 0.7$ | $0.1 \pm 0.1$ | $0.1 \pm 0.1$ | $4.1 \pm 1.5$ |

In Table 3, we compute the task loss and CVaR financial tail risk on the test set under perturbations of the hyperparameter $t$ from the tuned value $t_0$. Because Theorem 2 only holds when $t \in [B(\lambda_{\min}), \alpha]$, we exclude reporting results when the perturbed $t$ is outside of the required interval. We find that while our $t_0$ tuned based on the training set is not always optimal for the test set, it is generally close to the optimal.

# B Conformal Risk Control Algorithm

---

**Algorithm 3** Conformal Risk Control (CRC) for left-continuous, nondecreasing losses

---

**function** CRC(parameter space $\Lambda = [\lambda_{\min}, \lambda_{\max}]$, loss functions $\{L_i : \Lambda \to \mathbb{R}\}_{i=1}^N$, upper-bound $B : \Lambda \to \mathbb{R}$, risk threshold $\alpha \in \mathbb{R}$, numerical tolerance $\epsilon > 0$)

$\quad \underline{\lambda} \leftarrow \lambda_{\min}, \overline{\lambda} \leftarrow \lambda_{\max}$
$\quad$ **while** $\overline{\lambda} - \underline{\lambda} > \epsilon$ **do**
$\quad\quad \lambda \leftarrow (\underline{\lambda} + \overline{\lambda})/2$
$\quad\quad$ **if** $\frac{1}{N+1}\big(B(\lambda) + \sum_{i=1}^N L_i(\lambda)\big) \leq \alpha$ **then** $\underline{\lambda} \leftarrow \lambda$ **else** $\overline{\lambda} \leftarrow \lambda$
$\quad$ **return** $\underline{\lambda}$

---

# C Deferred proofs

Throughout the proofs, we sometimes rely on the property that a disutility function $\phi : \mathbb{R} \to \mathbb{R} \cup \{+\infty\}$ corresponding to an OCE risk measure (Definition 1) is a closed function. For completeness, we provide the definition of a closed function below.

**Definition 3** ([14]). *A function $f : \mathbb{R} \to \mathbb{R} \cup \{+\infty\}$ is* **closed** *if its epigraph* epi $f := \{(x, t) \in \mathbb{R}^2 \mid f(x) \leq t\}$ *is a closed set in* $\mathbb{R}^2$.

## C.1 Proof of Proposition 1

Define the set

$$\hat{\Lambda}' := \{\lambda \in \Lambda \mid \hat{R}(\lambda) \leq \alpha\}, \quad \text{where} \quad \hat{R}(\lambda) := \frac{1}{N+1} \sum_{i=1}^{N+1} L_i(\lambda),$$

and recall the definition of the set $\hat{\Lambda}$ from (2), which we state again below:

$$\hat{\Lambda} := \{\lambda \in \Lambda \mid h(\lambda) \leq \alpha\}, \quad \text{where} \quad h(\lambda) := \frac{1}{N+1}\left(B(\lambda) + \sum_{i=1}^N L_i(\lambda)\right).$$

Since $L_i \leq B$ almost surely for every $i \in [N]$, $\hat{R} \leq h$ almost surely. Therefore, $h(\lambda) \leq \alpha$ implies $\hat{R}(\lambda) \leq \alpha$ almost surely, so $\hat{\Lambda} \subseteq \hat{\Lambda}'$ almost surely.

Define $\overline{\lambda} := \sup \hat{\Lambda}$ and $\overline{\lambda}' := \sup \hat{\Lambda}'$. We follow the convention that $\sup \emptyset = -\infty$. Since $\hat{\Lambda} \subseteq \hat{\Lambda}'$ almost surely, we have $\overline{\lambda} \leq \overline{\lambda}'$ almost surely.

We now consider two cases based on whether $\hat{\Lambda}'$ is empty or not.

1. Suppose $\hat{\Lambda}' = \emptyset$. Then $\hat{\Lambda} = \emptyset$ almost surely, so it is (vacuously) true that for all $\lambda \in \hat{\Lambda}$, $\mathbb{E}[L_{N+1}(\lambda)] \leq \alpha$.

   If Assumption 3 holds, then $\overline{\lambda} = \overline{\lambda}' = -\infty$ almost surely, so $\hat{\lambda} = \max\{\lambda_{\min}, \overline{\lambda}\} = \lambda_{\min}$ almost surely. Therefore, $\mathbb{E}[L_{N+1}(\hat{\lambda})] \leq \alpha$.

2. Suppose $\hat{\Lambda}'$ is non-empty. In this case, $\overline{\lambda}' = \max \hat{\Lambda}'$ because (1) $\{L_i\}_{i=1}^{N+1}$ are left-continuous functions, so $\hat{R}$ is also left-continuous; (2) $\hat{\Lambda}'$ contains at least one point; and (3) we assume that $\Lambda$ is closed.

   Let $E$ be the random multiset of loss functions $\{L_1, \ldots, L_{N+1}\}$, and observe that $\overline{\lambda}'$ is a constant conditional on $E$. The random functions $L_i$ are exchangeable, which implies that

   $$L_{N+1}(\overline{\lambda}') \mid E \sim \text{Uniform}\left\{L_1(\overline{\lambda}'), \ldots, L_{N+1}(\overline{\lambda}')\right\}.$$

   Therefore,

   $$\mathbb{E}\left[L_{N+1}(\overline{\lambda}') \mid E\right] = \frac{1}{N+1} \sum_{i=1}^{N+1} L_i(\overline{\lambda}') = \hat{R}(\overline{\lambda}').$$

Note that $\hat{R}$ is a random function, because the $L_i$'s are random. By the law of total expectation (i.e., by further taking an expectation over the random multiset $E$), we have

$$\mathbb{E}\left[L_{N+1}(\bar{\lambda}')\right] = \mathbb{E}\left[\hat{R}(\bar{\lambda}')\right] \le \alpha$$

where the expectation is over randomness from $E$, and the inequality comes from the definitions of $\hat{\Lambda}'$ and $\bar{\lambda}'$.

Since $L_{N+1}$ is nondecreasing almost surely and $\bar{\lambda} \le \bar{\lambda}'$ almost surely, we have

$$\mathbb{E}\left[L_{N+1}(\bar{\lambda})\right] \le \mathbb{E}\left[L_{N+1}(\bar{\lambda}')\right] \le \alpha.$$

Since $\bar{\lambda} = \sup \hat{\Lambda}$ and $L_{N+1}$ is nondecreasing almost surely,

$$\forall \lambda \in \hat{\Lambda}: \quad \mathbb{E}[L_{N+1}(\lambda)] \le \mathbb{E}[L_{N+1}(\bar{\lambda})] \le \alpha.$$

If Assumption 3 holds, then observe that both $\mathbb{E}[L_{N+1}(\bar{\lambda})] \le \alpha$ and $\mathbb{E}[L_{N+1}(\lambda_{\min})] \le \alpha$. Therefore, $\mathbb{E}[L_{N+1}(\hat{\lambda})] \le \alpha$.

□

## C.2   Proof of Theorem 1

**Lemma 1.** *If $\phi : \mathbb{R} \to \mathbb{R} \cup \{+\infty\}$ is a disutility function corresponding to an optimized certainty equivalent risk (Definition 1), then $\phi$ is left-continuous.*

*Furthermore, if $\phi$ only takes on real values, then $\phi$ is continuous.*

*Proof of Lemma 1.* From the definition of an optimized certainty equivalent risk (Definition 1), $\phi : \mathbb{R} \to \mathbb{R} \cup \{+\infty\}$ is a closed, convex, nondecreasing function. By [14, Proposition 1.1.2], closedness implies $\phi$ is lower-semicontinuous.

Next, we show that since $\phi$ is lower-semicontinuous and nondecreasing, it is left-continuous. Fix any $x_0 \in \mathbb{R}$. Since $\phi$ is monotone, it has a left-limit at $x_0$, which we shall call $L$:

$$L := \lim_{x \uparrow x_0} \phi(x).$$

Since $\phi$ is nondecreasing, $L \le \phi(x_0)$. Choose any increasing sequence $\{x_n \in \mathbb{R}\}_{n \in \mathbb{N}}$ that converges to $x_0$. Then by definition of left-limit, we have

$$\lim_{n \to \infty} \phi(x_n) \to L.$$

Lower semicontinuity at $x_0$ yields

$$\phi(x_0) \le \liminf_{x \to x_0} \phi(x) \le \liminf_{n \to \infty} \phi(x_n) = \lim_{n \to \infty} \phi(x_n) = L.$$

We have thus shown $L \le \phi(x_0) \le L$, so $\phi(x_0) = L$, so $\phi$ is left-continuous at $x_0$. Since $x_0$ was arbitrary, $\phi$ is left-continuous on all of $\mathbb{R}$.

For the case where $\phi$ is real-valued (i.e., is never infinite), then it is a well-known fact that convex real-valued functions are continuous. □

*Proof of Theorem 1.* Fix any $t \in \mathbb{R}$. By Lemma 1, $\phi$ is left-continuous. Since $(\phi, \{L_i\}_{i=1}^{N+1}, B)$ are nondecreasing left-continuous functions, and the composition of nondecreasing left-continuous functions remains nondecreasing left-continuous, $(\{\tilde{L}_{i,t}\}_{i=1}^{N+1}, \tilde{B}_t)$ are nondecreasing left-continuous functions. Furthermore, we have $\tilde{B}_t(\lambda) \ge \tilde{L}_{i,t}(\lambda)$ almost surely.

For any $\hat{\lambda} \in \hat{\Lambda}_t$, we have

$$R[L_{N+1}(\hat{\lambda})] = \inf_{\hat{t} \in \mathbb{R}} \mathbb{E}\left[\hat{t} + \phi(L_{N+1}(\hat{\lambda}) - \hat{t})\right] \le \mathbb{E}\left[t + \phi(L_{N+1}(\hat{\lambda}) - t)\right] = \mathbb{E}[\tilde{L}_{N+1,t}(\hat{\lambda})] \le \alpha,$$

where the last inequality comes from Proposition 1. □

### C.3 Proof of Theorem 2

Theorem 2 states that CVaR risk can be controlled when the loss functions $L_i$ are monotone, as opposed to the stronger nondecreasing requirement from Proposition 1. In this section, we prove the more general result (Proposition 2) that when $L_i$ are monotone, any OCE risk measure whose disutility function can be written as the composition of another OCE risk measure's disutility $\phi_{\text{base}}$ and $[.]_+$ can also be controlled. We then show that Theorem 2 follows as corollary of this more general result.

**Lemma 2.** *If $f : \mathbb{R} \to \mathbb{R} \cup \{+\infty\}$ is a closed function (Definition 3), and $g : \mathbb{R} \to \mathbb{R}$ is continuous, then $f \circ g$ is closed.*

*Proof of Lemma 2.* Let $(x_n, t_n)$ be a sequence of points in $\text{epi}(f \circ g)$ such that $(x_n, t_n) \to (x, t)$. By definition of epigraph, $f(g(x_n)) \leq t_n$, so $(g(x_n), t_n) \in \text{epi } f$. Since $g$ is continuous, $g(x_n) \to g(x)$, and thus $(g(x_n), t_n) \to (g(x), t)$. Since $f$ is closed, $\text{epi } f$ is closed, so $(g(x), t) \in \text{epi } f$. In other words, $f(g(x)) \leq t$, so $(x, t) \in \text{epi}(f \circ g)$.

Thus, we have shown that every convergent sequence of points in $\text{epi}(f \circ g)$ converges within $\text{epi}(f \circ g)$, so $\text{epi}(f \circ g)$ is closed. Therefore, $f \circ g$ is closed. $\square$

**Proposition 2.** *Suppose that Assumptions 1 and 5 hold. Fix any $\alpha \in \mathbb{R}$, and let $\phi_{\text{base}} : \mathbb{R} \to \mathbb{R} \cup \{+\infty\}$ be a disutility function for some OCE risk measure. Let $R$ be the OCE risk measure corresponding to the transformed disutility function $\phi(x) := \phi_{\text{base}}([x]_+)$, and define $\hat{\Lambda}_t$ as in Theorem 1 using this transformed disutility function $\phi$. Then, for every $t \in [B(\lambda_{\min}), \alpha]$ and every $\lambda \in \hat{\Lambda}_t$, we have the risk control bound $R[L_{N+1}(\lambda)] \leq \alpha$.*

*Proof of Proposition 2.* First, we show that $\phi$ is a valid disutility function. By definition of a disutility function, $\phi_{\text{base}}$ is convex, closed, and nondecreasing. $\phi$ is convex because it is the composition of a convex, nondecreasing function $\phi_{\text{base}}$ and a convex function $[\cdot]_+$. $\phi$ is nondecreasing because it is the composition of two nondecreasing functions. By Lemma 2, $\phi$ is closed because it is the composition of a closed function $\phi_{\text{base}}$ and a continuous function $[\cdot]_+$. Furthermore, $\phi(0) = \phi_{\text{base}}([0]_+) = 0$. Finally, to show that $1 \in \partial\phi(0)$, we consider two cases. If $x < 0$, then $\phi(x) - \phi(0) = \phi(0) - \phi(0) = 0 > x$. If $x \geq 0$, then $\phi(x) - \phi(0) = \phi_{\text{base}}(x) \geq x$ since $1 \in \partial\phi_{\text{base}}(0)$. Thus, for all $x \in \mathbb{R}$, $\phi(x) - \phi(0) > x$, which shows that $1 \in \partial\phi(0)$. Therefore, $\phi$ is a valid disutility function.

Fix any $t \geq B(\lambda_{\min})$. Following Theorem 1, we define

$$\tilde{L}_{i,t}(\lambda) := t + \phi(L_i(\lambda) - t) \qquad \forall i \in [N+1]$$
$$\tilde{B}_t(\lambda) := t + \phi(B(\lambda) - t).$$

We will establish that $\tilde{L}_{i,t}$ is nondecreasing and left-continuous. By Assumption 5, $L_i$ is monotone in $\lambda$. If $L_i$ is nondecreasing in $\lambda$, then clearly $[L_i(\lambda) - t]_+$ is nondecreasing in $\lambda$ as well. If $L_i$ is decreasing in $\lambda$, observe that $L_i(\lambda_{\min}) \leq B(\lambda_{\min}) \leq t$ almost surely, so $[L_i(\lambda) - t]_+ = 0$ almost surely, for all $\lambda \in \Lambda$. Therefore, $[L_i(\lambda) - t]_+$ is almost surely nondecreasing in $\lambda$. By Assumption 1, $L_i$ is left-continuous, so $[L_i(\lambda) - t]_+$ is likewise left-continuous in $\lambda$. $\phi_{\text{base}}$ is also nondecreasing (by definition of disutility function) and left-continuous (by Lemma 1). The composition of two nondecreasing left-continuous functions remains nondecreasing and left-continuous, so $\phi(L_i(\lambda) - t) = \phi_{\text{base}}([L_i(\lambda) - t]_+)$ is nondecreasing and left-continuous in $\lambda$. Therefore, $\tilde{L}_{i,t}$ is nondecreasing and left-continuous.

By a similar argument, $\tilde{B}_t$ is also nondecreasing and left-continuous. Furthermore, since $B(\lambda) \geq L_i(\lambda)$ almost surely, and $\phi$ is nondecreasing, we have $\tilde{B}_t(\lambda) \geq \tilde{L}_{i,t}(\lambda)$ almost surely.

Thus, $\tilde{B}_t$ and $\{\tilde{L}_{i,t}\}_{i=1}^{N+1}$ satisfy Assumptions 1 and 2. Therefore, by Proposition 1, for every $\hat{\lambda} \in \hat{\Lambda}_t$,

$$R[L_{N+1}(\hat{\lambda})] = \inf_{\hat{t} \in \mathbb{R}} \mathbb{E}\left[\hat{t} + \phi\left(L_{N+1}(\hat{\lambda}) - \hat{t}\right)\right]$$
$$\leq \mathbb{E}\left[t + \phi\left(L_{N+1}(\hat{\lambda}) - t\right)\right] = \mathbb{E}\left[\tilde{L}_{N+1,t}(\hat{\lambda})\right]$$
$$\leq \alpha.$$

We remark that if $t \leq \alpha$ (which requires $\alpha \geq B(\lambda_{\min})$), then $\hat{\Lambda}_t$ is almost surely nonempty. Note that $\tilde{B}_t(\lambda_{\min}) = t \leq \alpha$. Every $\tilde{L}_{i,t}$ is almost surely bounded above by $\tilde{B}_t$, so $\tilde{L}_{i,t}(\lambda_{\min}) \leq \alpha$ almost surely. Therefore, $\lambda_{\min} \in \hat{\Lambda}_t$ almost surely.

On the other hand, if $t > \alpha$, then $\hat{\Lambda}_t$ is almost surely empty. This is because for every $\lambda \in \Lambda$, we have

$$\tilde{B}_t(\lambda) \geq \tilde{L}_{i,t}(\lambda) \geq \tilde{L}_{i,t}(\lambda_{\min}) \geq t > \alpha \quad \text{almost surely,}$$

so $\tilde{h}_t(\lambda) = \frac{1}{N+1}\left(\tilde{B}_t(\lambda) + \sum_{i=1}^{N} \tilde{L}_{i,t}(\lambda)\right) > \alpha$ almost surely. Therefore, for $t > \alpha$, Proposition 2 is almost surely vacuously true.

$\square$

*Proof of Theorem 2.* Theorem 2 follows as a corollary of Proposition 2, where we set $\phi_{\text{base}} = \phi_{\text{CVaR}^\delta}$. Then, $\phi = \phi_{\text{CVaR}^\delta}$:

$$\phi(x) := \phi_{\text{base}}([x]_+) = \phi_{\text{CVaR}^\delta}([x]_+) = \frac{1}{1-\delta}[[x]_+]_+ = \frac{1}{1-\delta}[x]_+ = \phi_{\text{CVaR}^\delta}(x).$$

$\square$

## C.4  Theorem 3 and its Proof

We start by stating and proving several technical lemmas that will help in the proof of Theorem 3. Throughout this section, we abbreviate "almost everywhere" as "a.e."

**Lemma 3.** *If $L : \Lambda \to \mathbb{R}$ is convex and $\phi : \mathbb{R} \to \mathbb{R} \cup \{+\infty\}$ is convex and nondecreasing, then the function $\tilde{L} : \Lambda \times \mathbb{R} \to \mathbb{R} \cup \{+\infty\}$ defined by $\tilde{L}(\lambda, t) := t + \phi(L(\lambda) - t)$ is convex in $(\lambda, t)$.*

*Proof.* Let $a \in [0, 1]$, and fix any $(\lambda_1, t_1), (\lambda_2, t_2) \in \Lambda \times \mathbb{R}$. Suppose $L$ is convex and $\phi$ is convex and nondecreasing. Then,

$$
\begin{aligned}
&\tilde{L}(a\lambda_1 + (1-a)\lambda_2, at_1 + (1-a)t_2) \\
&= at_1 + (1-a)t_2 + \phi(L(a\lambda_1 + (1-a)\lambda_2) - (at_1 + (1-a)t_2)) \\
&\leq at_1 + (1-a)t_2 + \phi(aL(\lambda_1) + (1-a)L(\lambda_2) - (at_1 + (1-a)t_2)) \\
&= at_1 + (1-a)t_2 + \phi(a(L(\lambda_1) - t_1) + (1-a)(L(\lambda_2) - t_2)) \\
&\leq at_1 + (1-a)t_2 + a\phi(L(\lambda_1) - t_1) + (1-a)\phi(L(\lambda_2) - t_2) \\
&= a\tilde{L}(\lambda_1, t_1) + (1-a)\tilde{L}(\lambda_2, t_2).
\end{aligned}
$$

$\square$

**Lemma 4.** *Suppose $L : \Theta \times \Lambda \to \mathbb{R}$ is a 0-1 step function in $\lambda$ of the form*

$$L(\theta, \lambda) = \mathbf{1}[g(\theta) < \lambda]$$

*where $g : \Theta \to \mathbb{R}$ is a continuous function whose level sets have measure zero. Then, $L$ is continuous a.e. in $\theta$.*

*Proof.* Fix $\lambda \in \mathbb{R}$, and define the preimages $U = g^{-1}((-\infty, \lambda))$, $V = g^{-1}((\lambda, \infty))$, and $W = g^{-1}(\{\lambda\})$. Clearly, $\Theta = U \cup V \cup W$. Since the level sets of $g$ have measure zero, $W$ has measure 0.

Since $g$ is continuous, $U$ and $V$ are open sets. For any $\theta \in U$, $L(\theta, \lambda) = 1$ is constant, so $L$ is continuous on $U$. Likewise, $L$ is constant and hence continuous on $V$.

Thus, $L$ is continuous in $\theta$ everywhere except on $W$, a set of measure 0. $\square$

**Lemma 5.** *Suppose $L : \Theta \times \Lambda \to \mathbb{R}$ is piecewise constant, left-continuous, and nondecreasing in $\lambda$ with the form*

$$L(\theta, \lambda) = c_0(\theta) + \sum_{j \in \mathbb{N}} c_j(\theta) \cdot \mathbf{1}[g_j(\theta) < \lambda]$$

*where $(c_j : \Theta \to \mathbb{R})_{j \in \mathbb{Z}_+}$ and $(g_j : \Theta \to \mathbb{R})_{j \in \mathbb{N}}$ are continuous functions, $(c_j(\theta))_{j \in \mathbb{N}}$ are nonnegative, and $(g_j(\theta))_{j \in \mathbb{N}}$ is nondecreasing. Assume that the level sets of $(g_j)_{j \in \mathbb{N}}$ have measure zero.*

Let $\phi : \mathbb{R} \to \mathbb{R}$ be an OCE disutility function, and let $t \in \mathbb{R}$. Then, the function $\tilde{L} : \Theta \times \Lambda \to \mathbb{R}$ defined as

$$\tilde{L}(\theta, \lambda) := t + \phi(L(\theta, \lambda) - t) = t + \phi\left(c_0(\theta) - t + \sum_{j \in \mathbb{N}} c_j(\theta) \cdot \mathbf{1}[g_j(\theta) < \lambda]\right)$$

is continuous a.e. in $\theta$, piecewise-constant left-continuous nondecreasing in $\lambda$, and can be written in the same form as $L$, i.e.,

$$\tilde{L}(\theta, \lambda) = \tilde{c}_0(\theta) + \sum_{j \in \mathbb{N}} \tilde{c}_j(\theta) \cdot \mathbf{1}[g_j(\theta) < \lambda]$$

where $(\tilde{c}_j)_{j \in \mathbb{Z}_+}$ are continuous functions and $(\tilde{c}_j(\theta))_{j \in \mathbb{N}}$ are nonnegative.

*Proof.* For all $k \in \mathbb{N}$, define

$$c_k'(\theta) := t + \phi\left(c_0(\theta) - t + \sum_{j=1}^{k} c_j(\theta)\right).$$

Since $\phi$ is a nondecreasing function and $(c_j(\theta))_{j \in \mathbb{N}}$ are nonnegative, $(c_j'(\theta))_{j \in \mathbb{Z}_+}$ is a nondecreasing sequence of reals. $(c_j)_{j \in \mathbb{Z}_+}$ are continuous by assumption, and $\phi$ is continuous by Lemma 1, so $(c_j')_{j \in \mathbb{Z}_+}$ are continuous functions.

Observe that $\tilde{L}$ is also piecewise-constant left-continuous nondecreasing in $\lambda$:

$$\tilde{L}(\theta, \lambda) = \begin{cases} c_0'(\theta), & \lambda \in (-\infty, g_1(\theta)] \\ c_k'(\theta), & \lambda \in (g_k(\theta), g_{k+1}(\theta)]. \end{cases}$$

Thus, we can write

$$\tilde{L}(\theta, \lambda) = \tilde{c}_0(\theta) + \sum_{j \in \mathbb{N}} \tilde{c}_j(\theta) \cdot \mathbf{1}[g_j(\theta) < \lambda],$$

where

$$\tilde{c}_0(\theta) = c_0'(\theta), \qquad \tilde{c}_j(\theta) := c_j'(\theta) - c_{j-1}'(\theta) \quad \forall j \in \mathbb{N}.$$

Because $(c_j'(\theta))_{j \in \mathbb{Z}_+}$ are nondecreasing, $(\tilde{c}_j(\theta))_{j \in \mathbb{N}}$ are nonnegative. Furthermore, $(\tilde{c}_j)_{j \in \mathbb{Z}_+}$ inherit continuity from $(c_j')_{j \in \mathbb{Z}_+}$.

By Lemma 4, each $\mathbf{1}[g_j(\theta) < \lambda]$ is continuous a.e. in $\theta$. The product of continuous a.e. functions is continuous a.e., so each term $\tilde{c}_j(\theta) \cdot \mathbf{1}[g_j(\theta) < \lambda]$ is continuous a.e. in $\theta$. The countable sum of continuous a.e. functions remains continuous a.e., so $\tilde{L}$ is continuous a.e. in $\theta$. $\qquad\square$

Now, we are ready to state the formal assumptions needed for Theorem 3.

**Assumption 6.** *The functions $B$ and $\{L_i\}_{i=1}^{N}$ are of the form*

$$B(\lambda) = b_0 + \sum_{j \in \mathbb{N}} b_j \cdot \mathbf{1}[d_j < \lambda], \qquad L_i(\theta, \lambda) = c_{i,0}(\theta) + \sum_{j \in \mathbb{N}} c_{i,j}(\theta) \cdot \mathbf{1}[g_{i,j}(\theta) < \lambda]$$

*where*

1. *$(c_{i,j} : \Theta \to \mathbb{R})_{i \in [N], j \in \mathbb{Z}_+}$ and $(g_{i,j} : \Theta \to \mathbb{R})_{i \in [N], j \in \mathbb{N}}$ are continuous functions;*
2. *$(b_j \in \mathbb{R})_{j \in \mathbb{N}}$ and $(c_{i,j}(\theta))_{i \in [N], j \in \mathbb{N}}$ are nonnegative;*
3. *$(d_j \in \mathbb{R})_{j \in \mathbb{N}}$ and $(g_{i,j}(\theta))_{j \in \mathbb{N}}$ for all $i$ are nondecreasing sequences, and $\{d_j\}_{j \in \mathbb{N}} \cup \{g_{i,j}(\theta)\}_{i \in [N], j \in \mathbb{N}}$ are all unique; and*
4. *$(g_{i,j})_{i \in [N], j \in \mathbb{N}}$ are differentiable a.e., and all of their level sets have measure zero.*

**Assumption 7.** *Suppose that the inner problem in (8) exhibits strong duality (e.g., Slater's condition holds). Let $\mu(\theta)$ denote the optimal Lagrange multiplier arising from the dual problem to (8). Define*

$$\tilde{\ell}(\theta, \lambda) := \begin{cases} \lambda, & \text{if } \sum_{i=1}^{N} \ell_i \text{ is strictly increasing in } \lambda \\ -\lambda, & \text{if } \sum_{i=1}^{N} \ell_i \text{ is strictly decreasing in } \lambda \ . \\ \frac{1}{N} \sum_{i=1}^{N} \ell_i(\theta, \lambda), & \text{if } \sum_{i=1}^{N} \ell_i \text{ is strictly convex in } \lambda \end{cases}$$

*The following regularity conditions hold:*

1. $\frac{\partial}{\partial \lambda}\tilde{\ell}$ exists and is continuously differentiable in $(\theta, \lambda)$ in a neighborhood around $(\theta, \lambda(\theta))$;
2. $B \cup \{L_i\}_{i=1}^N$ are twice continuously differentiable in $(\theta, \lambda)$ in a neighborhood around $(\theta, \lambda(\theta))$;
3. $\phi$ is twice continuously differentiable in neighborhoods around $B(\lambda(\theta)) - t$ and $L_i(\theta, \lambda(\theta)) - t$ for each $i \in [N]$; and
4. $\begin{bmatrix} \frac{\partial^2}{\partial \lambda^2}\tilde{\ell}(\theta, \lambda(\theta)) + \mu(\theta) \cdot \frac{\partial^2}{\partial \lambda^2}\tilde{h}_t(\theta, \lambda(\theta)) & \frac{\partial \tilde{h}_t}{\partial \lambda}(\theta, \lambda(\theta)) \\ \mu(\theta) \cdot \frac{\partial \tilde{h}_t}{\partial \lambda}(\theta, \lambda(\theta)) & \tilde{h}_t(\theta, \lambda(\theta)) - \alpha \end{bmatrix}$ is invertible.

With these assumptions stated, we now state the full version of Theorem 3.

**Theorem 3.** *Let $\phi : \mathbb{R} \to \mathbb{R}$ be an OCE disutility function. Suppose Assumptions 1 and 2 hold and that the inner problem in (8) is feasible.*

(i) *Suppose $\sum_{i=1}^N \ell_i$ is strictly decreasing in $\lambda$; the set $\{\theta \in \Theta \mid \tilde{h}_t(\theta, \lambda) = \alpha\}$ has measure zero for all $\lambda$; and $\{B\} \cup \{L_i\}_{i=1}^N$ are piecewise constant in $\lambda$. Under the standard differentiability and regularity conditions in Assumption 6, we may obtain a closed-form expression for $\frac{d\lambda}{d\theta}$ a.e.*

(ii) *Suppose $\sum_{i=1}^N \ell_i$ is strictly convex or strictly monotone in $\lambda$, and $\{B\} \cup \{L_i\}_{i=1}^N$ are convex in $\lambda$. Under the standard differentiability and regularity conditions in Assumption 7, the derivative $\frac{d\lambda}{d\theta}(\theta)$ exists and has a closed-form expression.*

*Proof.* **Setting (i).** By Lemma 5, each $\tilde{L}_{i,t}(\theta, \lambda) := t + \phi(L_i(\theta, \lambda) - t)$ is piecewise-constant left-continuous nondecreasing in $\lambda$ and can be written in the form

$$\tilde{L}_{i,t}(\theta, \lambda) = \tilde{c}_{i,0}(\theta) + \sum_{j \in \mathbb{N}} \tilde{c}_{i,j}(\theta) \cdot \mathbf{1}[g_{i,j}(\theta) < \lambda],$$

where $(\tilde{c}_{i,j})_{j \in \mathbb{Z}_+}$ are continuous functions and $(\tilde{c}_{i,j}(\theta))_{j \in \mathbb{N}}$ are nonnegative. Similarly, $\tilde{B}_t$ is piecewise-constant left-continuous nondecreasing and can be written in the form

$$\tilde{B}_t(\lambda) = \tilde{b}_0 + \sum_{j \in \mathbb{N}} \tilde{b}_j \cdot \mathbf{1}[d_j < \lambda]$$

where $(\tilde{b}_j)_{j \in \mathbb{N}}$ are nonnegative. Thus, $\tilde{h}_t(\theta, \lambda) := \frac{1}{N+1}\left(\tilde{B}_t(\lambda) + \sum_{i=1}^N \tilde{L}_{i,t}(\theta, \lambda)\right)$ is piecewise-constant left-continuous nondecreasing in $\lambda$ and can be written in the form

$$\tilde{h}_t(\theta, \lambda) = c_0^{(h)}(\theta) + \sum_{j \in \mathbb{N}} c_j^{(h)}(\theta) \cdot \mathbf{1}[g_j^{(h)}(\theta) < \lambda].$$

The sequence of steps is given by

$$(g_j^{(h)}(\theta))_{j \in \mathbb{N}} = \text{sort\_ascending}(\{d_j\}_{j \in \mathbb{N}} \cup \{g_{i,j}(\theta)\}_{i \in [N], j \in \mathbb{N}}),$$

which yields a strictly increasing sequence because $\{d_j\}_{j \in \mathbb{N}} \cup \{g_{i,j}(\theta)\}_{i \in [N], j \in \mathbb{N}}$ are all unique by assumption. The nonnegative coefficients $(c_j^{(h)}(\theta))_{j \in \mathbb{Z}_+}$ are defined appropriately in terms of $(\tilde{b}_j)_{j \in \mathbb{Z}_+}$ and $(\tilde{c}_{i,j})_{i \in [N], j \in \mathbb{Z}_+}$, and $(c_j^{(h)})_{j \in \mathbb{Z}_+}$ inherit their continuity.

The functions $(g_j^{(h)})_{j \in \mathbb{N}}$ are differentiable a.e. and continuous; these properties are inherited from $(d_j)_{j \in \mathbb{N}}$ (which are constant) and $(g_{i,j})_{i \in [N], j \in \mathbb{N}}$ (which are differentiable a.e. and continuous by assumption).

By Lemma 5, each $\tilde{L}_{i,t}$ is continuous a.e. in $\theta$. $\tilde{B}_t$ is constant in $\theta$, and therefore also continuous in $\theta$. $\tilde{h}_t$ is a countable, weighted sum of $\tilde{B}_t$ and $\tilde{L}_{i,t}$ for $i \in [N]$, so it is continuous a.e. in $\theta$.

Recall from (8) that

$$\lambda(\theta) := \arg\min_{\lambda \in \Lambda} \frac{1}{N}\sum_{i=1}^N \ell_i(\theta, \lambda) \ \text{ s.t. } \tilde{h}_t(\theta, \lambda) \leq \alpha.$$

Because each $\ell_i$ is strictly decreasing in $\lambda$, we may equivalently write the optimization problem as maximizing $\lambda$ subject to the constraint on $\tilde{h}_t$ and leave the optimal solution $\lambda(\theta)$ unaffected:

$$\lambda(\theta) = \max_{\lambda \in \Lambda} \lambda \ \text{ s.t. } \tilde{h}_t(\theta, \lambda) \leq \alpha.$$

By assumption, the optimization problem is feasible and $\tilde{h}_t(\theta, \lambda) \neq \alpha$ $\theta$-almost everywhere. Thus, for all $\theta$ except on a negligible set, one of the two following cases must hold:

1. $\tilde{h}_t(\theta, \lambda) < \alpha$ for every $\lambda \in \Lambda$.

   Then, $\lambda(\theta) = \max \Lambda$. We showed above that $\tilde{h}_t(\theta, \max \Lambda)$ is continuous a.e. in $\theta$. Thus, for $\theta$ a.e., there exists a neighborhood $\mathcal{N}(\theta)$ around $\theta$ for which $\tilde{h}_t(\theta', \max \Lambda) < \alpha$ for all $\theta' \in \mathcal{N}(\theta)$. Thus, $\lambda(\theta) = \max \Lambda$ for all $\theta' \in \mathcal{N}(\theta)$, so $\frac{d\lambda}{d\theta}(\theta) = \frac{d}{d\theta} \max \Lambda = 0$.

2. There exists an index $k \in \mathbb{N}$ such that

$$c_0^{(h)}(\theta) + \sum_{j=1}^{k} c_j^{(h)}(\theta) \cdot \mathbf{1}[g_j^{(h)}(\theta) < \lambda(\theta)] < \alpha,$$

$$c_0^{(h)}(\theta) + \sum_{j=1}^{k+1} c_j^{(h)}(\theta) \cdot \mathbf{1}[g_j^{(h)}(\theta) < \lambda(\theta)] > \alpha.$$

   In this case, because the objective seeks to maximize $\lambda$ and $(g_j^{(h)}(\theta))_{j \in \mathbb{N}}$ is a strictly increasing sequence, $\lambda(\theta) = g_{k+1}^{(h)}(\theta)$.

   As shown above, $g_{k+1}^{(h)}$ is differentiable a.e. Now, suppose that $g_{k+1}^{(h)}$ is differentiable at $\theta$. Consider any entry $\theta_q$ of the parameter vector $\theta$, and let $e_q$ denote the standard unit basis vector along coordinate $q$. Then the partial derivative of $\lambda$ with respect to $\theta_q$ is

$$\frac{\partial \lambda}{\partial \theta_q}(\theta) = \lim_{s \to 0} \frac{\lambda(\theta + se_q) - \lambda(\theta)}{s}$$

$$= \lim_{s \to 0} \frac{\lambda(\theta + se_q) - g_{k+1}^{(h)}(\theta)}{s}$$

$$= \lim_{s \to 0} \frac{g_{k+1}^{(h)}(\theta + se_q) - g_{k+1}^{(h)}(\theta)}{s}$$

$$= \frac{\partial g_{k+1}^{(h)}}{\partial \theta_q}(\theta).$$

   The third equality comes from observing that $(c_j^{(h)})_{j \in \mathbb{Z}_+}$ and $(g_j^{(h)})_{j \in \mathbb{N}}$ are continuous functions, so there exists an $\epsilon > 0$ such that for all $s \in [0, \epsilon]$,

$$c_0^{(h)}(\theta + se_q) + \sum_{j=1}^{k} c_j^{(h)}(\theta + se_q) \cdot \mathbf{1}[g_j^{(h)}(\theta + se_q) < \lambda(\theta + se_q)] < \alpha$$

$$c_0^{(h)}(\theta + se_q) + \sum_{j=1}^{k+1} c_j^{(h)}(\theta + se_q) \cdot \mathbf{1}[g_j^{(h)}(\theta + se_q) < \lambda(\theta + se_q)] > \alpha.$$

   Thus, for all $s \in [0, \epsilon]$, $\lambda(\theta + se_q) = g_{k+1}^{(h)}(\theta + se_q)$.

   Since $\frac{\partial \lambda}{\partial \theta_q} = \frac{\partial g_{k+1}^{(h)}}{\partial \theta_q}$ for all coordinates $q$, we have $\frac{d\lambda}{d\theta}(\theta) = \frac{dg_{k+1}^{(h)}}{d\theta}(\theta)$.

Therefore, we have derived a closed-form expression for $\frac{d\lambda}{d\theta}(\theta)$ almost everywhere.

**Setting (ii).** Since $\phi$ is convex and nondecreasing, $\{\tilde{B}_t\} \cup \{\tilde{L}_{i,t}\}_{i=1}^{N}$ are convex in $\lambda$ by Lemma 3. Therefore, $\tilde{h}_t$ is convex in $\lambda$.

Recall from (8) that

$$\lambda(\theta) := \arg\min_{\lambda \in \Lambda} \frac{1}{N} \sum_{i=1}^{N} \ell_i(\theta, \lambda) \text{ s.t. } \tilde{h}_t(\theta, \lambda) \leq \alpha.$$

If $\sum_{i=1}^{N} \ell_i$ is strictly increasing (or decreasing) in $\lambda$ but not strictly convex in $\lambda$, observe that we may replace the objective $\frac{1}{N} \sum_{i=1}^{N} \ell_i$ with simply $\lambda$ (or $-\lambda$) in (8) without changing the solution $\lambda(\theta)$. Concretely, let

$$\tilde{\ell}(\theta, \lambda) := \begin{cases} \lambda, & \text{if } \sum_{i=1}^{N} \ell_i \text{ is strictly increasing in } \lambda \\ -\lambda, & \text{if } \sum_{i=1}^{N} \ell_i \text{ is strictly decreasing in } \lambda \\ \frac{1}{N} \sum_{i=1}^{N} \ell_i(\theta, \lambda), & \text{if } \sum_{i=1}^{N} \ell_i \text{ is strictly convex in } \lambda \end{cases}$$

Then,

$$\lambda(\theta) = \arg\min_{\lambda \in \Lambda} \tilde{\ell}(\theta, \lambda) \text{ s.t. } \tilde{h}_t(\theta, \lambda) \leq \alpha. \tag{11}$$

Since $\lambda(\theta)$ is the solution to a convex optimization problem whose objective $\tilde{\ell}$ is either strictly convex or convex and strictly monotone, the solution is unique. Every convex optimization problem can be equivalently reformulated as a convex conic problem [1]. Then, the implicit function theorem, applied to the KKT equality conditions, gives sufficient conditions for $\frac{d\lambda}{d\theta}(\theta)$ to exist [3, 2, 1].

Specifically, define the Lagrangian $\mathcal{L} : \Lambda \times \mathbb{R} \times \Theta \to \mathbb{R}$ and the function $g : \Lambda \times \mathbb{R} \times \Theta \to \mathbb{R}^2$ by

$$\mathcal{L}(\lambda, \mu, \theta) := \tilde{\ell}(\theta, \lambda) + \mu \cdot (\tilde{h}_t(\theta, \lambda) - \alpha)$$

$$g(\lambda, \mu, \theta) := \begin{bmatrix} \frac{\partial \mathcal{L}}{\partial \lambda}(\lambda, \mu, \theta) \\ \mu \cdot (\tilde{h}_t(\theta, \lambda) - \alpha) \end{bmatrix}.$$

The KKT equality conditions for primal-dual optimality of $(\lambda(\theta), \mu(\theta))$ are precisely given by

$$g(\lambda(\theta), \mu(\theta), \theta) = \mathbf{0}.$$

By the implicit function theorem [8], if

(a) (11) satisfies strong duality (*e.g.*, if Slater's condition holds);
(b) $g(\lambda(\theta), \mu(\theta), \theta) = \mathbf{0}$;
(c) $g$ is continuous differentiable in $(\lambda, \mu, \theta)$ in a neighborhood around $(\lambda(\theta), \mu(\theta), \theta)$; and
(d) $\frac{\partial g}{\partial(\lambda, \mu)}(\lambda(\theta), \mu(\theta), \theta)$ is invertible,

then the derivative $\frac{d\lambda}{d\theta}(\theta)$ exists and is given by

$$\begin{bmatrix} \frac{d\lambda}{d\theta}(\theta) \\ \frac{d\mu}{d\theta}(\theta) \end{bmatrix} = - \left[ \frac{\partial g}{\partial(\lambda, \mu)}(\lambda(\theta), \mu(\theta), \theta) \right]^{-1} \frac{\partial g}{\partial \theta}(\lambda(\theta), \mu(\theta), \theta).$$

Assumption 7 directly satisfies condition (a), and the KKT equality conditions satisfy condition (b). Condition (c) is satisfied if $\tilde{h}_t$, $\frac{\partial \tilde{\ell}}{\partial \lambda}$, and $\frac{\partial \tilde{h}_t}{\partial \lambda}$ are continuously differentiable in $(\theta, \lambda)$ in a neighborhood around $(\theta, \lambda(\theta))$. Assumption 7.1 guarantees that this holds for $\frac{\partial \tilde{\ell}}{\partial \lambda}$, and Assumptions 7.2/7.3 guarantee that this holds for $\tilde{h}_t$ and $\frac{\partial h_t}{\partial \lambda}$.

Finally, note that

$$\frac{\partial g}{\partial(\lambda, \mu)}(\lambda(\theta), \mu(\theta), \theta) = \begin{bmatrix} \frac{\partial^2}{\partial \lambda^2}\tilde{\ell}(\theta, \lambda(\theta)) + \mu(\theta) \cdot \frac{\partial^2}{\partial \lambda^2}\tilde{h}_t(\theta, \lambda(\theta)) & \frac{\partial \tilde{h}_t}{\partial \lambda}(\theta, \lambda(\theta)) \\ \mu(\theta) \cdot \frac{\partial \tilde{h}_t}{\partial \lambda}(\theta, \lambda(\theta)) & \tilde{h}_t(\theta, \lambda(\theta)) - \alpha \end{bmatrix},$$

which Assumption 7.4 specifies is invertible, thus satisfying condition (d). $\qquad \square$

We now briefly remark on Theorem 3(i)'s use of Assumption 6.4 and its relevance in practice. Recall that Assumption 6.4 assumes $g_{i,j} : \Theta \to \mathbb{R}$ is differentiable a.e. and its level sets have measure zero. The following two lemmas show that both of these conditions are easily satisfied if $g_{i,j}$ is a neural network with Lipschitz continuous activation functions and a bias term in its final output layer. For instance, in Example 2, $g_{i,j}(\theta) = f_\theta(X_i)_j$ is the output of a neural network $f_\theta$ (albeit a neural network with a more complex U-Net [43, 23] architecture than a MLP).

**Lemma 6.** *A multi-layer perceptron (MLP) with Lipschitz continuous activation functions is locally Lipschitz continuous and therefore differentiable a.e. with respect to its parameters.*

*Specifically, let $g_K : \Theta \to \mathbb{R}^n$ denote a $K$-layer MLP for some $K \in \mathbb{N}$, defined recursively as*

$$g_k(\theta) := g_k(\theta_{1:k}) := \sigma_k(W_k \cdot g_{k-1}(\theta_{1:k-1}) + b_k) \qquad \forall k \in [K]$$
$$g_0 \in \mathbb{R}^d \text{ is the fixed input to the MLP.}$$

*Here, $(W_k, b_k)$ are the weight matrix and bias vector in each layer $k$, and we use the shorthand notation $\theta_{1:k} := (W_{1:k}, b_{1:k})$, and $\theta := \theta_{1:K}$. If each $\sigma_k$ is a Lipschitz continuous activation function, then $g_K$ is locally Lipschitz continuous and therefore differentiable a.e. in $\theta$.*

*Proof.* We use induction on $k$. For the base case $k = 0$, $g_0$ is constant and therefore (locally) Lipschitz continuous.

Now, fix some $\theta$ and suppose $g_{k-1}$ is locally Lipschitz continuous in a neighborhood $\mathcal{N}(\theta)$ around $\theta$ with local Lipschitz constant $L_{k-1}$:

$$\forall \theta' \in \mathcal{N}(\theta): \quad \|g_{k-1}(\theta') - g_{k-1}(\theta)\| \le L_{k-1} \|\theta' - \theta\|.$$

Define $\bar{n} := \sup_{\theta' \in \mathcal{N}(\theta)} \|\theta'\| < \infty$ and $\bar{g}_{k-1} := \|g_{k-1}(\theta)\| + L_{k-1} \operatorname{diam}(\mathcal{N}(\theta))$ so that $\|g_{k-1}(\theta')\| \le \bar{g}_{k-1}$ for all $\theta' \in \mathcal{N}(\theta)$. Let $M_k$ be the Lipschitz constant of $\sigma_k$. Then, for all $\theta, \theta'$:

$$\begin{aligned}
&\|g_k(\theta') - g_k(\theta)\| \\
&= \|\sigma_k(W_k' \cdot g_{k-1}(\theta') + b_k') - \sigma_k(W_k \cdot g_{k-1}(\theta) + b_k)\| \\
&\le M_k \|W_k' \cdot g_{k-1}(\theta') + b_k' - (W_k \cdot g_{k-1}(\theta) + b_k)\| \\
&\le M_k \left(\|W_k' \cdot g_{k-1}(\theta') - W_k \cdot g_{k-1}(\theta)\| + \|b_k' - b_k\|\right) \\
&\le M_k \left(\|W_k' \cdot g_{k-1}(\theta') - W_k \cdot g_{k-1}(\theta)\| + \|\theta' - \theta\|\right).
\end{aligned}$$

We have

$$\begin{aligned}
&\|W_k' \cdot g_{k-1}(\theta') - W_k \cdot g_{k-1}(\theta)\| \\
&= \|W_k' \cdot g_{k-1}(\theta') - W_k' \cdot g_{k-1}(\theta) + W_k' \cdot g_{k-1}(\theta) - W_k \cdot g_{k-1}(\theta)\| \\
&\le \|W_k' \cdot g_{k-1}(\theta') - W_k' \cdot g_{k-1}(\theta)\| + \|W_k' \cdot g_{k-1}(\theta) - W_k \cdot g_{k-1}(\theta)\| \\
&\le \|W_k'\| \|g_{k-1}(\theta') - g_{k-1}(\theta)\| + \|W_k' - W_k\| \|g_{k-1}(\theta)\| \\
&\le \bar{n} \cdot L_{k-1} \|\theta' - \theta\| + \|\theta' - \theta\| \bar{g}_{k-1}.
\end{aligned}$$

Putting these results together yields

$$\|g_k(\theta') - g_k(\theta)\| \le M_k(\bar{n}L_{k-1} + \bar{g}_{k-1} + 1) \|\theta' - \theta\|.$$

Thus, $g_k$ is locally Lipschitz in $\theta$ on the neighborhood $\mathcal{N}(\theta)$ with Lipschitz constant $L_k = M_k(\bar{n}L_{k-1} + \bar{g}_{k-1} + 1)$.

Hence, by induction, $g_K$ is locally Lipschitz in $\theta$. Finally, by Rademacher's Theorem [15, Corollary 4.12], $g_K$ is differentiable a.e. $\square$

**Lemma 7.** *Let $W \subseteq \mathbb{R}^n$ be an open set, and let $h : W \to \mathbb{R}$ be a continuous function. If $g : W \times \mathbb{R} \to \mathbb{R}$ is defined by $g(w, b) := h(w) + b$, then for every $c \in \mathbb{R}$, the $c$-level set of $g$, $L_c := \{(w, b) \in W \times \mathbb{R} \mid g(w, b) = c\}$, has measure zero.*

*Proof.* First, we establish measurability of $L_c$. $g$ is measurable because it is continuous, and $L_c = g^{-1}(\{c\})$ is the preimage of the Borel measurable singleton set $\{c\}$. Therefore, $L_c$ is a measurable set, and its indicator function $\mathbf{1}_{L_c}$ is a measurable function.

For any $m \in \mathbb{N}$, let $\lambda_m$ denote the Lebesgue measure on $\mathbb{R}^m$. Then,

$$
\begin{aligned}
\lambda_{n+1}(L_c) &= \int_{\mathbb{R}^{n+1}} \mathbf{1}_{L_c}(w, b) \, \mathrm{d}\lambda_{n+1}(w, b) \\
&= \int_{\mathbb{R}^n} \int_{\mathbb{R}} \mathbf{1}_{L_c}(w, b) \, \mathrm{d}\lambda_1(b) \, \mathrm{d}\lambda_n(w) && \text{Tonelli's theorem} \\
&= \int_{\mathbb{R}^n} \lambda_1(\{b \in \mathbb{R} \mid g(w, b) = c\}) \, \mathrm{d}\lambda_n \\
&= \int_{\mathbb{R}^n} \lambda_1(\{c - h(w)\}) \, \mathrm{d}\lambda_n && \{b \mid g(w, b) = c\} = \{c - h(w)\} \\
&= \int_{\mathbb{R}^n} 0 \, \mathrm{d}\lambda_n = 0 && \text{singleton sets have 0 measure}
\end{aligned}
$$

$\square$

## C.5 Convexity of $\tilde{h}$

In this section, for ease of exposition, we write $\tilde{h}(\lambda, t)$ instead of $\tilde{h}_t(\lambda)$. The following proposition is used to show that (12) is convex.

**Proposition 3.** *Under the assumptions of Theorem 1 or Theorem 2, $\tilde{h}(\lambda, t)$ is nondecreasing in $\lambda$ and convex in $t$. Furthermore, if $\{L_i\}_{i=1}^N$ and $B$ are convex, then $\tilde{h}$ is jointly convex in $(\lambda, t)$.*

*Proof.* First, we show that $\tilde{h}(\lambda, t)$ is nondecreasing in $\lambda$.

- In the setting of Theorem 1 (i.e., Assumptions 1 and 2), the functions $\{B, L_1, \ldots, L_N\}$ are all nondecreasing. Since $\phi$ is also nondecreasing, $\tilde{L}_{i,t}$ and $\tilde{B}_t$ are also nondecreasing. Thus, $\tilde{h}$ is nondecreasing in $\lambda$.
- In the setting of Theorem 2 (i.e., Assumptions 1 and 5 with CVaR risk), we showed in Appendix C.3 that $\tilde{L}_{i,t}$ and $\tilde{B}_t$ are nondecreasing. Thus, $\tilde{h}$ is nondecreasing in $\lambda$.

Second, we show that $\tilde{h}(\lambda, t)$ is convex in $t$. We start by showing that $\tilde{B}_t(\lambda)$ is convex in $t$. For any $a \in [0, 1]$,

$$
\begin{aligned}
\tilde{B}_{at+(1-a)t}(\lambda) &= at + (1-a)t + \phi(B(\lambda) - at + (1-a)t) \\
&= at + (1-a)t + \phi(a(B(\lambda) - t) + (1-a)(B(\lambda) - t)) \\
&\leq at + (1-a)t + a\phi(B(\lambda) - t) + (1-a)\phi(B(\lambda) - t) && \phi \text{ is convex} \\
&= a\tilde{B}_t(\lambda) + (1-a)\tilde{B}_t(\lambda).
\end{aligned}
$$

An identical argument shows that $\tilde{L}_{i,t}$ is convex in $t$. Since $\tilde{h}$ is the sum of functions that are convex in $t$, $\tilde{h}$ is convex in $t$.

If $\{L_i\}_{i=1}^N$ and $B$ are convex (in $\lambda$), then by Lemma 3, $\{\tilde{L}_{i,t}\}_{i=1}^N$ and $\tilde{B}_t$ are all convex in $(\lambda, t)$. Since $\tilde{h}$ is their mean, it is also convex in $(\lambda, t)$. $\square$

## D Experimental details

**Computational resources** Our experiments were performed on two computers with the following hardware:

1. $2\times$ AMD EPYC 7513 32-Core CPUs, $4\times$ NVIDIA A100 GPUs (80GiB GPU memory each), 1 TiB RAM
2. Intel Core i9-12900KS CPU, $2\times$ NVIDIA RTX A6000 GPUs (48GiB GPU memory each), 125 GiB RAM

Our code (see supplementary ZIP file) is written in Python and primarily relies on the PyTorch [40] deep learning library. We use the Adam optimizer [30].

## D.1 Tumor image segmentation

**Datasets** We used images from 4 public datasets (CVC-ClinicDB [13], CVC-ColonDB [12], ETIS-LaribPolypDB [46], Kvasir-SEG [28]), yielding a total of 2,188 images with per-pixel binary mask labels. Following a similar setup to [23], we used 1,450 images from the CVC-ClinicDB and Kvasir-SEG datasets to form a training set; we used the same training split as [23]. From the remaining 738 images, we created 10 different val (400 images) / test (338 images) splits from different random seeds.

Note that unlike [23], we did include the "CVC-300" dataset in our test set, as "CVC-300" is a duplicate of CVC-ColonDB.

**Model fine-tuning** We use a pretrained PraNet [23] model, which features a Res2Net [24] backbone and a U-Net-like [43] decoder. During model fine-tuning, we only update the weights of the decoder; we keep the Res2Net backbone weights frozen. We tune the learning rate with a grid search over $(10^{-2}, 10^{-3}, 10^{-4}, 10^{-5}, 10^{-6})$. Models are trained with a batch size of 400 for up to 100 epochs with early-stopping after 10 epochs of no improvement on the val set.

We compare two different forms of fine-tuning, as shown in Figure 1.

- "cross-entropy" fine-tuning refers to using the same multi-scale binary cross-entropy loss that [23] used to pretrain the PraNet model.

- "conformal risk training" refers to using the gradient (9). Recall that Example 2 derives the exact expression $\frac{d\lambda(\theta)}{d\theta} = \frac{d}{d\theta} f_\theta(X_i)_j$ where $j$ is the index of some pixel in some image $i$ of the training minibatch. This gradient may have high variance because it is the gradient through the model's output on exactly a single pixel among an entire minibatch of images. In practice, we reduce the variance of the gradient estimator by averaging the model's gradient over other pixels with similar model outputs:

$$\frac{d\lambda(\theta)}{d\theta} \approx \frac{1}{M} \sum_{(i,j)\in I} \frac{d}{d\theta} f_\theta(X_i)_j$$

  for the $M$ indices $I = \{(i_k, j_k)\}_{k=1}^M$ whose values $f_\theta(X_{i_k})_{j_k}$ are closest to $\lambda(\theta)$. (We set $M$ dynamically to be 0.5% of the number of positive pixels in each minibatch.) This approach is inspired by the variance reduction techniques introduced by [26, 39], which show reduced gradient estimation variance at the expense of possibly incurring some estimation bias. We leave it to future work to formally analyze bias-variance trade-off of this gradient estimator.

**Computational efficiency** Our conformal risk training procedure imposes negligible computational overhead compared to fine-tuning using the standard binary cross-entropy loss. For each minibatch of $M$ images with $d$ pixels each, the only overhead is incurred by the binary search procedure from Algorithm 1. In practice, we implement this by sorting the model outputs $\{f_\theta(X_i)_j\}_{i\in[M], j\in[d]}$, which requires $O(Md \log Md)$ time complexity and took on average <50 milliseconds of wall-clock time per minibatch.

## D.2 Battery storage operations

**Dataset** We use the same dataset as Donti et al. [19] in our battery storage problem. In this dataset, the target $y \in \mathbb{R}^{24}$ is the hourly PJM day-ahead system energy price for 2011-2016, for a total of 2189 days. However, whereas Donti et al. [19] excluded any days whose electricity prices are too high (>$500/MWh), our conformal risk training method is specifically designed to handle tail-risk, so we did not exclude these "outliers." For predicting target for a given day, the inputs $x \in \mathbb{R}^{77}$ include the previous day's log-prices, the given day's hourly load forecast, the previous day's hourly temperature, and several calendar-based features such as whether the given day is a weekend or a US holiday.

Unlike Donti et al. [19], we did not include the given day's hourly temperature as input features, as such features are unavailable in practice (although temperature *forecasts* may be available). To make the problem instance more challenging, we also added i.i.d. $\mathcal{N}(0, \sqrt{20})$ noise to the price targets.

We take a random 20% subset of the dataset as the test set; because the test set is selected randomly, it is considered exchangeable with the rest of the dataset. For each of 10 seeds, we further use a 65%/35% random split of the remaining data for training and calibration.

**Decision constraints**   We note that we use a different, but equivalent, parameterization of the battery state-of-charge constraints. Donti et al. [19] use the task loss

$$f(y, z) = \sum_{t=1}^{T} y_t (z^{\text{in}} - z^{\text{out}})_t + \epsilon^{\text{flex}} \left\| z^{\text{state}} - \frac{C}{2} \mathbf{1} \right\|^2 + \epsilon^{\text{ramp}} \left( \left\| z^{\text{in}} \right\|^2 + \left\| z^{\text{out}} \right\|^2 \right)$$

with constraints, for all $t = 1, \ldots, T$:

$$z_0^{\text{state}} = C/2, \quad z_t^{\text{state}} = z_{t-1}^{\text{state}} - z_t^{\text{out}} + \gamma z_t^{\text{in}},$$
$$0 \le z^{\text{in}} \le c^{\text{in}}, \quad 0 \le z^{\text{out}} \le c^{\text{out}}, \quad 0 \le z_t^{\text{state}} \le C.$$

Instead of using a $z^{\text{state}}$ variable to represent state of charge, we equivalently express the task loss and constraints in terms of a variable $z^{\text{net}} \in \mathbb{R}^T$ representing the difference of the state of charge from the starting point. Letting $z = (z^{\text{in}}, z^{\text{out}}, z^{\text{net}})$ denote the decision vector, we have

$$f(y, z) = y^\top (z^{\text{in}} - z^{\text{out}}) + \epsilon^{\text{flex}} \left\| z^{\text{net}} \right\|^2 + \epsilon^{\text{ramp}} \left( \left\| z^{\text{in}} \right\|^2 + \left\| z^{\text{out}} \right\|^2 \right)$$

with constraint set

$$\mathcal{Z} = \left\{ (z^{\text{in}}, z^{\text{out}}, z^{\text{net}}) \in \mathbb{R}^{3T} \,\middle|\, \begin{array}{l} \forall t \in [T]: \; z_t^{\text{net}} := \sum_{\tau=1}^{t} \gamma z_\tau^{\text{in}} - z_\tau^{\text{out}}, \\[6pt] 0 \le z^{\text{in}} \le c^{\text{in}}, \quad 0 \le z^{\text{out}} \le c^{\text{out}}, \quad -C/2 \le z^{\text{net}} \le C/2 \end{array} \right\}.$$

This reparameterization is necessary for the decision variable $z$ to satisfy our "decision-scaling" requirement: if $z \in \mathcal{Z}$, then $\forall \lambda \in [0, 1]$, $\lambda z \in \mathcal{Z}$.

**Model and fine-tuning**   Our price forecasting model $\hat{y}_\theta$ is a fully-connected multilayer perceptron (MLP) with 3 hidden layers, each with 256 units, LeakyReLU activation, and batch normalization [27] layers. We use a batch size of 400.

We pretrain the MLP to predict prices using a mean squared error (MSE) loss. During pretraining, we tune the learning rate across $(10^{-4}, 10^{-3.5}, 10^{-3}, 10^{-2.5}, 10^{-2}, 10^{-1.5})$, and we use L2 weight deecay strength of $10^{-4}$. We pretrain for up to 500 epochs with early-stopping after 20 epochs of no improvement on the val set.

During fine-tuning, we tune the learning rate with a grid search over $(10^{-2}, 10^{-3}, 10^{-4}, 10^{-5})$ and we keep the $10^{-4}$ L2 weight decay. We fine-tune for up to 100 epochs with early-stopping after 10 epochs of no improvement on the val set. During fine-tuning, we use a 90%/10% weighted combination of the fine-tuning loss and the MSE loss.

We compare two different forms of fine-tuning, as shown in Figure 2.

- "task loss" refers to fine-tuning the model $\hat{y}_\theta$ using the decision-focused learning task loss from [19]. As explained in Section 5.2, we define the decision for an input $x$ as

$$\hat{z}_\theta(x) := \arg\min_{z \in \mathcal{Z}} f(\hat{y}_\theta(x), z),$$

  and the task loss incurred is $f(y, \hat{z}_\theta(x))$. Thus, "task loss" fine-tuning refers to using $f(y, \hat{z}_\theta(x))$ as the training loss function.

- "conformal risk training" refers to using the gradient (9). In this battery storage problem, we seek to control the CVaR tail risk of the financial term $L(\theta, \lambda) = \lambda(\hat{z}_\theta^{\text{in}}(X) - \hat{z}_\theta^{\text{out}}(X))^\top Y$ while minimizing the task loss $\ell(\theta, \lambda) = f(Y, \lambda \hat{z}_\theta(X))$. We seek to solve

$$\min_{\theta \in \Theta, \lambda \in \Lambda} \mathbb{E}_{(X,Y) \sim \mathcal{P}} [f(Y, \lambda \hat{z}_\theta(X))] \quad \text{s.t.} \; \text{CVaR}_{(X,Y) \sim \mathcal{P}}^\delta [\lambda(\hat{z}_\theta^{\text{in}}(X) - \hat{z}_\theta^{\text{out}}(X))^\top Y] \le \alpha.$$

  During each minibatch of training, we pick the $t$ that allows for the largest $\lambda$. In other words, we pick $\lambda(\theta)$ by solving the convex optimization problem

$$\lambda(\theta) = \arg\max_{\lambda \in \Lambda} \max_{t \in \mathbb{R}} \lambda \quad \text{s.t.} \; \tilde{h}_t(\theta, \lambda) \le \alpha, \; B(\lambda_{\min}) \le t \le \alpha. \tag{12}$$

  This problem is convex because $\tilde{h}_t$ is jointly convex in $(\lambda, t)$ by Proposition 3. The gradient term $\frac{d\lambda(\theta)}{d\theta}$ is then given by Theorem 3(ii) with the $\phi_{\text{CVaR}^\delta}$ disutility function.

**Computational efficiency** Our conformal risk training procedure imposes negligible computational overhead compared to fine-tuning using the task loss. For each minibatch, the main overhead incurred is from solving and differentiating through (12), which we found in practice to require on average <50 milliseconds of wall-clock time per minibatch using `cvxpylayers` [1].

# E   Conformal risk training subsumes conformal training

In this section, we illustrate how our method, conformal risk training, includes conformal training (ConfTr) from Stutz et al. [47] as a special case.

Consider the multi-class classification problem with classes $\mathcal{Y} = \{1, \ldots, K\}$. For a nonconformity score function $s_\theta : \mathcal{X} \times \mathcal{Y} \to \mathbb{R}$ parameterized by $\theta$, conformal prediction [49, 45, 4] forms a prediction set

$$C_\theta(X; \lambda) = \{k \in \mathcal{Y} \mid s_\theta(X, k) \leq 1 - \lambda\}.$$

The overall goal posed by Stutz et al. [47] is to optimize inefficiency (*i.e.*, minimize the average size of the conformal prediction set) subject to a minimum coverage rate. The minimum coverage constraint can be expressed using the loss function $L(\theta, \lambda) := \mathbf{1}[s_\theta(X, Y) > 1 - \lambda]$, which is left-continuous and nondecreasing in $\lambda$. Since $\mathbb{E}[L(\theta, \lambda)] = \Pr(s_\theta(X, Y) > 1 - \lambda) = \Pr(Y \notin C_\theta(X; \lambda))$, the constraint $\mathbb{E}[L(\theta, \lambda)] \leq \alpha$ is equivalent to $\Pr(Y \in C_\theta(X; \lambda)) \geq 1 - \alpha$.

To optimize inefficiency, Stutz et al. [47] proposes using cost function that is the sum of a soft assignment of classes:

$$\ell(\theta, \lambda) := \max\left(0, \sum_{k=1}^{K} \sigma\left(\frac{(1 - \lambda) - s_\theta(X, k)}{T}\right) - 1\right),$$

where $T$ is a temperature hyperparameter. Thus, this problem is an instance of the end-to-end optimal risk control problem

$$\min_{\theta \in \Theta, \lambda \in \Lambda} \mathbb{E}[\ell(\theta, \lambda)] \quad \text{s.t.} \ \mathbb{E}[L(\theta, \lambda)] \leq \alpha.$$

Given a dataset $D = \{(X_i, Y_i)\}_{i=1}^{N}$ drawn exchangeably from $\mathcal{P}$, let $L_i(\theta, \lambda) := \mathbf{1}[s_\theta(X_i, Y_i) > 1 - \lambda]$. Clearly, every $L_i$ is bounded by $B = 1$. Following the conformal risk control procedure, and noting that smaller values of $\lambda$ yield lower objective values, we may pick

$$
\begin{aligned}
\lambda(\theta) &:= \sup\left\{\lambda \in \mathbb{R} \ \middle| \ \frac{1}{N+1}\left(1 + \sum_{i=1}^{N}\mathbf{1}[s_\theta(X_i, Y_i) > 1 - \lambda]\right) \leq \alpha\right\} \\
&= \sup\left\{\lambda \in \mathbb{R} \ \middle| \ \frac{1}{N+1}\left(1 + \sum_{i=1}^{N}(1 - \mathbf{1}[s_\theta(X_i, Y_i) \leq 1 - \lambda])\right) \leq \alpha\right\} \\
&= \sup\left\{\lambda \in \mathbb{R} \ \middle| \ 1 - \frac{1}{N+1}\sum_{i=1}^{N}\mathbf{1}[s_\theta(X_i, Y_i) \leq 1 - \lambda] \leq \alpha\right\} \\
&= \sup\left\{\lambda \in \mathbb{R} \ \middle| \ \sum_{i=1}^{N}\mathbf{1}[s_\theta(X_i, Y_i) \leq 1 - \lambda] \geq (N+1)(1-\alpha)\right\} \\
&= 1 - s_{\theta, (\lceil (N+1)(1-\alpha) \rceil)}
\end{aligned}
\tag{13}
$$

where $s_{\theta, (j)}$ denotes the $j$-th smallest value of the set $S := \{s_\theta(X_i, Y_i)\}_{i=1}^{N} \cup \{\infty\}$. By Proposition 1, $\lambda(\theta)$ satisfies $\Pr(Y \in C_\theta(X; \lambda(\theta))) \geq 1 - \alpha$.

We note that whereas Stutz et al. [47] approximates $\frac{\mathrm{d}\lambda(\theta)}{\mathrm{d}\theta}$ using differentiable approximate ranking and sorting operations, the gradient can be computed exactly [26, 51, 39]. That is, if $s_\theta(X_i, Y_i)$ is differentiable w.r.t. $\theta$ for all $i = 1, \ldots, N$ and if $s_{\theta, (\lceil (N+1)(1-\alpha) \rceil)}$ is unique amongst the calibration scores, then by [51, Theorem 2]

$$\frac{\mathrm{d}\lambda(\theta)}{\mathrm{d}\theta} = \begin{cases} -\frac{\mathrm{d}}{\mathrm{d}\theta}s_\theta(X_{\sigma(k)}, Y_{\sigma(k)}), & \text{if } \alpha \geq \frac{1}{N+1} \\ 0, & \text{otherwise} \end{cases}$$

where $k = \lceil (N+1)(1-\alpha) \rceil$ and $\sigma : [N] \to [N]$ denotes the permutation that sorts the scores in ascending order, such that $s_\theta(X_{\sigma(i)}, Y_{\sigma(i)}) \leq s_\theta(X_{\sigma(j)}, Y_{\sigma(j)})$ for all $i < j$.

