# OpenReview forum: "Conformal Risk Training: End-to-End Optimization of Conformal Risk Control"
_NeurIPS.cc/2025/Conference — NeurIPS 2025 poster_

### Official Review · Reviewer_A5AC · 2025-07-02

**Clarity:** 4
**Significance:** 3
**Originality:** 3
**Rating:** 5
**Confidence:** 3

**Summary:**

This paper addresses two key limitations of standard Conformal Risk Control (CRC) methods [Angelopoulos et al.]- 1) while CRC provides finite-sample guarantees on expected loss, many real-world high-stakes applications require control over tail risks (such as Conditional Value-at-Risk (CVaR)), 2) CRC is typically applied post hoc to a pretrained model, which may degrade performance due to the lack of feedback between risk calibration and model learning.

To address the first limitation, the authors introduce a new post-hoc conformal risk control framework, called Conformal OCE Risk Control (CORC). This framework extends the original CRC method to a broader family of risk measures known as Optimized Certainty Equivalent (OCE) risks. These include not only the standard expected loss (recovering CRC) but also more tail-sensitive metrics like CVaR.

To address the second limitation, the authors propose a new training strategy called conformal risk training, which incorporates feedback from CORC into the model’s training process. This approach jointly optimizes both the model parameters and the risk parameter by minimizing performance loss while ensuring the risk stays within the target upper bound. It is conceptually similar to the conformal training method introduced by David (cite), but extends it to the conformal risk control setting.

The proposed conformalized risk training method is evaluated on two high-stakes tasks—tumor image segmentation and battery energy storage optimization. The results demonstrate that the method provides valid risk control while also improving efficiency and utility.

[Angelopoulos et al.] A. N. Angelopoulos, S. Bates, A. Fisch, L. Lei, and T. Schuster. Conformal Risk Control, Apr. 2023

**Questions:**

1. In your experiments involving conformal risk training, did you apply an additional layer of CRC or CORC after model fine-tuning? Or were the reported results directly from the trained model?
2. Have you derived a lower bound on the test-time risk, similar to the approach taken in the original CRC paper?
3. Do you have any results comparing CRC and CORC purely as post-hoc methods (i.e., using the same pretrained models without conformal risk training)?

**Ethical Concerns:**

["NO or VERY MINOR ethics concerns only"]

**Final Justification:**

This paper addresses an important methodological gap that is valuable to the conformal prediction community. The authors provide both theoretical justifications and empirical evidence demonstrating the effectiveness of their proposed method. In addition, their rebuttal offered clear and informative responses that satisfactorily resolved my concerns. Based on these considerations, I recommend acceptance of this paper.

**Limitations:**

Yes.

**Quality:**

3

**Strengths And Weaknesses:**

Strengths:
1. Clarity and presentation: This paper is very well-written and well-structured. The authors clearly present their problem statement and articulate their motivations effectively. They also provide an easy-to-follow description of the methodology along with solid theoretical justification. I particularly appreciate the inclusion of intuitive examples that help clarify the technical details throughout the paper.
2. Novelty and Significance: This paper addresses an important challenge in high-stakes applications—controlling tail risks—and fills a notable gap in the conformal prediction literature. It extends the existing conformal risk control framework in a principled manner to accommodate a broad class of Optimized Certainty Equivalent (OCE) risk functions. Additionally, the paper examines specific instances within the OCE family, such as Conditional Value-at-Risk (CVaR), and demonstrates that the standard non-decreasing assumption can be relaxed to monotonicity in this case. To the best of my knowledge, this is also the first work to incorporate conformal risk control into the conformal training paradigm.
3. Soundness: The authors provide rigorous theoretical justification for the validity of their method and demonstrate improved efficiency through numerical experiments.

Weaknesses:
1. Please correct me if I have overlooked anything, but the empirical evaluation appears limited. The method is tested on only two datasets, each evaluated under a limited set of configurations (e.g., varying the target risk level $\alpha$ or tail risk level $\delta$. The work would be further strengthened by evaluating the proposed method on a more diverse set of classification tasks. Additionally, it may be helpful to explore the impact of varying calibration set sizes, as the proposed approach requires an additional data split to ensure validity. This introduces a trade-off between reduced data efficiency and the benefits of the proposed method. Investigating the performance in low-calibration regimes could provide valuable insights into the method's robustness and practical applicability. It would be particularly interesting to explore the potential benefits of the method when the calibration sample size is small.

---

> ### Author Rebuttal · Authors · 2025-07-31
>
> We appreciate the reviewer for recognizing the novelty and significance of our Conformal Risk Training method for learning to control the broad class of optimized certainty-equivalent risks. We respond to each point raised by the reviewer.
>
> **Empirical evaluation**
>
> We have taken note of the reviewer's desire to see evaluation on more datasets. While we did not have time to run more experiments during this 1-week rebuttal period, we are actively looking into other datasets to test our method.
>
> The reviewer also suggested evaluating the effect of varying the calibration set size on tail risk control. Below are 2 tables that show the average task loss and the CVaR on the battery storage problem as the calibration set size changes for post-hoc conformal CVaR Control. Values are given as mean ± std over 10 random seeds for choosing the calibration set.
>
> As the tables show, a larger calibration set is generally better, both in terms of reducing the task loss and achieving a tighter CVaR bound closer to $\alpha$ (_i.e._, not being overly conservative). The reason a larger calibration set size $N$ helps is because the term $\frac{1}{N+1} \tilde{B}_t(\lambda)$ appears in $\tilde{h}_t(\lambda)$. Increasing $N$ reduces the effect of the upper bound $B$ and enables choosing larger (less conservative) $\hat\lambda$. However, there are diminishing gains as $N$ increases, which we clearly observe when $\delta = 0.9$.
>
>
> Table 1: Average task loss
>
> | calib size | α=2, δ=.9 | α=2, δ=.95 | α=2, δ=.99 | α=5, δ=.9 | α=5, δ=.95 | α=5, δ=.99 | α=10, δ=.9 | α=10, δ=.95 | α=10, δ=.99 |
> |-|-|-|-|-|-|-|-|-|-|
> | 100 | 0 ± 0 | -0.64 ± 1.3 | -0.8 ± 0.073 | 0 ± 0 | -1.6 ± 3.4 | -2 ± 0.18 | -7.7 ± 16 | -3.2 ± 6.7 | -4 ± 0.37 |
> | 300 | 0 ± 0 | -1.1 ± 2.3 | -1.4 ± 0.26 | -4.2 ± 13 | -2.6 ± 5.7 | -3.4 ± 0.66 | -13 ± 20 | -7.2 ± 12 | -6.8 ± 1.3 |
> | 613 (default) | 0 ± 0 | -1.5 ± 2.7 | -1.8 ± 0.51 | -4.2 ± 13 | -3.6 ± 6.7 | -4.5 ± 1.3 | -13 ± 20 | -7.3 ± 13 | -9 ± 2.6 |
>
> Table 2: CVaR
>
> | calib size | α=2, δ=.9 | α=2, δ=.95 | α=2, δ=.99 | α=5, δ=.9 | α=5, δ=.95 | α=5, δ=.99 | α=10, δ=.9 | α=10, δ=.95 | α=10, δ=.99 |
> |-|-|-|-|-|-|-|-|-|-|
> | 100 | 0 ± 0 | 0.19 ± 0.41 | 0.61 ± 0.11 | 0 ± 0 | 0.48 ± 1 | 1.5 ± 0.28 | 0.96 ± 2.1 | 0.97 ± 2.1 | 3 ± 0.57 |
> | 300 | 0 ± 0 | 0.31 ± 0.66 | 1 ± 0.2 | 0.39 ± 1.2 | 0.79 ± 1.7 | 2.6 ± 0.49 | 1.5 ± 2.5 | 2.1 ± 3.5 | 5.1 ± 0.98 |
> | 613 (default) | 0 ± 0 | 0.46 ± 0.81 | 1.3 ± 0.24 | 0.39 ± 1.2 | 1.1 ± 2 | 3.3 ± 0.59 | 1.5 ± 2.5 | 2.3 ± 4.1 | 6.7 ± 1.2 |
>
>
> **Applying CRC / CORC after fine-tuning**
>
> In all of our results, we always apply CRC or CORC after model training with a separate held-out validation set in order to formally guarantee risk control.
>
> **Lower bound on test-time risk**
>
> We have not derived a lower bound on the test-time risk. If we apply the same derivation from the original CRC paper to OCE risks, and we assume that $B$ is constant, we end up with
>
> $$
> \mathbb{E}[\tilde{L}\_{N+1,t}(\hat\lambda)]
> \geq \mathbb{E}\left[ \frac{1}{N+1} \sum\_{i=1}^{N+1} \tilde{L}\_{i,t}(\hat\lambda'') \right]
> \geq \alpha - \frac{2 \tilde{B}\_t}{N+1}
> $$
>
> where
>
> $$
> \hat\lambda''
> := \inf\left\\{ \lambda \in \Lambda \ \middle|\ \frac{1}{N+1} \left( \tilde{B}\_t + \sum\_{i=1}^{N+1} \tilde{L}\_{i,t}(\hat\lambda) \right) \leq \alpha \right\\}.
> $$
>
> However, we have $R[L\_{N+1}(\lambda)] \leq \mathbb{E}[\tilde{L}\_{N+1,t}(\lambda)]$, and the inequality is generally not tight unless $t$ is chosen to be the minimizer of the variational form of the OCE risk. Therefore, we cannot directly compare $R[L\_{N+1}(\hat\lambda)]$ with $\alpha - \frac{2 \tilde{B}\_t}{N+1}$.
>
>
> **Comparing CRC and CORC as purely post-hoc methods**
>
> We are not 100% sure what the reviewer means with their question, so we will try to answer the question in 2 different ways.
>
> If the reviewer is asking about comparing post-hoc CRC against post-hoc CORC, we would like to point out that if the risk measure is the expected loss, then CRC and CORC are exactly equivalent. On the other hand, if the risk measure is not the expected loss, then CRC is undefined. Therefore, it does not make sense to compare post-hoc CRC against post-hoc CORC in this setting.
>
> If the reviewer is instead asking to see comparisons of post-hoc CRC vs. Conformal Risk Training (with expected loss), or comparisons of post-hoc CORC vs. Conformal Risk Training (with CVaR risk), we note that this comparison is the focus of our experimental section, and we apologize if our experimental section did not make this clear---we will clarify this accordingly in the camera-ready paper.

---

> > ### Comment · Reviewer_A5AC · 2025-08-05
> >
> > I thank the authors for their detailed responses and for providing additional experimental results. The clarifications and new evidence sufficiently address my questions. In light of this, I would like to raise my score and recommend the paper for acceptance.

---

> > > ### Author Response · Authors · 2025-08-05
> > > **Thank you**
> > >
> > > Thank you for the thoughtful review and the positive reassessment. We are glad our responses addressed your concerns, and we will incorporate the clarifications into the revised manuscript.

---

> ### Comment · Area_Chair_JXu6 · 2025-08-04
>
> Dear reviewer, please engage with the author rebuttal.

---

### Official Review · Reviewer_mggh · 2025-07-02

**Clarity:** 2
**Significance:** 3
**Originality:** 4
**Rating:** 4
**Confidence:** 4

**Summary:**

The paper has two key contributions:
(i) a generalization of Conformal Risk Control for general Optimized Certainty Equivalent (OCE) risks, which enables the control of risks beyond average risks: e.g., CVaR and Entropic risks. The paper pays particular attention to the CVaR case.
(ii) A conformal training scheme, which allows for the propagation of gradients through the calibration process of the proposed conformal OCE risk control procedure (which subsumes the standard Conformal Risk Control / Vanilla Conformal procedures). This is formally derived, a distinct feature in comparison to previous works.
Both schemes seem to have favorable empirical results.

**Questions:**

I'd appreciate it if the authors could clarify my questions in the 'Strengths&Weaknesses' section regarding the proof of Theorem 3.

**Ethical Concerns:**

["NO or VERY MINOR ethics concerns only"]

**Final Justification:**

The authors reasonably answered my concerns. I was unfortunately unable to carefully review the new version of the theorem, as it is very lengthy and technical, but their comments beforehand inspire some confidence. I generally lean towards acceptance: the paper is on a relevant topic, and I think their contributions -- both on controlling OCE risks as well as deriving gradients for conformal training -- are of significant interest to the conformal prediction community. I do, however, keep my score to a borderline accept due to (i) the new theorem, which I was unable to fully check, and (ii) the title, which I believe to have a slight mismatch with the paper.

**Limitations:**

Limitations were appropriately discussed.

**Paper Formatting Concerns:**

The figures are rather small and hard to read. In a camera-ready version, it would be a good idea to use the extra space to increase their size.

**Quality:**

3

**Strengths And Weaknesses:**

- The generalization of Conformal Risk Control for Optimized Certainty Equivalent risks is quite nice, and the ability to work with more general risks such as CVaR is highly desirable. The only cost seems to be (besides boundedness assumptions) the introduction of the $t$ hyperparameter, which the authors propose to be selected using a separate data split. How sensitive is the choice of $t$? Is there some guidance for how to select it a priori?

- On Theorem 3 (regarding the conformal training scheme):
    - I strongly advise the authors to write the closed-form gradients in the proposition of the theorem, at the very least in the supplementary material, rather than hiding them in the proofs.
    - This is, as far as I am aware, the most formal treatment of conformal training to date, for which I applaud the authors. I just have some (hopefully) minor comments:
        - On the proof of case (i):
            - The writing of the proof is a bit tough to follow, in particular due to simultaneous notational overload and notational abuse. Perhaps it can be more clarified?
            - Line 960: it is claimed that because $(\varphi, c\_j, g\_j)$ are differentiable almost everywhere, so is $\tilde{c}_j$. But as far as I can tell this involves a use of the chain rule, which need not apply for almost-everywhere differentiability. Did I miss something? Or maybe $\varphi$ should be assumed to be differentiable everywhere? (Also the role of the $g_j$ seems delicate here, and I'm not entirely certain that differentiability is indeed preserved.)
        - On the proof of case (ii):
            - Lines 986-988 claim that we can replace $\ell_i$ with $\lambda$ (or $-\lambda$), but I'm not sure this is correct; could the authors clarify why this is so?
            - The idea of the proof seems to then be to reduce the problem to a standard bilevel optimization (via a sort of duality argument). Is my interpretation correct?
    - The theorem establishes that the gradient exists almost everywhere, and somewhat states how to compute them in the proof. However, is it guaranteed that the gradients, where defined, are at least non-zero? For example, a combinatorial loss (e.g., AUC, Accuracy) is also differentiable almost everywhere, but has nil gradients where it is defined (and thus its gradients are useless). I *think* that generally the gradients in the proposed approach will generally be useful, but that there may be some situations under which the gradients become nil.

- Finally, I am somewhat bothered by the title of the paper. As mentioned, the paper has two key contributions, these being (i) the new calibration procedure for OCE risks, and (ii) the conformal training approach.
Contribution (i) seems quite solid, and I think would have even stood as a paper on its own.
Contribution (ii) is also nice, but essentially spans a single theoretical result, and is reliant on the formulation of (i) (though it does seem like it could be restated for the special case of the more usual risks).
Overall, the paper's content reads as the primary contribution being (i), with (ii) being a nice consequence.
However, the title refers solely to (ii), making no mention at all to (i) or Optimized Certainty Equivalents; one would think that it is a mere adaptation of conformal training onto conformal risk control (of average risks). It's more than that!
I realise this is a bit nitpicky on my part, but it feels like a genuine mismatch between the title and content.

---

> ### Author Rebuttal · Authors · 2025-07-31
>
> We thank the reviewer for recognizing our Conformal OCE Risk Control method as a desirable generalization of Conformal Risk Control to work with more general risks such as CVaR. We also thank the reviewer for recognizing our formal treatment of conformal training. We respond to each of the points raised by the reviewer.
>
> **Sensitivity to $t$ hyperparameter**
>
> As the reviewer points out, Conformal OCE Risk Control is dependent on picking a good hyperparameter $t$ to tighten the OCE risk bound. From a practical / implementation perspective, we recommend selecting $t$ using a separate data split.
>
> From a more theoretical perspective, we can see that $t$ shows up in two inequalities in the Proof of Theorem 1 (Appendix C.2):
>
> $$
> R[L_{N+1}(\hat\lambda)]
> \leq \mathbb{E}\left[ t + \phi(L_{N+1}(\hat\lambda) - t) \right]
> = \mathbb{E}[\tilde{L}_{N+1,t}(\hat\lambda)]
> \leq \alpha.
> $$
>
> The first inequality is tight at the value $t$ that minimizes the variational OCE expression. For example, when $R = \text{CVaR}^\delta$, $t=\text{VaR}^\delta[L(\hat\lambda)]$ makes the first inequality tight. On the other hand, theoretically analyzing the best choice of $t$ to tighten the 2nd inequality is difficult in general. We welcome suggestions, but for now, we leave this to future work.
>
> In the 2 tables below, we show the empirical sensitivity of the average task loss and the empirical CVaR on the choice of $t$ for post-hoc Conformal CVaR Control in the battery storage task. Here, $\Delta t$ is the (absolute) amount by which we change $t$ from the value tuned on the training set. Evidently, the value of $t$ tuned on the training set is not always optimal, but the resulting task loss tends to be close (within 1 stddev) of the task loss from the optimal $t$.
>
> Due to space limitations, we can only show a few choices of $\Delta t$, but we will include a proper figure in the camera-ready.
>
> _Table 1_: Task loss
> | Δt | α=2, δ=0.9 | α=2, δ=0.95 | α=2, δ=0.99 | α=5, δ=0.9 | α=5, δ=0.95 | α=5, δ=0.99 | α=10, δ=0.9 | α=10, δ=0.95 | α=10, δ=0.99 |
> |-|-|-|-|-|-|-|-|-|-|
> | -5 | 0 ± 0 | 0 ± 0 | 0 ± 0 | 0 ± 0 | 0 ± 0 | 0 ± 0 | -8.4 ± 18 | 0 ± 0 | -2 ± 3.2 |
> | -1 | 0 ± 0 | 0 ± 0 | -0.39 ± 0.64 | 0 ± 0 | -2.9 ± 6.1 | -3.7 ± 1.7 | -8.4 ± 18 | -6.2 ± 13 | -8.4 ± 3 |
> | 0 | 0 ± 0 | -1.5 ± 2.7 | -1.8 ± 0.51 | -4.2 ± 13 | -3.6 ± 6.7 | -4.5 ± 1.3 | -13 ± 20 | -7.3 ± 13 | -9 ± 2.6 |
> | 1 | 0 ± 0 | -2.2 ± 3 | -0.95 ± 0.7 | -8.4 ± 18 | -5.3 ± 7.3 | -4.5 ± 0.76 | -13 ± 20 | -10 ± 14 | -9.3 ± 2.1 |
> | 5 | -1.9 ± 6 | -0.69 ± 1.5 | 0 ± 0 | -8.4 ± 18 | -5.1 ± 5.9 | 0 ± 0 | -13 ± 20 | -13 ± 15 | -4.8 ± 3.5 |
>
> _Table 2_: CVaR
> | Δt | α=2, δ=0.9 | α=2, δ=0.95 | α=2, δ=0.99 | α=5, δ=0.9 | α=5, δ=0.95 | α=5, δ=0.99 | α=10, δ=0.9 | α=10, δ=0.95 | α=10, δ=0.99 |
> |-|-|-|-|-|-|-|-|-|-|
> | -5 | 0 ± 0 | 0 ± 0 | 0 ± 0 | 0 ± 0 | 0 ± 0 | 0 ± 0 | 1 ± 2.3 | 0 ± 0 | 1.3 ± 2.1 |
> | -1 | 0 ± 0 | 0 ± 0 | 0.27 ± 0.43 | 0 ± 0 | 0.87 ± 1.8 | 2.7 ± 1.1 | 1 ± 2.3 | 1.9 ± 4 | 6.1 ± 1.6 |
> | 0 | 0 ± 0 | 0.46 ± 0.81 | 1.3 ± 0.24 | 0.39 ± 1.2 | 1.1 ± 2 | 3.3 ± 0.59 | 1.5 ± 2.5 | 2.3 ± 4.1 | 6.7 ± 1.2 |
> | 1 | 0 ± 0 | 0.68 ± 0.9 | 0.77 ± 0.61 | 1 ± 2.3 | 1.6 ± 2.2 | 3.4 ± 0.53 | 1.5 ± 2.5 | 3.2 ± 4.4 | 6.9 ± 1 |
> | 5 | 0.17 ± 0.53 | 0.31 ± 0.68 | 0 ± 0 | 1 ± 2.3 | 1.7 ± 1.9 | 0 ± 0 | 1.5 ± 2.5 | 4.2 ± 4.6 | 3.8 ± 3 |
>
> **Theorem 3**
>
> We thank the reviewer for a very detailed reading of our proof, and we apologize for the heavy notation. If the reviewer can point to specific lines that are unclear, we welcome feedback to improve the clarity of the proof.
>
> We also deeply appreciate the reviewer for identifying our regrettable mistake in applying the chain rule to the composition of differentiable-almost-everywhere functions. We noticed that we actually made the same mistake in both case (i) and case (ii). The mistake in case (i) is as the reviewer pointed out. The mistake in case (ii) lies in our claim that $\tilde{L}_{i,t}$ is continuously differentiable in $\theta$.
>
> To correct this mistake, one option is indeed to assume that $\phi$ is differentiable almost everywhere, but then this excludes CVaR disutility function $\phi_{\text{CVaR}^\delta}(x) = \frac{1}{1-\delta} [x]_+$ which is not differentiable at the origin.
>
> Instead, we will update Theorem 3 to make the more generic assumption that $\tilde{L}_{i,t}$ is differentiable almost everywhere in $(\theta, \lambda)$. And for case (i), we will directly assume that $\tilde{L}\_{i,t}$ has the form
>
> $$
> \tilde{L}\_{i,t}(\theta,\lambda) = c\_{i,0}(\theta) + \sum\_{j \in \mathbb{N}} c\_{i,j}(\theta) \cdot \mathbf{1}[g\_{i,j}(\theta) < \lambda].
> $$
>
> These changes make our theoretical result less elegant, but do not fundamentally change the applicability of the theory to CVaR and related risk measures.
>
> We also note that our mistake does not affect our empirical experiments. Because the disutility function is only non-differentiable at the origin, the likelihood in practice that $c_0(\theta) - t + \sum\_{j=1}^k c\_j(\theta) = 0$ is minimal, if not negligible.
>
> **Theorem 3 case (ii)**
>
> We apologize for the confusion about replacing $\ell_i$ with $\lambda$. What we mean is that if $\ell\_i(\theta, \lambda)$ are strictly increasing in $\lambda$, then
>
> $$\begin{aligned}
> & \arg\min\_\lambda \frac{1}{N} \sum\_{i=1} \ell\_i(\theta, \lambda)
> \quad\text{s.t.}\ \
> \tilde{h}\_t(\theta, \lambda) \leq \alpha \\\\
> &= \ \arg\min\_\lambda \frac{1}{N} \sum\_{i=1} \lambda
> \quad\text{s.t.}\ \
> \tilde{h}\_t(\theta, \lambda) \leq \alpha.
> \end{aligned}$$
>
> We will clarify this in the camera-ready revision.
>
> Also, the interpretation that we are reducing the problem to a standard bilevel optimization is correct. However, we are not using any duality argument.
>
>
> **Non-zero gradients**
>
> The reviewer is correct that Theorem 3 does not guarantee the gradients to be non-zero. For example, if $\ell$ and $L$ depend on a neural network that outputs a constant value, then clearly $\lambda(\theta)$ is constant, and therefore the gradient is zero. Thus, it is up to the user to define $\ell$ and $L$ such that the gradient is non-zero.
>
> **Paper title**
>
> We thank the reviewer for appreciating both of our contributions, and we are open to suggestions to improve the paper title (if permitted by the NeurIPS conference). We considered including "optimized certainty-equivalent" in the title, but we found that it made the title longer than we liked.
>
> Some options that we considered include:
>
> - Conformal Risk Training: End-to-end optimization of conformal optimized certainty-equivalent risk control
> - Conformal Risk Training: Learning to control optimized certainty-equivalent risks
> - Conformal Risk Training: Differentiable control of optimized certainty-equivalent risks

---

> ### Comment · Reviewer_mggh · 2025-08-03
>
> ~~Once again, we’d like to thank the reviewer for their attentiveness, and hope
> we managed to clarify our results.~~ Thank you for your response. Some additional questions/points:
>
> **The $t$ hyperparameter:**
> - Between the two inequalities, I think the first one seems by far the most important. Consider a fixed $t$ (i.e., not chosen on a separate split). While the latter becomes tight with fast $O(1/n)$ rates as $n \to \infty$, for the former this need not be the case.
>
>     It's neat that the "optimal" choice of $t$ would be given by VaR. I would like to see a proof of it, though.
>     I would also suggest adding this to the paper: $t$ is not just some arbitrary parameter, but rather one whose optimal choice corresponds to a fairly meaningful quantity, which could perhaps inform a practitioner with little data on how to choose it.
> - Thank you for the tables, but I'm afraid I don't quite understand them. What exactly is the 'task loss'? What is the range of values for the task loss, cvar (or rather, the undelying data for the cvar) and for the $t$? And, just to double-check, the $\alpha$ should be the desired upper bound for the CVaR, right? Without this information it's hard to interpret the results.
>
> **Theorem 3:**
> - Thank you for the correction. I think I'm good with case (ii) (though I suggest the be more explicit in the text about the convex conic problem bit). As for case (i), I do see how the proposed assumption would resolve the issue. However, I am concerned about how applicable the theorem then becomes. Can you show me that this assumption does hold for CVaR, and maybe for some other risk measure?
> - Regarding clarity, I'm afraid there aren't really any specific lines that are hard to follow, but rather the proof as a whole. It is technical and with lots of notation (to be frank, there are just too many letters), which makes it hard to follow. It also feels like there isn't a clear thread connecting each step. Perhaps the easiest improvement would be to try to reduce the amount of notation, and/or to reintroduce all of them right before the proof (e.g. in a table).
>
> **Non-zero gradients:** thank you for your response. I think this may be worth briefly mentioning in the main paper.
>
> **Paper title:** the proposed alternate titles would be definite improvements. If length is a concern, perhaps "optimized certainty-equivalent" could be shortened to OCE. I would perhaps also suggest changing "Conformal Risk Training" to something like "Conformal OCE Risk Training" to further distance from plain "Conformal Training + (Vanilla) Conformal Risk Control".

---

> ### Comment · Area_Chair_JXu6 · 2025-08-04
>
> Dear reviewer, please engage with the author rebuttal.

---

> ### Author Response · Authors · 2025-08-07
> **Response Part 1/2**
>
> We thank the reviewer for detailed feedback and questions. Due to character count limits, our response is split across 2 comments.
>
> **Non-zero gradients**
>
> In our camera-ready revision, we will add a note to Theorem 3 that when the gradient exists, it may be zero.
>
> **Paper title**
>
> We are not sure about whether NeurIPS allows changing the title after submission. If the Area Chair could clarify the policy, that would be helpful.
>
> **$t$ hyperparameter**
>
> The reviewer correctly notes that the first inequality does not depend on $N$, whereas the 2nd inequality becomes tight with rate $O(1/N)$. This is evident from Theorems 1 and 2 in the original CRC paper [1], which show that when $L_{i,t}$ are i.i.d. and continuous, then
>
> $$
> \alpha - \frac{2 \tilde{B}\_t(\hat\lambda)}{N+1}
> \leq
> \mathbb{E}\left[ \tilde{L}\_{N+1,t}(\hat\lambda) \right]
> \leq \alpha.
> $$
>
> Thus, for large enough $N$, choosing $t$ to be close to the minimizer of the OCE risk definition will be nearly optimal.
>
> In the case of CVaR, the fact that $t = \text{VaR}$ achieves the infimum in the OCE risk definition is a classical result from optimization (see, e.g., Sections 2-3 in [2]). We refer to Table 1 in [3] for a list of minimizers of other common OCE risks such as mean-variance and entropic risk.
>
> _References_
>
> [1] Anastasios Nikolas Angelopoulos, Stephen Bates, Adam Fisch, Lihua Lei, and Tal Schuster. 2023. Conformal Risk Control. In _International Conference on Learning Representations_, 2023. https://openreview.net/forum?id=33XGfHLtZg
>
> [2] Stanislav Uryasev and R. Tyrrell Rockafellar. 2001. Conditional value-at-risk: optimization approach. In _Stochastic Optimization: Algorithms and Applications_. Springer US, Boston, MA, 411–435. https://doi.org/10.1007/978-1-4757-6594-6_17
>
> [3] Ayon Ghosh, L. A. Prashanth, and Krishna Jagannathan. 2024. Concentration bounds for optimized certainty equivalent risk estimation. https://doi.org/10.48550/arXiv.2405.20933
>
> **$t$ sensitivity tables**
>
> We apologize for not clearly explaining the tables we posted in the rebuttal. The "task loss" refers to $f(y, \lambda \hat{z}\_\theta(x))$ from the battery storage example. Let $D = \\{(x_i,y_i)\\}\_{i=1}^{438}$ denote the test set of (input features, electricity prices) examples.
>
> For $\lambda = 1$, the optimal task loss on the test set ranges from -544 to -61 (assuming perfect knowledge of $Y$, and thus perfect decisions $z$). When using electricity prices predicted by models $\hat{y}\_\theta$ trained to minimize mean-squared error, the decision is made as $\hat{z}\_\theta(x) = \arg\min\_{z \in \mathcal{Z}} f(\hat{y}\_\theta(x), z)$, and the task loss ranges from -446 to +69.
>
> Your understanding of $\delta$, and $\alpha$ is correct: $\delta \in (0,1)$ is the quantile level of the tail, and $\alpha$ is the desired upper bound on $\text{CVaR}\_{(X,Y,\lambda)}^\delta[f(Y,\lambda \hat{z}\_\theta(X))]$.
>
> The tables show the mean ± 1 std of the task loss $\mathbb{E}\_{(X,Y) \sim D}[f(Y,\lambda \hat{z}\_\theta(X))]$ and the CVaR $\text{CVaR}^\delta\_{(X,Y) \sim D}[f(Y,\lambda \hat{z}\_\theta(X))]$, respectively, for 10 models $\hat{y}\_\theta$ trained with different random seeds. The random seed sets the model's weight initialization as well as the training vs. calibration data split used for calibrating $\lambda$.

---

> ### Author Response · Authors · 2025-08-07
> **Response Part 2/2**
>
> **$t$ sensitivity tables, continued**
>
> Following another reviewer's suggestion (Reviewer UbDa), we also implemented sensitivity analysis with a relative change in $t$, in addition to an absolute change in $t$. In doing so, we noticed a bug that affected our sensitivity analysis for absolute changes in $t$. Let $t_0$ denote the optimal value of $t$ tuned on the training set. In Theorem 2, our CVaR risk guarantee only holds when $t \geq B(\lambda_{\min})$, but we forgot to impose this constraint on $t_0 + \Delta t$. We fixed the bug, and the tables below report sensitivity analyses for both absolute and relative changes in $t$. The cases where $t_0 + \Delta t < 0$ are reported as NaN. (In the battery storage problem, $B(\lambda_{\min}) = 0$.)
>
> The following table shows the values of $t_0$ (mean ± 1 std) selected by tuning on the training set:
>
> | | α=2, δ=.9 | α=2, δ=.95 | α=2, δ=.99 | α=5, δ=.9 | α=5, δ=.95 | α=5, δ=.99 | α=10, δ=.9 | α=10, δ=.95 | α=10, δ=.99 |
> |-|-|-|-|-|-|-|-|-|-|
> | t | 0.0002 ± 0.00064 | 5.4e-10 ± 1.2e-09 | 0.81 ± 0.29 | 0.043 ± 0.03 | 0.00035 ± 0.0011 | 2 ± 0.73 | 0.15 ± 0.077 | 0.094 ± 0.07 | 4.1 ± 1.5 |
>
> _Results for absolute $\Delta t$, _i.e._, $t = t_0 + \Delta t$_:
>
> Task loss
>
> | Δt (absolute) | α=2, δ=.9 | α=2, δ=.95 | α=2, δ=.99 | α=5, δ=.9 | α=5, δ=.95 | α=5, δ=.99 | α=10, δ=.9 | α=10, δ=.95 | α=10, δ=.99 |
> |-|-|-|-|-|-|-|-|-|-|
> | -5 | nan | nan | nan | nan | nan | nan | nan | nan | -6.5 ± 0.92 |
> | -1 | nan | nan | -1.3 ± 0.18 | nan | nan | -4.1 ± 1.2 | nan | nan | -9.1 ± 2.3 |
> | 0 | -8.6 ± 3.2 | -4.3 ± 1.6 | -1.8 ± 0.51 | -21 ± 7.9 | -11 ± 4 | -4.5 ± 1.3 | -36 ± 6.3 | -21 ± 7.9 | -9 ± 2.6 |
> | 1 | -6.8 ± 1.9 | -4.4 ± 1.1 | -0.95 ± 0.7 | -20 ± 6.7 | -11 ± 3.6 | -4.5 ± 0.76 | -35 ± 6.3 | -22 ± 7.5 | -9.3 ± 2.1 |
> | 5 | 0 | 0 | 0 | -0.41 ± 0.87 | -1.9 ± 0.65 | 0 | -32 ± 7.2 | -22 ± 5.6 | -4.8 ± 3.5 |
>
> CVaR
>
> | Δt (absolute) | α=2, δ=.9 | α=2, δ=.95 | α=2, δ=.99 | α=5, δ=.9 | α=5, δ=.95 | α=5, δ=.99 | α=10, δ=.9 | α=10, δ=.95 | α=10, δ=.99 |
> |-|-|-|-|-|-|-|-|-|-|
> | -5 | nan | nan | nan | nan | nan | nan | nan | nan | 4.4 ± 0.25 |
> | -1 | nan | nan | 0.89 ± 0.05 | nan | nan | 3 ± 0.51 | nan | nan | 6.6 ± 0.7 |
> | 0 | 1.6 ± 0.32 | 1.6 ± 0.3 | 1.3 ± 0.24 | 4 ± 0.78 | 4 ± 0.77 | 3.3 ± 0.59 | 7.3 ± 2.1 | 8.2 ± 1.5 | 6.7 ± 1.2 |
> | 1 | 1.3 ± 0.31 | 1.7 ± 0.21 | 0.77 ± 0.61 | 3.8 ± 0.78 | 4.3 ± 0.66 | 3.4 ± 0.53 | 7.3 ± 2.1 | 8.5 ± 1.4 | 6.9 ± 1 |
> | 5 | 0 | 0 | 0 | 0.061 ± 0.14 | 0.72 ± 0.31 | 0 | 6.5 ± 1.6 | 8.5 ± 1 | 3.8 ± 3 |
>
> _Results for relative $\Delta t$, _i.e._, $t = (1 + \Delta t) t_0$_:
>
> Task loss
>
> | Δt (relative) | α=2, δ=.9 | α=2, δ=.95 | α=2, δ=.99 | α=5, δ=.9 | α=5, δ=.95 | α=5, δ=.99 | α=10, δ=.9 | α=10, δ=.95 | α=10, δ=.99 |
> |-|-|-|-|-|-|-|-|-|-|
> | -0.5 | -8.6 ± 3.2 | -4.3 ± 1.6 | -1.5 ± 0.49 | -21 ± 8 | -11 ± 4 | -3.9 ± 1.2 | -36 ± 6.3 | -21 ± 7.9 | -7.7 ± 2.5 |
> | -0.1 | -8.6 ± 3.2 | -4.3 ± 1.6 | -1.8 ± 0.52 | -21 ± 7.9 | -11 ± 4 | -4.4 ± 2.3 | -36 ± 6.3 | -21 ± 7.9 | -8.9 ± 2.6 |
> | 0 | -8.6 ± 3.2 | -4.3 ± 1.6 | -1.8 ± 0.51 | -21 ± 7.9 | -11 ± 4 | -4.5 ± 1.3 | -36 ± 6.3 | -21 ± 7.9 | -9 ± 2.6 |
> | 0.1 | -8.6 ± 3.2 | -4.3 ± 1.6 | -1.8 ± 0.5 | -21 ± 7.9 | -11 ± 4 | -4.6 ± 1.2 | -36 ± 6.3 | -21 ± 7.9 | -9.1 ± 2.5 |
> | 0.5 | -8.6 ± 3.2 | -4.3 ± 1.6 | -1.7 ± 0.4 | -21 ± 7.9 | -11 ± 4 | -4.1 ± 1 | -36 ± 6.3 | -22 ± 7.9 | -8.3 ± 2 |
>
> CVaR
>
> | Δt (relative) | α=2, δ=.9 | α=2, δ=.95 | α=2, δ=.99 | α=5, δ=.9 | α=5, δ=.95 | α=5, δ=.99 | α=10, δ=.9 | α=10, δ=.95 | α=10, δ=.99 |
> |-|-|-|-|-|-|-|-|-|-|
> | -0.5 | 1.6 ± 0.32 | 1.6 ± 0.3 | 1.1 ± 0.22 | 4 ± 0.78 | 4 ± 0.77 | 2.8 ± 0.55 | 7.3 ± 2.1 | 8.2 ± 1.6 | 5.7 ± 1.1 |
> | -0.1 | 1.6 ± 0.32 | 1.6 ± 0.3 | 1.3 ± 0.24 | 4 ± 0.78 | 4 ± 0.77 | 3.3 ± 0.59 | 7.3 ± 2.1 | 8.2 ± 1.6 | 6.5 ± 1.2 |
> | 0 | 1.6 ± 0.32 | 1.6 ± 0.3 | 1.3 ± 0.24 | 4 ± 0.78 | 4 ± 0.77 | 3.3 ± 0.59 | 7.3 ± 2.1 | 8.2 ± 1.5 | 6.7 ± 1.2 |
> | 0.1 | 1.6 ± 0.32 | 1.6 ± 0.3 | 1.3 ± 0.23 | 4 ± 0.78 | 4 ± 0.77 | 3.4 ± 0.59 | 7.3 ± 2.1 | 8.2 ± 1.5 | 6.7 ± 1.2 |
> | 0.5 | 1.6 ± 0.32 | 1.6 ± 0.3 | 1.2 ± 0.29 | 4 ± 0.78 | 4 ± 0.77 | 3.1 ± 0.73 | 7.3 ± 2.1 | 8.2 ± 1.5 | 6.2 ± 1.5 |
>
> Especially from the relative $\Delta t$ tables, it is clear that the task loss and CVaR are not very sensitive to changes in $t$. In particular, these results show that the sensitivity to $t$ is _less_ than that suggested by the tables we presented in our initial rebuttal that did not include the needed constraint on $t$.
>
> **Theorem 3**
>
> We are working on an improved re-write of Theorem 3 to address the concerns you raised, and we hope to share it with you soon. We have a version of the proof that ensures the updated assumption is valid for neural networks with smooth activation functions, and we are working on a generalization to ReLU-like activations.
>
> In terms of improving the clarity of the proof, we are splitting the proof into several lemmas which are easier to understanding on their own.

---

> > ### Comment · Reviewer_mggh · 2025-08-08
> >
> > Thank you for the clarifications. These sensitivity analyses are very relevant, and I strongly suggest their inclusion in the final version. I also look forward to seeing the rewrite of Theorem 3. \
> > Aside from the title (which hinges on conference policy), my concerns were sufficiently addressed.

---

> > > ### Author Response · Authors · 2025-08-09
> > > **Response Part 1/6: Executive Summary and Revised Theorem 3 Statement**
> > >
> > > We are deeply grateful to Reviewer mggh for their detailed reading of our Theorem 3 statement and proof. The reviewer's feedback has significantly improved the quality, readability, and correctness of our work.
> > >
> > > In this comment, we give a summary of the changes we make to Theorem 3 and its proof, as well as the updated complete statement of Theorem 3. In subsequent comments, we give the proof. To the extent possible, we have kept the numbering of equations, theorems, lemmas, and assumptions consistent with the original submission.
> > >
> > >
> > > ## Executive summary
> > >
> > > We have made the assumptions of Theorem 3 more explicit, while keeping the spirit of the original statement intact. Below is a summary of the main changes. Throughout our comments, we may abbreviate "almost everywhere" as "a.e."
> > >
> > > For Setting (i):
> > >
> > > 1. We realized that it suffices for the coefficients $c\_{i,j}(\theta)$ to be continuous functions of $\theta$. In fact, neither $c\_{i,j}$ nor $\phi$ need to be differentiable.
> > >
> > > 2. To enforce continuity of $\tilde{h}\_t$, we require that all level sets of $g\_{i,j}$ have measure zero. We note that this condition is not difficult to satisfy. For example, it holds for neural networks with a trainable bias term in the final output layer, such as those we use in our experiments.
> > >
> > > 3. We clarify that all $d\_j$ and $g\_{i,j}(\theta)$ must take on unique values in order for the gradient to exist, for the same reason that $f(x,y)=\max(x,y)$ is only differentiable when $x \neq y$.
> > >
> > > 4. We simplify the notation in the proof by refactoring several notation-heavy steps into standalone lemmas.
> > >
> > >
> > > For Setting (ii):
> > >
> > > 1. Whereas we previously assumed existence of $\lambda'(\theta)$, we now state explicit sufficient conditions (from the Implicit Function Theorem) that ensure its existence. We hope this addresses your request that we be "more explicit in the text about the convex conic problem bit."
> > >
> > > We present the revised theorem statement below, and for completeness, we have copied all proofs into subsequent comments. The core ideas of the proofs remain the same from the original proofs, and we hope that the revised structure is easier to follow, even if it is more verbose.
> > >
> > >
> > > ## Complete statement of Theorem 3
> > >
> > > Let $\phi:\Bbb{R}\to\Bbb{R}$ be an OCE disutility function. Suppose Assumptions 1 and 2 hold and that the inner problem in Eq. (7) is feasible.
> > >
> > > (i) Suppose $\sum\_{i=1}^N \ell\_i$ is strictly decreasing in $\lambda$; the set $\\{\theta\in\Theta \mid \tilde{h}\_t(\theta,\lambda)=\alpha\\}$ has measure zero for all $\lambda$; and $\\{B\\}\cup\\{L\_i\\}\_{i=1}^N$ are piecewise constant in $\lambda$. Under the standard differentiability assumptions in Assumption 6 (see below), we may obtain a closed-form expression for the derivative $\lambda'(\theta)$ a.e.
> > >
> > > (ii) Suppose $\sum\_{i=1}^N \ell\_i$ is strictly convex or strictly monotone in $\lambda$, and $\\{B\\}\cup\\{L\_i\\}\_{i=1}^N$ are convex in $\lambda$. Under the standard differentiability assumptions in Assumption 7 (see below), the derivative $\lambda'(\theta)$ exists and has a closed-form expression.

---

> > > ### Author Response · Authors · 2025-08-09
> > > **Response Part 2/6: Assumptions and Lemmas for Theorem 3(i)**
> > >
> > > ## Assumption 6
> > > The functions $B$ and $\\{L\_i\\}\_{i=1}^N$ are of the form
> > >
> > > $$
> > > B(\lambda)
> > > =b\_0+\sum\_{j\in\Bbb{N}} b\_j\cdot{\bf 1}[d\_j<\lambda],\qquad
> > > L\_i(\theta,\lambda)=c\_{i,0}(\theta)+\sum\_{j\in\Bbb{N}} c\_{i,j}(\theta)\cdot{\bf 1}[g\_{i,j}(\theta)<\lambda]
> > > $$
> > >
> > > where
> > >
> > > 1. $(c\_{i,j}:\Theta\to\Bbb{R})\_{i\in[N], j\in\Bbb{Z}\_+}$ and $(g\_{i,j}:\Theta\to\Bbb{R})\_{i\in[N],j\in\Bbb{N}}$ are continuous functions;
> > > 1. $(b\_j\in\Bbb{R})\_{j\in\Bbb{N}}$ and $(c\_{i,j}(\theta))\_{i\in[N], j\in\Bbb{N}}$ are nonnegative;
> > > 1. $(d\_j\in\Bbb{R})\_{j\in\Bbb{N}}$ and $(g\_{i,j}(\theta))\_{j\in\Bbb{N}}$ for all $i$ are nondecreasing sequences, and $\\{d\_j\\}\_{j\in\Bbb{N}}\cup\\{g\_{i,j}(\theta)\\}\_{i\in[N], j\in\Bbb{N}}$ are all unique; and
> > > 1. $(g\_{i,j})\_{i\in[N],j\in\Bbb{N}}$ are differentiable a.e., and all of their level sets have measure zero.
> > >
> > > **Practicality of these assumptions:** In Example 2 from our submission, $c\_{i,j}$ are constants, and $g\_{i,j}$ are neural networks. Neural networks with standard activation functions (e.g., ReLU, sigmoid, tanh) are continuous everywhere and differentiable a.e. with respect to their parameters. Furthermore, if they have a bias term in their final output layer, then their level sets (w.r.t. their parameters) have measure zero as a consequence of Tonelli's Theorem, thus satisfying point 4 above.
> > >
> > > ## Lemma 1
> > > If $\phi:\Bbb{R}\to\Bbb{R}$ is a disutility function corresponding to an optimized certainty equivalent risk, then $\phi$ is continuous.
> > >
> > > *Proof*: Convex, real-valued functions (that are not infinite) are continuous. $\blacksquare$
> > >
> > > ## Lemma 3
> > > Suppose $L:\Theta\times\Lambda\to\Bbb{R}$ is a step function in $\lambda$ of the form
> > >
> > > $$
> > > L(\theta,\lambda)={\bf 1}[g(\theta)<\lambda]
> > > $$
> > >
> > > where $g:\Theta\to\Bbb{R}$ is continuous and whose level sets have measure zero. Then, $L$ is continuous a.e. in $\theta$.
> > >
> > > *Proof*: Fix a scalar $\lambda$, and define the preimages $U=g^{-1}((-\infty,\lambda))$, $V=g^{-1}((\lambda,\infty))$, and $W=g^{-1}(\\{\lambda\\})$. Clearly, $\Theta=U \cup V \cup W$. Since the level sets of $g$ have measure zero, $W$ has measure 0.
> > >
> > > Since $g$ is continuous, $U,V$ are open sets. For any $\theta \in U$, $L(\theta,\lambda)=1$ is constant, so $L$ is continuous on $U$. Likewise, $L$ is constant and continuous on $V$.
> > >
> > > Thus, $L$ is continuous in $\theta$ everywhere except on $W$, a set of measure 0. $\blacksquare$
> > >
> > > ## Lemma 4
> > > Suppose $L:\Theta\times\Lambda\to\Bbb{R}$ is piecewise constant, nondecreasing in $\lambda$ with the form
> > >
> > > $$
> > > L(\theta,\lambda)=c\_0(\theta)+\sum\_{j\in\Bbb{N}} c\_j(\theta)\cdot{\bf 1}[g\_j(\theta)<\lambda]
> > > $$
> > >
> > > where $(c\_j:\Theta\to\Bbb{R})\_{j\in\Bbb{Z}\_+}$ and $(g\_j:\Theta\to\Bbb{R})\_{j\in\Bbb{N}}$ are continuous functions, $(c\_j(\theta))\_{j\in\Bbb{N}}$ are nonnegative, and $(g\_j(\theta))\_{j\in\Bbb{N}}$ is nondecreasing. Assume that the level sets of $(g\_j)\_{j\in\Bbb{N}}$ have measure zero.
> > >
> > > Let $\phi:\Bbb{R}\to\Bbb{R}$ be an OCE disutility function, and let $t\in\Bbb{R}$. Then, the function $\tilde{L}:\Theta\times\Lambda\to\Bbb{R}$ defined as
> > >
> > > $$
> > > \tilde{L}(\theta,\lambda)
> > > :=t+\phi(L(\theta,\lambda)-t)
> > > =t+\phi\left(c\_0(\theta) - t+\sum\_{j\in\Bbb{N}} c\_j(\theta)\cdot{\bf 1}[g\_j(\theta)<\lambda]\right)
> > > $$
> > >
> > > is continuous a.e. in $\theta$ and can be written in the same form as $L$, i.e.,
> > >
> > > $$
> > > \tilde{L}(\theta,\lambda)=\tilde{c}\_0(\theta)+\sum\_{j\in\Bbb{N}}\tilde{c}\_j(\theta)\cdot{\bf 1}[g\_j(\theta)<\lambda]
> > > $$
> > >
> > > where $(\tilde{c}\_j)\_{j\in\Bbb{Z}\_+}$ are continuous functions and $(\tilde{c}\_j(\theta))\_{j\in\Bbb{N}}$ are nonnegative.
> > >
> > > _Proof_: For $k\in\Bbb{N}$, define
> > >
> > > $$
> > > c'\_k(\theta):=t+\phi\left(c\_0(\theta)-t+\sum\_{j=1}^k c\_j(\theta)\right).
> > > $$
> > >
> > > Since $\phi$ is a nondecreasing function and $(c\_j(\theta))\_{j\in\Bbb{N}}$ are nonnegative, $(c\_j'(\theta))\_{j\in\Bbb{Z}\_+}$ is a nondecreasing sequence of reals. $(c\_j)\_{j\in\Bbb{Z}\_+}$ are continuous by assumption, and $\phi$ is continuous by Lemma 1, so $(c\_j')\_{j\in\Bbb{Z}\_+}$ are continuous functions.
> > >
> > > Observe that $\tilde{L}$ is also piecewise constant, nondecreasing in $\lambda$:
> > >
> > > $$\begin{aligned}
> > > \tilde{L}(\theta,\lambda)&=\begin{cases}
> > > c\_0'(\theta),&\lambda\in(-\infty,g\_1(\theta)] \\\\
> > > c\_k'(\theta),&\lambda\in(g\_k(\theta),g\_{k+1}(\theta)]
> > > \end{cases}\\\\
> > > &=\tilde{c}\_0(\theta)+\sum\_{j\in\Bbb{N}}\tilde{c}\_j(\theta)\cdot{\bf 1}[g\_j(\theta)<\lambda],
> > > \end{aligned}$$
> > >
> > > where $\tilde{c}\_0(\theta)=c\_0'(\theta)$ and $\tilde{c}\_j(\theta) :=c\_j'(\theta) - c\_{j-1}'(\theta)$ for all $j\in\Bbb{N}$.
> > >
> > > Because $(c\_j'(\theta))\_{j\in\Bbb{Z}\_+}$ are nondecreasing, $(\tilde{c}\_j(\theta))\_{j\in\Bbb{N}}$ are nonnegative. Furthermore, $(\tilde{c}\_j)\_{j\in\Bbb{Z}\_+}$ inherit continuity from $(c\_j')\_{j\in\Bbb{Z}\_+}$.
> > >
> > > By Lemma 3, each ${\bf 1}[g\_j(\theta)<\lambda]$ is continuous a.e. in $\theta$. The countable product and countable sum of continuous a.e. functions is continuous a.e., so $\tilde{L}$ is continuous a.e. in $\theta$. $\blacksquare$

---

> > > ### Author Response · Authors · 2025-08-09
> > > **Response Part 3/6: Proof of Theorem 3(i)**
> > >
> > > By Lemma 4, each $\tilde{L}\_{i,t}(\theta,\lambda) :=t+\phi(L\_i(\theta,\lambda) - t)$ is piecewise constant in $\lambda$ and can be written in the form
> > >
> > > $$
> > > \tilde{L}\_{i,t}(\theta,\lambda)
> > > =\tilde{c}\_{i,0}(\theta)+\sum\_{j\in\Bbb{N}} \tilde{c}\_{i,j}(\theta)\cdot{\bf 1}[g\_{i,j}(\theta)<\lambda],
> > > $$
> > >
> > > where $(\tilde{c}\_{i,j})\_{j\in\Bbb{Z}\_+}$ are continuous functions $(\tilde{c}\_{i,j}(\theta))\_{j\in\Bbb{N}}$ are nonnegative. Similarly, $\tilde{B}\_t$ is piecewise constant and can be written in the form
> > >
> > > $$
> > > \tilde{B}\_t(\lambda)
> > > =\tilde{b}\_0+\sum\_{j\in\Bbb{N}} \tilde{b}\_j\cdot{\bf 1}[d\_j<\lambda]
> > > $$
> > >
> > > where $(\tilde{b}\_j)\_{j\in\Bbb{N}}$ are nonnegative. Thus, $\tilde{h}\_t(\theta,\lambda) :=\frac{1}{N+1} \left(\tilde{B}\_t(\lambda)+\sum\_{i=1}^N \tilde{L}\_{i,t}(\theta,\lambda)\right)$ is piecewise constant and can be written in the form
> > >
> > > $$
> > > \tilde{h}\_t(\theta,\lambda)
> > > =c\_0^{(h)}(\theta)+\sum\_{j\in\Bbb{N}} c\_j^{(h)}(\theta)\cdot{\bf 1}[g\_j^{(h)}(\theta)<\lambda].
> > > $$
> > >
> > > The sequence of steps is given by
> > >
> > > $$
> > > (g\_j^{(h)}(\theta))\_{j\in\Bbb{N}}=\mathrm{sort\\_ascending}(\\{d\_j\\}\_{j\in\Bbb{N}}\cup\\{g\_{i,j}(\theta)\\}\_{i\in[N], j\in\Bbb{N}}),
> > > $$
> > >
> > > which yields a strictly increasing sequence because $\\{d\_j\\}\_{j\in\Bbb{N}}\cup\\{g\_{i,j}(\theta)\\}\_{i\in[N], j\in\Bbb{N}}$ are all unique by assumption. The nonnegative coefficients $(c\_j^{(h)}(\theta))\_{j\in\Bbb{Z}\_+}$ are defined appropriately in terms of $(\tilde{b}\_j)\_{j\in\Bbb{Z}\_+}$ and $(\tilde{c}\_{i,j})\_{i\in[N], j\in\Bbb{Z}\_+}$, and $(c\_j^{(h)})\_{j\in\Bbb{Z}\_+}$ inherit their continuity.
> > >
> > > The functions $(g\_j^{(h)})\_{j\in\Bbb{N}}$ are differentiable a.e. and continuous; these properties are inherited from $(d\_j)\_{j\in\Bbb{N}}$ (which are constant) and $(g\_{i,j})\_{i\in[N],j\in\Bbb{N}}$ (which are differentiable a.e. and continuous by assumption).
> > >
> > > By Lemma 4, each $\tilde{L}\_{i,t}$ is continuous a.e. in $\theta$. $\tilde{B}\_t$ is constant in $\theta$, and therefore also continuous in $\theta$. $\tilde{h}\_t$ is a countable, weighted sum of $\tilde{B}\_t$ and $\tilde{L}\_{i,t}$ for $i\in[N]$, so it is continuous a.e. in $\theta$.
> > >
> > > Recall from (7) that
> > >
> > > $$
> > > \lambda(\theta)
> > > :=\arg\min\_{\lambda\in\Lambda} \frac{1}{N} \sum\_{i=1}^N \ell\_i(\theta,\lambda)
> > > \quad\text{s.t. } \tilde{h}\_t(\theta,\lambda) \leq \alpha.
> > > $$
> > >
> > > Because each $\ell\_i$ is strictly decreasing in $\lambda$, we may equivalently write the optimization problem as maximizing $\lambda$ subject to the constraint on $\tilde{h}\_t$ and leave the optimal solution $\lambda(\theta)$ unaffected:
> > >
> > > $$
> > > \lambda(\theta)=\max\_{\lambda\in\Lambda} \lambda
> > > \quad\text{s.t. } \tilde{h}\_t(\theta,\lambda) \leq \alpha.
> > > $$
> > >
> > > Since $\\{B\\}\cup\\{L\_i\\}\_{i=1}^N$ are piecewise constant, left-continuous, nondecreasing functions in $\lambda$, and $\phi$ is continuous (Lemma 1) and nondecreasing, $\\{\tilde{B}\_t,\,\tilde{h}\_t\\}\cup\\{\tilde{L}\_{i,t}\\}\_{i=1}^N$ are also piecewise constant, left-continuous, and nondecreasing in $\lambda$.
> > >
> > > _continued in next comment..._

---

> > > ### Author Response · Authors · 2025-08-09
> > > **Response Part 4/6: Continuation of Proof of Theorem 3(i)**
> > >
> > > _...continued from previous comment_
> > >
> > > By assumption, the optimization problem is feasible and $\tilde{h}\_t(\theta,\lambda) \neq \alpha$ $\theta$-almost everywhere, so one of the two following cases must hold $\theta$-almost everywhere:
> > >
> > > 1. $\tilde{h}\_t(\theta,\lambda)<\alpha$ for every $\lambda\in\Lambda$.
> > >
> > >    Then, $\lambda(\theta)=\max\Lambda$. We showed above that $\tilde{h}\_t(\theta,\max\Lambda)$ is continuous a.e. in $\theta$. Thus, for $\theta$ a.e., there exists a neighborhood $\mathcal{N}(\theta)$ around $\theta$ for which $\tilde{h}\_t(\theta',\max\Lambda)<\alpha$ for all $\theta'\in\mathcal{N}(\theta)$. Thus, $\lambda(\theta)=\max \Lambda$ for all $\theta'\in\mathcal{N}(\theta)$, so $\lambda'(\theta)=\frac{d}{d\theta} \max\Lambda=0$.
> > >
> > > 2. There exists an index $k\in\Bbb{N}$ such that
> > >
> > >    $$\begin{aligned}
> > >    c\_0^{(h)}(\theta)+\sum\_{j=1}^k c\_j^{(h)}(\theta)\cdot{\bf 1}[g\_j^{(h)}(\theta)<\lambda(\theta)] &< \alpha, \\\\
> > >    c\_0^{(h)}(\theta)+\sum\_{j=1}^{k+1} c\_j^{(h)}(\theta)\cdot{\bf 1}[g\_j^{(h)}(\theta)<\lambda(\theta)] &> \alpha.
> > >    \end{aligned}$$
> > >
> > >    In this case, because the objective seeks to maximize $\lambda$ and $(g\_j^{(h)}(\theta))\_{j\in\Bbb{N}}$ is a strictly increasing sequence, $\lambda(\theta)=g\_{k+1}^{(h)}(\theta)$.
> > >
> > >    As shown above, $g\_{k+1}^{(h)}$ is differentiable a.e. Now, suppose that $g\_{k+1}^{(h)}$ is differentiable at $\theta$. Consider any entry $\theta\_q$ of the parameter vector $\theta$, and let $e\_q$ denote the standard unit basis vector along coordinate $q$. Then the partial derivative of $\lambda$ with respect to $\theta\_q$ is
> > >
> > >    $$\begin{aligned}
> > >    \frac{\partial\lambda}{\partial\theta\_q}(\theta)
> > >    &=\lim\_{s \to 0} \frac{\lambda(\theta+s e\_q) - \lambda(\theta)}{s} \\\\
> > >    &=\lim\_{s \to 0} \frac{\lambda(\theta+s e\_q) - g\_{k+1}^{(h)}(\theta)}{s} \\\\
> > >    &=\lim\_{s \to 0} \frac{g\_{k+1}^{(h)}(\theta+s e\_q) - g\_{k+1}^{(h)}(\theta)}{s} \\\\
> > >    &=\frac{\partial g\_{k+1}^{(h)}}{\partial \theta\_q}(\theta).
> > >    \end{aligned}$$
> > >
> > >    The third equality comes from observing that $(c\_j^{(h)})\_{j\in\Bbb{Z}\_+}$ and $(g\_j^{(h)})\_{j\in\Bbb{N}}$ are continuous functions, so there exists an $\epsilon > 0$ such that for all $s\in[0,\epsilon]$,
> > >
> > >    $$\begin{aligned}
> > >    c\_0^{(h)}(\theta+se\_q)+\sum\_{j=1}^k c\_j^{(h)}(\theta+se\_q)\cdot{\bf 1}[g\_j^{(h)}(\theta+se\_q)<\lambda(\theta+se\_q)]
> > >    &< \alpha, \\\\
> > >    c\_0^{(h)}(\theta+se\_q)+\sum\_{j=1}^{k+1} c\_j^{(h)}(\theta+se\_q)\cdot{\bf 1}[g\_j^{(h)}(\theta+se\_q)<\lambda(\theta+se\_q)]
> > >    &> \alpha.
> > >    \end{aligned}$$
> > >
> > >    Thus, for all $s\in[0,\epsilon]$, $\lambda(\theta+s e\_q)=g\_{k+1}^{(h)}(\theta+s e\_q)$.
> > >
> > >    Since $\frac{\partial \lambda}{\partial\theta\_q}=\frac{\partial g\_{k+1}^{(h)}}{\partial\theta\_q}$ for all coordinates $q$, we have $\lambda'(\theta)=\frac{dg\_{k+1}^{(h)}}{d\theta}(\theta)$.
> > >
> > > Therefore, we have derived a closed-form expression for $\lambda'(\theta)$ almost everywhere. $\blacksquare$

---

> > > ### Author Response · Authors · 2025-08-09
> > > **Response Part 5/6: Assumptions and Lemmas for Theorem 3(ii)**
> > >
> > > ## Assumption 7
> > >
> > > Suppose that the inner problem in Eq. (7) exhibits strong duality (e.g., Slater's condition holds). Let $\mu(\theta)$ denote the optimal Lagrange multiplier arising from the dual problem to Eq. (7). Define
> > >
> > > $$
> > > \tilde{\ell}(\theta,\lambda) :=\begin{cases}
> > > \lambda,
> > > &\text{if $\sum\_{i=1}^N \ell\_i$ is strictly increasing in $\lambda$} \\\\
> > > -\lambda,
> > > &\text{if $\sum\_{i=1}^N \ell\_i$ is strictly decreasing in $\lambda$} \\\\
> > > \frac{1}{N} \sum\_{i=1}^N \ell\_i(\theta,\lambda),
> > > &\text{if $\sum\_{i=1}^N \ell\_i$ is strictly convex in $\lambda$}
> > > \end{cases}.
> > > $$
> > >
> > > The following differentiability conditions hold:
> > > 1. $\frac{\partial}{\partial\lambda} \tilde\ell$ exists and is continuously differentiable in $(\theta,\lambda)$ at $(\theta,\lambda(\theta))$;
> > > 1. $\\{B\\}\cup\\{L\_i\\}\_{i=1}^N$ are twice continuously differentiable in $\lambda$ at $(\theta,\lambda(\theta))$;
> > > 1. $L$ is continuously differentiable in $\theta$ at $(\theta,\lambda(\theta))$;
> > > 1. $\phi$ is twice continuously differentiable at $B(\lambda(\theta)) - t$ and $L\_i(\theta,\lambda(\theta)) - t$ for each $i\in[N]$; and
> > > 1. $\begin{bmatrix}
> > > \frac{\partial^2}{\partial\lambda^2} \tilde\ell(\theta,\lambda(\theta))+\mu(\theta)\cdot\frac{\partial^2}{\partial\lambda^2} \tilde{h}\_t(\theta,\lambda(\theta)))
> > > &\frac{\partial\tilde{h}\_t}{\partial\lambda}(\theta,\lambda(\theta)) \\\\
> > > \mu(\theta)\cdot\frac{\partial\tilde{h}\_t}{\partial\lambda}(\theta,\lambda(\theta))
> > > &\tilde{h}\_t(\theta,\lambda(\theta)) - \alpha
> > > \end{bmatrix}$ is invertible.
> > >
> > > ## Lemma 2
> > >
> > > If $L:\Lambda\to\Bbb{R}$ is convex and $\phi:\Bbb{R}\to\Bbb{R}$ is convex and nondecreasing, then the function $\tilde{L}:\Lambda\times\Bbb{R}\to\Bbb{R}$ defined by $\tilde{L}(\lambda, t) :=t+\phi(L(\lambda) - t)$ is convex in $(\lambda, t)$.
> > >
> > > *Proof*: This lemma is unchanged from our original submission. See proof in original submission. $\blacksquare$

---

> ### Author Response · Authors · 2025-08-09
> **Response Part 6/6: Proof of Theorem 3(ii)**
>
> Since $\phi$ is convex and nondecreasing, $\\{\tilde{B}\_t\\}\cup\\{\tilde{L}\_{i,t}\\}\_{i=1}^N$ are convex in $\lambda$ by Lemma 2. Therefore, $\tilde{h}\_t$ is convex in $\lambda$.
>
> Recall from Eq. (7) that
>
> $$
> \lambda(\theta)
> :=\arg\min\_{\lambda\in\Lambda} \frac{1}{N} \sum\_{i=1}^N \ell\_i(\theta,\lambda)
> \quad\text{s.t. } \tilde{h}\_t(\theta,\lambda) \leq \alpha.
> $$
>
> If $\sum\_{i=1}^N \ell\_i$ is strictly increasing (or decreasing) in $\lambda$ but not strictly convex in $\lambda$, observe that we may replace the objective $\frac{1}{N} \sum\_{i=1}^N \ell\_i$ with simply $\lambda$ (or $-\lambda$) in Eq. (7) without changing the solution $\lambda(\theta)$. Thus,
>
> $$
> \lambda(\theta)=\min\_{\lambda\in\Lambda} \tilde\ell(\theta,\lambda)
> \quad\text{s.t. } \tilde{h}\_t(\theta,\lambda) \leq \alpha.
> $$
>
> Since $\lambda(\theta)$ is the solution to a convex optimization problem whose objective $\tilde\ell$ is either strictly convex or convex and strictly monotone, the solution is unique. Every convex optimization problem can be equivalently reformulated as a convex conic problem [1]. Then, the implicit function theorem, applied to the KKT equality conditions, gives sufficient conditions for $\lambda'(\theta)$ to exist [2].
>
> Specifically, define the Lagrangian $\mathcal{L}:\Lambda\times\Bbb{R}\times\Theta\to\Bbb{R}$ and the function $g:\Lambda\times\Bbb{R}\times\Theta\to\Bbb{R}^2$ by
>
> $$
> \begin{aligned}
> \mathcal{L}(\lambda,\mu,\theta)
> &:=\tilde\ell(\theta,\lambda)+\mu\cdot(\tilde{h}\_t(\theta,\lambda) - \alpha)
> \\\\
> g(\lambda,\mu,\theta)
> &:=\begin{bmatrix}
> \frac{\partial\mathcal{L}}{\partial\lambda}(\lambda,\mu,\theta) \\\\
> \mu\cdot(\tilde{h}\_t(\theta,\lambda) - \alpha)
> \end{bmatrix}.
> \end{aligned}
> $$
>
> The KKT equality conditions for primal-dual optimality of $(\lambda(\theta),\mu(\theta))$ are precisely given by
>
> $$
> g(\lambda(\theta),\mu(\theta),\theta)=0.
> $$
>
> By applying the implicit function theorem to the KKT equality conditions [2], if
>
> 1. The optimization problem for $\lambda(\theta)$ exhibits strong duality;
> 1. $g(\lambda(\theta),\mu(\theta),\theta)=0$;
> 1. $g$ is continuous differentiable in $(\lambda,\mu,\theta)$ at $(\lambda(\theta),\mu(\theta),\theta)$; and
> 1. $\frac{\partial g}{\partial(\lambda,\mu)}(\lambda(\theta),\mu(\theta),\theta)$ is invertible,
>
> then the derivatives $\lambda'(\theta)$ and $\mu'(\theta)$ exist and are given by
>
> $$
> \begin{bmatrix}
> \lambda'(\theta) \\\\
> \mu'(\theta)
> \end{bmatrix}
> =-\left[ \frac{\partial g}{\partial(\lambda,\mu)}(\lambda(\theta),\mu(\theta),\theta)\right]^{-1} \frac{\partial g}{\partial\theta}(\lambda(\theta),\mu(\theta),\theta).
> $$
>
> Strong duality is assumed in Assumption 7.
>
> The requirement that $g$ is continuously differentiable at $(\lambda(\theta),\mu(\theta),\theta)$ is satisfied if $\tilde{h}\_t$, $\frac{\partial\tilde\ell}{\partial\lambda}$, and $\frac{\partial\tilde{h}\_t}{\partial\lambda}$ are continuously differentiable in $(\theta,\lambda)$ at $(\theta,\lambda(\theta))$. Assumption 7 ensures that this is the case.
>
> Finally, we note that
>
> $$
> \frac{\partial g}{\partial(\lambda,\mu)}(\lambda(\theta),\mu(\theta),\theta)
> =\begin{bmatrix} \frac{\partial^2}{\partial\lambda^2} \tilde\ell(\theta,\lambda(\theta))+\mu(\theta)\cdot\frac{\partial^2}{\partial\lambda^2} \tilde{h}\_t(\theta,\lambda(\theta)))
> &\frac{\partial\tilde{h}\_t}{\partial\lambda}(\theta,\lambda(\theta)) \\\\
> \mu(\theta)\cdot\frac{\partial\tilde{h}\_t}{\partial\lambda}(\theta,\lambda(\theta))
> &\tilde{h}\_t(\theta,\lambda(\theta)) - \alpha
> \end{bmatrix},
> $$
>
> which we assume in Assumption 7 to be invertible. $\blacksquare$
>
> [1] Agrawal, Akshay, et al. “Differentiable Convex Optimization Layers.” Advances in Neural Information Processing Systems, 2019. Neural Information Processing Systems.
>
> [2] Barratt, Shane. _On the Differentiability of the Solution to Convex Optimization Problems_. arXiv:1804.05098, https://doi.org/10.48550/arXiv.1804.05098.

---

### Official Review · Reviewer_wLXq · 2025-07-02

**Clarity:** 3
**Significance:** 3
**Originality:** 3
**Rating:** 5
**Confidence:** 4

**Summary:**

This paper is exploring post-hoc risk control, for a broader class or risk measures. They particularly generalize the CRC method for the cases beyond expected loss, like Cvar. They provide precise theoretical arguments for the validity of their method. Furthermore, they extend conformal training ideas to their risk control setup, by differentiating through post-hoc risk control. This way, they enhance the efficiency of the whole decision making system, while controlling the desired risk. Their arguments are also sufficiently supported by experiments.

**Questions:**

Posed earlier.

**Ethical Concerns:**

["NO or VERY MINOR ethics concerns only"]

**Final Justification:**

I thank the authors for their response. I also have read the other reviewers comments, and I believe the paper has clear and interesting results that is of particular interest of the community. Hence, I raise my score and vote for acceptance.

**Limitations:**

I think the assumptions made should be discussed, particularly with the help of some explicit examples of loss functions.

**Quality:**

3

**Strengths And Weaknesses:**

- The conjunction of ML and reliable decision making is a timely and very important research direction. This paper studies a somewhat neglected but very important topic of "how can we use ML predictions while controlling tail sensitive downstream risks", and the findings of the paper can contribute positively to the growing understanding of the community.

- I would suggest to add some explicit examples for CRC loss function. For instance, it is not obvious how the notion of "action" or "decision" would show up in this formulation, as at the end of the day a decision maker wants to make a decision.
- The assumption 4 (the same comment applies to assumption 3) needs more clarifications. The authors claim their contribution to be in controlling Cvar, however under assumption 4, controlling Cvar is trivial by just picking $\lambda= \lambda_{\rm min}$.  Hence, perhaps the goal under assumption 4 is to pick the "largest $\lambda$ possible" while controlling Cvar. This is not clearly stated, and it is not the direct subject of study in the paper (although it looks like the algorithm does try to satisfy such an objective).
- In light of these, a more general concern emerges. Why would a 1-dimensional characterization of the decision space ($\lambda$) makes sense at all when trying to control Cvar. Although CRC also assumes a 1-dimensional characterization, however, at least some of the use cases of CRC, like coverage of sets or FDR control, do have some fundamental 1-dimensional characteristics. For the case of FDR control neyman-pearson lemma shows the existence of a 1-dimensional rule, and for prediction sets [1, 2] explore similar results.

These being said, I believe the paper exceeds the acceptance threshold, and i am willing to increase my score, if the authors take concrete actions to resolve/clarify these concerns in the revised manuscript.

[1] Least Ambiguous Set-Valued Classifiers with Bounded Error Levels
[2] Decision Theoretic Foundations for Conformal Prediction: Optimal Uncertainty Quantification for Risk-Averse Agents

---

> ### Author Rebuttal · Authors · 2025-07-31
>
> We thank the reviewer for recognizing Conformal Risk Training as a promising method for controlling tail risk by design. We respond to each point raised by the reviewer.
>
> **More examples of CRC loss functions**
>
> Our paper highlights three explicit examples of CRC loss functions: false negative rate, battery storage operations loss, and miscoverage loss.
>
> In the false negative rate loss function (see Example 1), the _decision_ refers to classifying a pixel as either tumorous (1) or benign (0), based on the model's predicted probability $f_\theta(x)_j \in [0,1]$ for pixel $j$. This decision is determined by the threshold $\lambda$, and the decision loss is measured by the false negative rate $L(\theta, \lambda)$ given in Example 1.
>
> In the battery storage optimization example (see Section 5.2), the _decision_ refers to the vector $z := (z^\text{in}, z^\text{out}, z^\text{net}) \in \mathbb{R}^{72}$ which describes how much to charge or discharge a battery for each hour of the day. The decision is scaled by the parameter $\lambda$ to reduce CVaR tail risk to the desired level.
>
> Furthermore, in Appendix E, we show how standard conformal prediction is a special case of conformal risk control that aims to control the miscoverage loss. Here, the _decision_ refers to constructing a prediction set $C_\theta(X; \lambda)$, whose size depends on the risk control parameter $\lambda$.
>
> The original Conformal Risk Control (CRC) paper (Angelopoulos et al., 2023), the Risk-Controlling Prediction Sets (RCPS) paper (Bates et al., 2021), and many other works provide various examples of monotone CRC losses, just to name a few:
> - financial risk from portfolio optimization
> - false discovery rate in multilabel classification
> - ranking loss
> - projective distance
>
> We would be happy to provide a few more examples of these monotone CRC losses in our camera-ready revision to better motivate our work.
>
> **Clarifying assumptions 3 & 4**
>
> We apologize for the lack of clarity around assumptions 3 & 4. The reviewer correctly points out that under these assumptions, picking $\lambda = \lambda_{\min}$ always achieves OCE risk control of a random loss function $L(\lambda)$. This is why we use the objective $\ell(\theta, \lambda)$ to determine an _optimal_ $\lambda$. Previous works such as Conformal Decision Theory paper (Lekeufack et al., 2024) only consider $\ell$ that are strictly decreasing in $\lambda$; in that case, they are indeed concerned with choosing the _largest_ $\lambda$ such that risk control (of expected loss) is achieved. Our framework is more general by allowing $\ell$ that are either monotone or convex in $\lambda$, in which case the optimal $\lambda$ (with respect to $\ell$) may not necessarily be the largest.
>
> We will revise our paper to clarify this point.
>
>
> **1-d characterization of decision space**
>
> As the reviewer points out, the original CRC formulation also assumes a 1-d characterization. Set coverage, FDR control, FNR control, etc. all have fundamental 1-d characteristics.
>
> For controlling CVaR in the battery storage problem, we are motivated by the "decision-scaling" rule: $z\_\theta(x, \lambda) = \lambda \cdot \hat{z}\_\theta(x)$. In this setting, $\hat{z}\_\theta(x)$ represents the decision that some base decision-model suggests. We scale this decision by $\lambda$, where $\lambda$ intuitively quantifies how much we trust the base decision-model. If we fully trust the model, we set $\lambda = 1$ and perfectly follow the base decision-model. If we don't trust the model at all, we set $\lambda = 0$, resulting in a do-nothing decision: $\hat{z}\_\theta(x) = 0$. The amount of trust (which we call "aggressiveness" in our paper) is naturally characterized by a 1-d parameter. We will revise our paper to more clearly explain this 1-d characterization.
>
> Nonetheless, we recognize that not all decisions can be easily boiled down to a 1-d parameter. Notably, the paper "How to Trust Your Diffusion Model" (Teneggi et al., 2023), provides an example of multi-dimensional $\lambda$ in the RCPS setting. We believe that investigating similar generalizations of our CORC method to accommodate multi-dimensional $\lambda$ would be an interesting direction of future work.

---

> ### Comment · Reviewer_wLXq · 2025-08-04
>
> I thank the authors for their response. I also have read the other reviewers comments, and I believe the paper has clear and interesting results that is of particular interest to the community. Hence, I raise my score and vote for acceptance.

---

> > ### Author Response · Authors · 2025-08-05
> > **Thank you**
> >
> > Thank you for the thoughtful review and the positive reassessment. We’re glad our responses addressed your concerns, and we will incorporate the clarifications into the revised manuscript.

---

### Official Review · Reviewer_yYUJ · 2025-07-03

**Clarity:** 3
**Significance:** 3
**Originality:** 3
**Rating:** 5
**Confidence:** 3

**Summary:**

This work introduces Conformal Risk Training, an extension of the traditional Conformal Risk Control framework to a broader class of risk measures, including CVaR, entropic risk etc. Their end-to-end approach integrates risk control directly into the training process by differentiating through the risk-controlling decision rule. The authors provide theoretical guarantees for risk control by applying monotone transformations of the loss function and develop techniques for computing gradients in common bi-level optimization settings.

**Questions:**

1. I think a discussion on the tightness of the CORC bounds might be helpful. Especially, under what conditions might these guarantees become overly conservative?
2. what happens when the loss functions $L_i(\lambda)$ is not globally monotonic? For example, could one consider piecewise monotonic, quasimonotonic, or approximate monotonic losses and still retain valid risk control?
3. While differentiability of the inner solution $\lambda(\theta)$ is addressed under specific conditions, can the authors formally analyze its stability with respect to perturbations in $\theta$?
4. Can the CORC framework be extended to non exchangeable data, such as time series settings?
5. I believe this work has strong connections to Robust Optimization, especially with the recently evolving end to end robust optimization methods. I think a discussion along this direction can help connect it to a broader scope of works.
6. Can the authors provide insight into when end-to-end conformal risk training yields better efficiency or generalization compared to post-hoc CRC? Are there conditions under which the added complexity of bilevel training is provably beneficial?

**Ethical Concerns:**

["NO or VERY MINOR ethics concerns only"]

**Final Justification:**

I am maintaining my score. The authors addressed the main concerns, and while some broader connections and extensions remain open, that does not diminish the contributions of the paper. Overall, this is a solid and timely contribution to risk aware learning with strong theoretical and empirical results.

**Limitations:**

as discussed above

**Paper Formatting Concerns:**

1. Algorithm 2 can be a bit more clear.
2. Figures lack clarity, could benefit from using white background, adding legend at the bottom instead.

**Quality:**

3

**Strengths And Weaknesses:**

1. The paper extends CRC beyond expected loss to risk measures like CVaR addressing an important open question. Introduces a novel bi-level optimization framework that differentiates through the conformal risk control procedure, enabling joint learning of model parameters and risk-sensitive thresholds.
2. The authors provides finite sample, distribution free risk guarantees for a broad class of risks under monotonicity assumptions. They derive closed form gradient expressions under common conditions.
3. The framework appears broadly applicable to any bounded, monotone loss, which makes it suitable for a wide range of decision making tasks beyond the two domains tested.
4. Theoretical guarantees in the paper seem to depend on the assumption that loss functions are monotone and exchangeable. These assumptions may not hold in many practical settings, especially in tasks involving structured or high-dimensional outputs.
4. The framework introduces an extra hyper parameter t, whose selection requires a hold out set and may impact practical performance if not tuned well.
5.  Solving nested optimization problems (especially with CVaR constraints) can be computationally expensive or sensitive to initialization.
6. While baseline comparisons are made, no direct comparison is made with other non conformal risk sensitive training approaches (e.g., DRO, robust optimization).
7. The manuscript doesn't make connections with the rapidly growing literature on decision focused learning and contextual stochastic optimization. in particular, it omits discussion of the recent EJOR survey by Sadana et al. on contextual optimization and the JAIR survey by Mandi et al. on Decision Focused Learning, both of which place bilevel risk-aware training in a broader taxonomy.

---

> ### Author Rebuttal · Authors · 2025-07-31
>
> We thank the reviewer for recognizing Conformal Risk Training as a promising method for controlling risk by design. We respond to each of the points raised by the reviewer.
>
> **Monotone assumption**
>
> The reviewer correctly points out that in some real-world risk control settings, the monotonicity assumption does not apply, especially in tasks involving structured or high-dimensional outputs. For example, the Learn then Test (Angelopoulos et al., 2025) paper gives multiple examples of non-monotone losses and develops a method for controlling their expected loss with high probability. However, as we (and many other papers in the conformal literature) demonstrate, the monotonicity assumption nonetheless encompasses a large number of real-world loss functions.
>
> Unfortunately, relaxing the global monotonicity assumption on the loss functions $L_i$ remains challenging. As shown by the original Conformal Risk Control (CRC) paper (see "Proof of Proposition 2" in their Appendix E), even functions $L_i$ that are _monotone in expectation_ cannot be controlled using Conformal Risk Control methods. For a loss $L$ that is approximately monotone, the original CRC paper proposes the idea of "monotonizing non-monotone risks"—_i.e._ computing the tightest monotone upper-bound of $L$. This idea can be directly translated to our Conformal OCE Risk Control (CORC) method, though we do not explore this idea further.
>
> **Exchangeability assumption**
>
> While our work uses the standard assumption that the loss functions $L_i$ are exchangeable, this assumption can be relaxed following standard conformal prediction arguments. For example, the CRC paper demonstrates how to apply CRC under covariate shift by using importance weighting, and it is straightforward to adapt that methodology to our CORC method. We can include a discussion of this in our camera-ready paper if the reviewer would like.
>
> Specifically for time series, split conformal methods such as what we use are generally not directly applicable. The online variants of conformal methods (e.g., adaptive conformal inference (Gibbs and Candes, 2021)) generally rely on very different proof techniques compared to the exchangeable data setting. Thus, we believe that developing an online variant of CORC would be a valuable direction of future work.
>
> **Sensitivity of hyperparameter $t$, and tightness of CORC bound**
>
> The reviewer keenly points out the importance of tuning the hyperparameter $t$ to tighten the OCE risk bound. As seen in the Proof of Theorem 1 (Appendix C.2), $t$ shows up in two inequalities:
>
> $$
> R[L_{N+1}(\hat\lambda)]
> \leq \mathbb{E}\left[ t + \phi(L_{N+1}(\hat\lambda) - t) \right]
> = \mathbb{E}[\tilde{L}_{N+1,t}(\hat\lambda)]
> \leq \alpha.
> $$
>
> The first inequality is tight at the value $t$ that minimizes the variational OCE expression. For example, when $R = \text{CVaR}^\delta$, $t=\text{VaR}^\delta[L(\hat\lambda)]$ makes the first inequality tight. On the other hand, theoretically analyzing the best choice of $t$ to tighten the 2nd inequality is difficult in general. We welcome suggestions, but for now, we leave this to future work.
>
> In the 2 tables below, we show the empirical sensitivity of the average task loss and the empirical CVaR on the choice of $t$ for post-hoc Conformal CVaR Control in the battery storage task. Here, $\Delta t$ is the (absolute) amount by which we change $t$ from the value tuned on the training set. Evidently, the value of $t$ tuned on the training set is not always optimal, but the resulting task loss tends to be close (within 1 stddev) of the task loss from the optimal $t$.
>
> Due to space limitations, we can only show a few choices of $\Delta t$, but we will include a proper figure in the camera-ready.
>
> _Table 1_: Task loss
> | Δt | α=2, δ=0.9 | α=2, δ=0.95 | α=2, δ=0.99 | α=5, δ=0.9 | α=5, δ=0.95 | α=5, δ=0.99 | α=10, δ=0.9 | α=10, δ=0.95 | α=10, δ=0.99 |
> |-|-|-|-|-|-|-|-|-|-|
> | -5 | 0 ± 0 | 0 ± 0 | 0 ± 0 | 0 ± 0 | 0 ± 0 | 0 ± 0 | -8.4 ± 18 | 0 ± 0 | -2 ± 3.2 |
> | -1 | 0 ± 0 | 0 ± 0 | -0.39 ± 0.64 | 0 ± 0 | -2.9 ± 6.1 | -3.7 ± 1.7 | -8.4 ± 18 | -6.2 ± 13 | -8.4 ± 3 |
> | 0 | 0 ± 0 | -1.5 ± 2.7 | -1.8 ± 0.51 | -4.2 ± 13 | -3.6 ± 6.7 | -4.5 ± 1.3 | -13 ± 20 | -7.3 ± 13 | -9 ± 2.6 |
> | 1 | 0 ± 0 | -2.2 ± 3 | -0.95 ± 0.7 | -8.4 ± 18 | -5.3 ± 7.3 | -4.5 ± 0.76 | -13 ± 20 | -10 ± 14 | -9.3 ± 2.1 |
> | 5 | -1.9 ± 6 | -0.69 ± 1.5 | 0 ± 0 | -8.4 ± 18 | -5.1 ± 5.9 | 0 ± 0 | -13 ± 20 | -13 ± 15 | -4.8 ± 3.5 |
>
> _Table 2_: CVaR
> | Δt | α=2, δ=0.9 | α=2, δ=0.95 | α=2, δ=0.99 | α=5, δ=0.9 | α=5, δ=0.95 | α=5, δ=0.99 | α=10, δ=0.9 | α=10, δ=0.95 | α=10, δ=0.99 |
> |-|-|-|-|-|-|-|-|-|-|
> | -5 | 0 ± 0 | 0 ± 0 | 0 ± 0 | 0 ± 0 | 0 ± 0 | 0 ± 0 | 1 ± 2.3 | 0 ± 0 | 1.3 ± 2.1 |
> | -1 | 0 ± 0 | 0 ± 0 | 0.27 ± 0.43 | 0 ± 0 | 0.87 ± 1.8 | 2.7 ± 1.1 | 1 ± 2.3 | 1.9 ± 4 | 6.1 ± 1.6 |
> | 0 | 0 ± 0 | 0.46 ± 0.81 | 1.3 ± 0.24 | 0.39 ± 1.2 | 1.1 ± 2 | 3.3 ± 0.59 | 1.5 ± 2.5 | 2.3 ± 4.1 | 6.7 ± 1.2 |
> | 1 | 0 ± 0 | 0.68 ± 0.9 | 0.77 ± 0.61 | 1 ± 2.3 | 1.6 ± 2.2 | 3.4 ± 0.53 | 1.5 ± 2.5 | 3.2 ± 4.4 | 6.9 ± 1 |
> | 5 | 0.17 ± 0.53 | 0.31 ± 0.68 | 0 ± 0 | 1 ± 2.3 | 1.7 ± 1.9 | 0 ± 0 | 1.5 ± 2.5 | 4.2 ± 4.6 | 3.8 ± 3 |
>
>
> **Nested optimization**
>
> We believe the reviewer may have a misunderstanding of our method. We never directly solve any optimization problem that involves a CVaR constraint. Instead, our method replaces the CVaR constraint with the sufficient criterion from Theorem 2.
>
> For Conformal Risk Training, while it is true that we have to calculate $\lambda(\theta)$ at every minibatch, we note that this computation is generally very efficient via the bisection-search method shown in Algorithm 1.
>
> **Comparisons with other risk-sensitive training approaches (e.g., DRO, robust optimization)**
>
> We were indeed inspired by other risk-sensitive training approaches, especially the conditional robust optimization (CRO) methods by Yeh et al. (2024) and Chenreddy et al. (2024). However, our Conformal CVaR Risk Control method fundamentally solves a different problem than robust optimization. Whereas end-to-end robust optimization (RO) aims to learn the _easiest_, say, 90%-confidence uncertainty set with respect to which to make decisions, our Conformal CVaR Risk Control method aims to control risk for the _hardest_ 10% of outcomes. In other words, RO and DRO methods aim to be robust to the most likely outcomes, whereas our approach focuses on tail risks which encompass the most unlikely, and costly, outcomes. Without strong distributional assumptions on the loss function, RO and DRO methods do not provide any tail risk guarantees.
>
> **Literature connections to DFL and Contextual Stochastic Optimization**
>
> Our work is very much inspired by the broader decision-focused learning (DFL) literature. Due to the page limit in our submission, we only cited DFL methods that used conformal prediction for uncertainty quantification (e.g., Yeh et al. 2024, and He et al. 2025), and we did not have space in our submission to discuss connections with the broader DFL literature. In the camera-ready revision, we will definitely discuss and cite the Sadana and Mandi survey papers.
>
> **Stability of gradient**
>
> Here, we briefly discuss the stability of the gradient for the two cases described in Theorem 3. For case (i), where $B$ and $L_i$ are piecewise constant in $\lambda$, the exact gradient may exhibit high variance. Thus, as we note in Appendix D.1, for the image segmentation example we average the gradient over pixels with similar model outputs. This variance reduction technique was first discussed by Hong 2009 and Noorani et al. (2024) for the simpler conformal prediction settting where the risk parameter $\lambda$ is chosen as an empirical quantile. In contrast, for the CORC setting (whether Theorem 3 (i) or (ii)), the risk parameter $\lambda$ is chosen as the solution to an optimization objective. Analyzing stability then requires understanding the sensitivity of the solution of the optimization problem to its parameters. Characterizing this stability is therefore challenging, and we agree this is a valuable direction for additional research.
>
> **Efficiency and generalization of Conformal Risk Training vs. post-hoc CRC**
>
> Like the original Conformal Training method, the benefit of Conformal Risk Training is largely empirical. In general, we would expect Conformal Risk Training to yield larger improvements when the Conformal Risk Training objective $\ell(\theta, \lambda)$ (e.g., specificity) is substantially different from the model pretraining objective (e.g., cross-entropy loss).
>
> **Formatting concerns**
>
> We apologize that the reviewer finds Algorithm 2 unclear. If the reviewer could please elaborate on which specific aspect of Algorithm 2 should be improved, we are happy to make changes. As for the figures, we will take the reviewer's suggestion into account when preparing camera-ready figures.

---

> > ### Comment · Reviewer_yYUJ · 2025-08-06
> >
> > Thank you for the detailed response. I appreciate the authors clarifications and empirical evidence addressing the concerns. The explanations on monotonicity, exchangeability, and the role of the hyperparameter $t$ were helpful. Having read the other discussions as well, I am satisfied with the responses and will maintain my score. I continue to believe this is a valuable and impactful contribution to the literature on risk aware training.

---

### Official Review · Reviewer_UbDa · 2025-07-05

**Clarity:** 3
**Significance:** 4
**Originality:** 3
**Rating:** 4
**Confidence:** 4

**Summary:**

The paper presents two primary contributions. First, it extends the Conformal Risk Control (CRC) framework beyond expected loss to a broader class of Optimized Certainty-Equivalent (OCE) risks, which includes tail-risk measures like Conditional Value-at-Risk (CVaR). This represents a  leap forward, as the OCE class includes common tail-risk measures like the CVaR. This work provides the community with a practical methodology for creating models with provable guarantees on worst-case scenarios, moving beyond simple averages. The core technical insight is demonstrating that any OCE risk can be bounded via a monotone transformation of the loss, thereby preserving the essential monotonicity property that underpins the conformal guarantee. Proposition 1 prepares the extension for OCE risks, and improves more generally previous CRC by considering a lambda dependent bound. The main result is then theorem 2 that considers CVaR among OCE, and it proposes algorithm 1 for building the set of feasible lambda (controlling the risk). It discusses briefly the feasibility of the risk control, as well as the role of t. It introduces an assumption 5 where L_i’s and B can be either increasing or decreasing.

Second, it introduces Conformal Risk Training (CRT), an end-to-end training methodology that embeds the risk control procedure differentiably into the model's fitting loop, and generalizes previous attempts in conformal training who inspired partly the paper. The stated goal is to achieve provable risk guarantees while mitigating the degradation in performance often seen with post-hoc calibration methods (say the choice of lambda to get the guarantee on the False Positive Rate implies a drop on the False Negative Rate). Consequently the an end-to-end optimization method propose an interesting way to deal with the inherent tension between risk control and model performance. The model is optimized not only for accuracy but also for its "amenability" to risk control, by using Lagrangian relaxation, where the constraint on the risk is added in the global criterion. It learns to produce predictions that can satisfy the risk constraint with minimal performance degradation. Theorem 3 gives conditions for that specific approach for selecting the proper risk; the formal version is given in appendix where parametric families of risk and bound are given (piece constant in lambda), in order to be able to compute the gradient.

A short Empirical Validation & Key Results are given:  for  validation the CRT framework's efficacy through two real-world applications that highlight the  dual benefits of safety and performance.
* Controlling False Negative Rate in Tumor Segmentation: CRT is used to control the False Negative Rate (FNR) at a user-specified level α. Standard post-hoc CRC also achieves this safety guarantee, by incurring a high False Positive Rate (FPR) (such as 80%). The CRT model, in contrast, successfully resolves this trade-off, by reducing the FPR by reducing significantly and  maintaining the FNR guarantee at different risk levels.
* Controlling CVaR in Battery Storage Operation: In a financial task where an agent optimizes battery charging cycles based on price forecasts, CRT is used to control the CVaR of monetary losses. CRT can successfully bounds this tail risk at the target level and it also yields a higher average profit compared to the post-hoc CRC baseline. This result showcases its ability to manage downside risk while actively improving upside performance.

**Questions:**

Additional Questions to address :
1. In addition to the questions asked in the S&W, I would to have a clearer explanation of the computation of the gradient in the example 2, and the importance of the assumption of having all values f_theta(X_i)_j  to be unique. What happens if not the case ? And If they are very close? How does it influence the quality or existence of the derivative?
2. It is said in assumption that we can add increasing or decreasing losses: is it a generalization? Is it really useful?
3. What about using the standard cross entropy loss in image segmentation problem,  instead of specificity minimizing specificity.
4. Can you provide sensitivity analysis on t, and give a complete methodology for doing the estimation, taking into account all the datasets needed train, final calibration (intermediate split)… to see how much a standard training (to see how the standard split train test validation is changed).
5. Compare with other approaches that give a guarantee on the loss (like probabilistic for instance).

**Ethical Concerns:**

["NO or VERY MINOR ethics concerns only"]

**Limitations:**

Partly. I really appreciate the direction the paper proposed, and the technics used that have nice theoretical properties and guarantees.
Nevertheless, I think that the paper does not face with enough depth and rigor the limitations or real impact of the paper : comparison with LTT, RCPS or other risk control approach in the paper; the choice of hyperparameter t, or the importance of the quality of B, the feasiblity problem, the feasibility of the computation of the gradient, and the stability / computational aspects of the gradient.
I am ready to increase my score if the authors provide a more rigorous, organised and deeper treatment of these points.

**Quality:**

3

**Strengths And Weaknesses:**

Strengths:

The paper is well written and opens the way to the very promising direction of building models that offer a finite horizon controlled of the risk by design. Indeed, while standard Conformal Prediction comes into play only for evaluating or correcting for the uncertainty of a model, hence it could be an important enabler of reliable and trustworthy AI.
The two examples presented and discussed in the paper motivate well the development of Conformal Training by their important and the practical impact it can have.
The technics and proofs seem ok, and the approach for computing the gradient seems efficient in some cases, with an interesting and promising direction based on bilevel optimization, although numerous theoretical and practical problems could happen, and be discussed.

Weaknesses and critical analysis
1. Real Impact of OCE and Novelty of CVaR Control somehow Overstated.
The entire framework, both for OCE risk control and CRT, is built upon the foundational assumption of a monotone loss function L(lambda) in the hyperparameter lambda. While a relaxation is noted for OCE and CVaR under specific conditions, this does not resolve the core limitation for the general framework. While CVaR or other OCE may allow to consider new applications, I wonder if it is really a big extension for dealing nonmonotonous loss function. Other approach such as Learn Then Test, Angelopoulos et al, AoAS 2025  or RCPS Angelopoulos et al could be used. In addition, the control of CVaR and other distortion measures is also addressed in Chen et al, cited in the paper. While the paper does extend traditional CRC to CVaR and potential othe risks,  this contribution is rather incremental and not the only way of doing it. A better discussion on the various possibilities of controlling of CVaR would be useful to understand the impact of the contribution. As well as the ability to address the distortion risk measures. Would it be possible to claim theorem 2 for general OCE risk instead of only for CVaR, is there any reason that precludes a general formulation and adding a corollary for CVaR.
2. The OCE Formulation Introduces an Unanalyzed Complexity with the new hyperparameter t from the variational definition of the risk measure. t to be selected without access to the calibration data. The paper's proposed solution—tuning t on a separate held-out set—is a practical heuristic but lacks theoretical grounding, as there is no analysis of the method's sensitivity to this choice. How does a sub-optimal t affect the tightness of the risk bound and, consequently, the model's ultimate performance? This introduces a new, unanalyzed layer of complexity to the calibration process.

3. As a side and related effect is the feasibility of the constraint. It seems that there is an interplay between t, the constraint alpha, and the bound used B(lambda), as we can see in theorem 2, or in theorem 3. It is a theoretically and practically important point that deserves an explicit and detailed treatment discussion in the paper. Could you discuss in a coherent way the problem of existence, and possibly how it is related to the existence of a gradient.

4. The Practicality of Conformal Risk Training is Limited by Gradient Computation.
The CRT method's central mechanism is to make use of the implicit value theorem by considering a bilevel optimisation, and propagate through the risk-control parameter lambda(theta). The paper shows this is feasible for highly-constrained settings with Piecewise Constant Losses or strict convexity / monotony : The gradient exists because the optimal lambda(theta) snaps to a specific model output value.
In addition every batch must be split in a calibration and prediction, but the size and the role of the split is not detailed and it becomes a bit blurry. The complexity and subtleties of its computation are hidden, as in appendix problem of stabilities and variance reduction p. 27 line 1043 do require averaging. Could you dedicate a specific part to the existence of the gradient and the numerical computation of the gradient. The ability to compute or approximate properly the gradient narrows the the applicability of CRT.

---

> ### Author Rebuttal · Authors · 2025-07-31
>
> We thank the reviewer for recognizing Conformal Risk Training as a promising method for controlling risk by design. We respond to each point raised by the reviewer.
>
> **Comparison against related methods**
>
> The review highlights 3 related works on risk control: Learn then Test (LTT), Risk Controlling Prediction Sets (RCPS), and the Distortion Risk Control method by Chen et al. (henceforth referred to as "Chen"). All 3 methods provide *high-probability* bounds on risk, as opposed to the stronger *certain (marginal)* bound that our Conformal OCE Risk Control (CORC) method achieves. As shown in Appendix B of the original Conformal Risk Control (CRC) paper, the marginal bounds from conformal methods are significantly more statistically efficient compared to LTT/RCPS.
>
> We further discuss each of these methods in order:
> - LTT: LTT is primarily used to control the expected loss of non-monotone loss functions. While the generic LTT framework in principle supports a general risk measure $R$, one must be able to compute p-values for null hypotheses of the form $H_\lambda:\ R(\lambda)>\alpha$. The Hoeffding–Bentkus inequality p-values described in the LTT paper only apply to expected losses; while a recent work (“Quantile Learn then Test”, Farzaneh et al. 2024) proposes a p-value for $R=\text{VaR}$ (value-at-risk), to the best of our knowledge, we are unaware of p-values for $R=\text{CVaR}$. We note that a conservative approach would be to use LTT to control the expectation of $\tilde{L}_t(\lambda)=t+\phi_{\text{CVaR}^\delta}(L(\lambda)-t)$, which upper-bounds $\text{CVaR}^\delta[L(\lambda)]$. If the reviewer would like to see a comparison between CORC and this approach to LTT, we can try to implement this comparison.
> - RCPS: RCPS is used to control risk resulting from set-valued predictors. For our battery storage CVaR example, it is unclear to us how to formulate the problem using set-valued predictors. Furthermore, RCPS requires the loss functions to be monotone in $\lambda$ (with monotonicity in the same direction). Thus, we do not believe RCPS to be suitable for controlling CVaR for our battery storage example, where the loss functions can be either monotone increasing or decreasing.
> - Chen: The main result from the Chen method is an *asymptotic* (as opposed to finite-sample) high-probability distortion risk control bound. The Chen paper does introduce a finite-sample variant, which is more conservative. If the reviewer would like to see a comparison between CORC and the more conservative Chen method, we can try to implement this comparison.
>
> We will include a more thorough discussion of these three related works in the camera-ready revision.
>
> **Generalizing Theorem 2 to all OCE risk measures**
>
> Theorem 2 does not apply more generally to all OCE risk measures. For example, the expected loss is an OCE risk measure. As the original CRC paper makes clear, monotonicity of the loss function (where all loss functions are monotone in the same direction) is necessary for controlling the expected loss.
>
> **Sensitivity of hyperparameter $t$**
>
> Tuning the hyperparameter $t$ to tighten the OCE risk bound is indeed important. In the Proof of Theorem 1 (Appendix C.2), $t$ shows up in two inequalities:
>
> $$
> R[L_{N+1}(\hat\lambda)]
> \leq\mathbb{E}\left[t+\phi(L_{N+1}(\hat\lambda)-t)\right]
> =\mathbb{E}[\tilde{L}_{N+1,t}(\hat\lambda)]
> \leq \alpha.
> $$
>
> The 1st inequality is tight at the value $t$ that minimizes the variational OCE expression. For example, when $R=\text{CVaR}^\delta$, $t=\text{VaR}^\delta[L(\hat\lambda)]$ makes the 1st inequality tight. On the other hand, theoretically analyzing the best choice of $t$ to tighten the 2nd inequality is difficult in general. We welcome suggestions, but for now, we leave this to future work.
>
> In the 2 tables below, we show empirical sensitivity of the average task loss and empirical CVaR to the choice of $t$ for post-hoc Conformal CVaR Control in the battery storage task. Here, $\Delta t$ is the (absolute) amount by which we change $t$ from the value tuned on the training set. Evidently, the value of $t$ tuned on the training set is not always optimal, but the resulting task loss tends to be close (within 1 stddev) of the task loss from the optimal $t$.
>
> Due to space limitations, we can only show a few choices of $\Delta t$, but we will include a proper figure in the camera-ready.
>
> Task loss
> | Δt | α=2, δ=.9 | α=2, δ=.95 | α=2, δ=.99 | α=5, δ=.9 | α=5, δ=.95 | α=5, δ=.99 | α=10, δ=.9 | α=10, δ=.95 | α=10, δ=.99 |
> |-|-|-|-|-|-|-|-|-|-|
> | -5 | 0±0 | 0±0 | 0±0 | 0±0 | 0±0 | 0±0 | -8.4±18 | 0±0 | -2±3.2 |
> | -1 | 0±0 | 0±0 | -0.39±0.64 | 0±0 | -2.9±6.1 | -3.7±1.7 | -8.4±18 | -6.2±13 | -8.4±3 |
> | 0 | 0±0 | -1.5±2.7 | -1.8±0.51 | -4.2±13 | -3.6±6.7 | -4.5±1.3 | -13±20 | -7.3±13 | -9±2.6 |
> | 1 | 0±0 | -2.2±3 | -0.95±0.7 | -8.4±18 | -5.3±7.3 | -4.5±0.76 | -13±20 | -10±14 | -9.3±2.1 |
> | 5 | -1.9±6 | -0.69±1.5 | 0±0 | -8.4±18 | -5.1±5.9 | 0±0 | -13±20 | -13±15 | -4.8±3.5 |
>
> CVaR
> | Δt | α=2, δ=.9 | α=2, δ=.95 | α=2, δ=.99 | α=5, δ=.9 | α=5, δ=.95 | α=5, δ=.99 | α=10, δ=.9 | α=10, δ=.95 | α=10, δ=.99 |
> |-|-|-|-|-|-|-|-|-|-|
> | -5 | 0±0 | 0±0 | 0±0 | 0±0 | 0±0 | 0±0 | 1±2.3 | 0±0 | 1.3±2.1 |
> | -1 | 0±0 | 0±0 | 0.27±0.43 | 0±0 | 0.87±1.8 | 2.7±1.1 | 1±2.3 | 1.9±4 | 6.1±1.6 |
> | 0 | 0±0 | 0.46±0.81 | 1.3±0.24 | 0.39±1.2 | 1.1±2 | 3.3±0.59 | 1.5±2.5 | 2.3±4.1 | 6.7±1.2 |
> | 1 | 0±0 | 0.68±0.9 | 0.77±0.61 | 1±2.3 | 1.6±2.2 | 3.4±0.53 | 1.5±2.5 | 3.2±4.4 | 6.9±1 |
> | 5 | 0.17±0.53 | 0.31±0.68 | 0±0 | 1±2.3 | 1.7±1.9 | 0±0 | 1.5±2.5 | 4.2±4.6 | 3.8±3 |
>
> Regarding the interplay among $t$, $\alpha$, $B$, it is true that a bad choice of $t$ may lead to $\hat\Lambda\_t$ being empty. As noted in Thm 1, if $\hat\Lambda\_t$ is empty (so $\sup\hat\Lambda\_t=-\infty$), we set $\hat\lambda=\lambda\_{\min}$.
>
> **Gradient computation**
>
> First, we would like to emphasize that the post-hoc Conformal OCE Risk Control method we introduce does not require any gradient computation. We acknowledge that Conformal Risk Training is only applicable when the gradient $\frac{d\lambda}{d\theta}$ can be computed (or approximated), and Theorem 3 is our best attempt to obtain sufficient conditions for the existence of the gradient.
>
> Regarding the size and role of batch splitting, we recommend that the size of the batch $N$ be large enough such that $\tilde{h}\_t(\lambda\_{\min})\leq\alpha$, so that the gradient $\frac{d\lambda}{d\theta}$ is non-zero. If the batch size is smaller, the model will still train with gradients $\frac{\partial\ell}{\partial\theta}(\theta,\lambda\_{\min})$, but this may result in reduced training efficiency. We will make a note of this in the camera-ready.
>
> Regarding the assumption that all values $f_\theta(X_i)_j$ are unique in Example 2, this assumption is necessary for an *exact* gradient to exist. This is similar to how the exact gradient for a max-pooling function $g(\mathbf{x})=\max_i x_i$ (e.g., in convolutional neural nets) only exists when the entries of the vector $\mathbf{x}$ are unique. When there are multiplicities, the exact gradient is technically undefined.
>
> When there are multiplicities, or when there are multiple model outputs with very similar values, the gradient sensitivity can be large. This is the motivation and intuition behind our choice to average the model's gradients over the $M$ pixels with the most similar output values (see Appendix D.1). This averaging reduces variance in the gradient estimation, as noted by Hong (2009) and Noorani et al. (2025). We would be happy to move this discussion to the main text in the camera-ready revision.
>
> **Standard cross entropy loss in image segmentation**
>
> Our understanding of the reviewer's question is, what if we set the training objective $\ell(\theta,\lambda)$ to be the cross-entropy (CE) loss? We claim that this is equivalent to simply training the model with CE loss (without conformal risk training), and then applying post-hoc CORC.
>
> Because CE loss does not depend on any threshold $\lambda$, $\ell(\theta,\lambda)=\ell(\theta)$. Thus, the risk constraint can be dropped, since picking $\lambda=\lambda_{\min}$ will suffice (by Assumption 4). That is,
>
> $$
> \min_{\theta,\lambda}\mathbb{E}[\ell(\theta)]\text{ s.t. }R(\theta,\lambda)\leq\alpha
> \quad\equiv\quad
> \min_\theta\mathbb{E}[\ell(\theta)].
> $$
>
> We can make a note of this in our camera-ready revision to highlight that Conformal Risk Training generalizes empirical risk minimization (ERM) to the setting where the objective $\ell$ depends on a risk control parameter $\lambda$ that is constrained to satisfy a risk constraint. When $\ell$ does not depend on $\lambda$, then Conformal Risk Training reduces to standard ERM.
>
> As shown in Figure 1, further training the pre-trained image segmentation model with CE loss and then applying post-hoc CRC (shown in orange) does not improve model performance over directly applying post-hoc CRC to the pre-trained model (shown in blue).
>
> To further emphasize how Conformal Risk Training (CRT) with the specificity (FPR) objective differs from ERM with CE loss, the following table shows the average CE loss on the test set for 3 different training settings (mean±std over 10 random seeds). Notably, at a given risk level $\alpha$, CRT achieves lower FPR but incurs higher CE loss. This highlights the decision-focused nature of CRT.
>
> | | pretrain | cross-entropy | CRT α=.01 | CRT α=.05 | CRT α=.10 |
> |-|-|-|-|-|-|
> | CE loss* | 2.8 ± .1 | 2.9 ± .2 | 11.9 ± .4 | 5.8 ± 2.5 | 7.8 ± 6.1 |
> | FNR @ α=.01 | .009 ± .003 | .008 ± .003 | .008 ± .005 |-|-|
> | FPR @ α=.01 | .841 ± .014 | .811 ± .049 | .465 ± .057 |-|-|
> | FNR @ α=.05 | .048 ± .008 | .050 ± .014 |-| .049 ± .014 |-|
> | FPR @ α=.05 | .524 ± .022 | .580 ± .106 |-| .385 ± .054 |-|
> | FNR @ α=.1 | .101 ± .019 | .101 ± .017 |-|-| .106 ± .021 |
> | FPR @ α=.1 | .256 ± .035 | .293 ± .092 |-|-| .152 ± .041 |
>
> **In this table, "CE loss" refers to the weighted cross-entropy loss described in the PraNet (Fan et al., 2020) paper.*

---

> > ### Comment · Reviewer_UbDa · 2025-08-06
> >
> > - What do you mean by significantly more statistically efficient compared to LTT/RCPS ?The control is of different nature…My point is that controlling non monotonous risk is not that new, and I wonder if the LTT or RCPS would be much more conservative or impractical . I appreciate the authors’ proposition to perform a comparison for CVaR and LTT, and also with « Chen ».
> >
> > - Could you explain why Theorem 2 only applies to CVaR ? Is there a crucial property of \phi that make it possible. I do not get the point from your answer. By the way, could you clarify the role of assumption 5 with either decreasing or increasing: how is it possible and is it a useful relaxing assumptions that can cover some interesting. Thanks for your precision.
> >
> > - The sensitivity analysis is interesting but is it possible to make relative change in order to appreciate the impact. But as we see, the impact can be quite high for high \alpha (5, 10). Is there any reason that would explain this increase.
> >
> >     - I appreciate the details given about the instability about the gradient computation. Following the example of $max(x_i)$ given by authors, the function is non-differentiable on the subsets where there is exists i,just s.t.$x_i=x_j$. On the complementary open set, the function is differentiable. Hence, would it be possible to add some mathematical rigor by specifying the open set where the function $\ell$ is differentiable. Possibly, it might be possible to claim that the function is differentable almost everywhere if the probabilities of ties f_t(Xi)_j is null. If the function is not differentiable is it possible to use sub gradient instead? Anyway, it would be great if the authors could give more rigorous details about the computation, and how the optimization is performed (by some SGD algorithm).

---

> > > ### Author Response · Authors · 2025-08-09
> > > **Comparison against LTT / Chen**
> > >
> > > As we mentioned in our previous comments, we are working on a comparison between our Conformal OCE Risk Control method and the LTT / Chen methods. Due to the limited amount of time, we unfortunately will not have time to share the comparison with the reviewer before the end of the author-reviewer discussion period. Thank you for your understanding and patience.
> > >
> > > We still aim to include the comparison in our camera-ready revision.

---

> ### Comment · Area_Chair_JXu6 · 2025-08-04
>
> Dear reviewer, please engage with the author rebuttal.

---

> ### Author Response · Authors · 2025-08-07
> **Response Part 1/3**
>
> We thank the reviewer for detailed and thougtful follow up questions. Due to character count limits, our response is split across 3 comments.
>
> **Statistical efficiency vs. LTT / RCPS**
>
> As noted in Appendix B of the original paper on Conformal Risk Control by Angelopoulos et al. (2023):
>
> > Conformal risk control is substantially more sample-efficient than RCPS/LTT. On the scale of the risk, RCPS/LTT are far more conservative, converging to $\alpha$ at a $1/\sqrt{n}$ rate, while conformal risk control converges at $\alpha$ at a $1/n$ rate.
>
> For example, RCPS for controlling expected loss with the Hoeffding inequality chooses $\lambda$ such that empirical risk on the calibration set is below $\alpha - \sqrt{\frac{1}{2n} \log\frac{1}{\delta}}$. Although there exist tighter inequalities than the Hoeffding bound, they tend to share the $O(1/\sqrt{n})$ rate.
>
> In contrast, CRC picks $\lambda$ such that the empirical risk on the calibration set is below $\alpha - \frac{B-\alpha}{n}$. Our Conformal OCE Risk Control method likewise inherits this $O(1/n)$ rate.
>
> We have started working on a comparison with LTT and/or the Chen method and will report results when they are available.

---

> ### Author Response · Authors · 2025-08-07
> **Response Part 2/3**
>
> **Sensitivity Analysis on $t$**
>
> Following the reviewer's suggestion, we implemented sensitivity analysis with a relative change in $t$. In doing so, we noticed a bug that affected our sensitivity analysis for absolute changes in $t$. Let $t_0$ denote the optimal value of $t$ tuned on the training set. In Theorem 2, our CVaR risk guarantee only holds when $t \geq B(\lambda_{\min})$, but we forgot to impose this constraint on $t_0 + \Delta t$. We fixed the bug, and the tables below report sensitivity analyses for both absolute and relative changes in $t$. The cases where $t_0 + \Delta t < 0$ are reported as NaN. (In the battery storage problem, $B(\lambda_{\min}) = 0$.)
>
> The following table shows the values of $t_0$ (mean ± 1 std) selected by tuning on the training set:
>
> | | α=2, δ=.9 | α=2, δ=.95 | α=2, δ=.99 | α=5, δ=.9 | α=5, δ=.95 | α=5, δ=.99 | α=10, δ=.9 | α=10, δ=.95 | α=10, δ=.99 |
> |-|-|-|-|-|-|-|-|-|-|
> | t | 0.0002 ± 0.00064 | 5.4e-10 ± 1.2e-09 | 0.81 ± 0.29 | 0.043 ± 0.03 | 0.00035 ± 0.0011 | 2 ± 0.73 | 0.15 ± 0.077 | 0.094 ± 0.07 | 4.1 ± 1.5 |
>
> _Results for absolute $\Delta t$, _i.e._, $t = t_0 + \Delta t$_:
>
> Task loss
>
> | Δt (absolute) | α=2, δ=.9 | α=2, δ=.95 | α=2, δ=.99 | α=5, δ=.9 | α=5, δ=.95 | α=5, δ=.99 | α=10, δ=.9 | α=10, δ=.95 | α=10, δ=.99 |
> |-|-|-|-|-|-|-|-|-|-|
> | -5 | nan | nan | nan | nan | nan | nan | nan | nan | -6.5 ± 0.92 |
> | -1 | nan | nan | -1.3 ± 0.18 | nan | nan | -4.1 ± 1.2 | nan | nan | -9.1 ± 2.3 |
> | 0 | -8.6 ± 3.2 | -4.3 ± 1.6 | -1.8 ± 0.51 | -21 ± 7.9 | -11 ± 4 | -4.5 ± 1.3 | -36 ± 6.3 | -21 ± 7.9 | -9 ± 2.6 |
> | 1 | -6.8 ± 1.9 | -4.4 ± 1.1 | -0.95 ± 0.7 | -20 ± 6.7 | -11 ± 3.6 | -4.5 ± 0.76 | -35 ± 6.3 | -22 ± 7.5 | -9.3 ± 2.1 |
> | 5 | 0 | 0 | 0 | -0.41 ± 0.87 | -1.9 ± 0.65 | 0 | -32 ± 7.2 | -22 ± 5.6 | -4.8 ± 3.5 |
>
> CVaR
>
> | Δt (absolute) | α=2, δ=.9 | α=2, δ=.95 | α=2, δ=.99 | α=5, δ=.9 | α=5, δ=.95 | α=5, δ=.99 | α=10, δ=.9 | α=10, δ=.95 | α=10, δ=.99 |
> |-|-|-|-|-|-|-|-|-|-|
> | -5 | nan | nan | nan | nan | nan | nan | nan | nan | 4.4 ± 0.25 |
> | -1 | nan | nan | 0.89 ± 0.05 | nan | nan | 3 ± 0.51 | nan | nan | 6.6 ± 0.7 |
> | 0 | 1.6 ± 0.32 | 1.6 ± 0.3 | 1.3 ± 0.24 | 4 ± 0.78 | 4 ± 0.77 | 3.3 ± 0.59 | 7.3 ± 2.1 | 8.2 ± 1.5 | 6.7 ± 1.2 |
> | 1 | 1.3 ± 0.31 | 1.7 ± 0.21 | 0.77 ± 0.61 | 3.8 ± 0.78 | 4.3 ± 0.66 | 3.4 ± 0.53 | 7.3 ± 2.1 | 8.5 ± 1.4 | 6.9 ± 1 |
> | 5 | 0 | 0 | 0 | 0.061 ± 0.14 | 0.72 ± 0.31 | 0 | 6.5 ± 1.6 | 8.5 ± 1 | 3.8 ± 3 |
>
> _Results for relative $\Delta t$, _i.e._, $t = (1 + \Delta t) t_0$_:
>
> Task loss
>
> | Δt (relative) | α=2, δ=.9 | α=2, δ=.95 | α=2, δ=.99 | α=5, δ=.9 | α=5, δ=.95 | α=5, δ=.99 | α=10, δ=.9 | α=10, δ=.95 | α=10, δ=.99 |
> |-|-|-|-|-|-|-|-|-|-|
> | -0.5 | -8.6 ± 3.2 | -4.3 ± 1.6 | -1.5 ± 0.49 | -21 ± 8 | -11 ± 4 | -3.9 ± 1.2 | -36 ± 6.3 | -21 ± 7.9 | -7.7 ± 2.5 |
> | -0.1 | -8.6 ± 3.2 | -4.3 ± 1.6 | -1.8 ± 0.52 | -21 ± 7.9 | -11 ± 4 | -4.4 ± 2.3 | -36 ± 6.3 | -21 ± 7.9 | -8.9 ± 2.6 |
> | 0 | -8.6 ± 3.2 | -4.3 ± 1.6 | -1.8 ± 0.51 | -21 ± 7.9 | -11 ± 4 | -4.5 ± 1.3 | -36 ± 6.3 | -21 ± 7.9 | -9 ± 2.6 |
> | 0.1 | -8.6 ± 3.2 | -4.3 ± 1.6 | -1.8 ± 0.5 | -21 ± 7.9 | -11 ± 4 | -4.6 ± 1.2 | -36 ± 6.3 | -21 ± 7.9 | -9.1 ± 2.5 |
> | 0.5 | -8.6 ± 3.2 | -4.3 ± 1.6 | -1.7 ± 0.4 | -21 ± 7.9 | -11 ± 4 | -4.1 ± 1 | -36 ± 6.3 | -22 ± 7.9 | -8.3 ± 2 |
>
> CVaR
>
> | Δt (relative) | α=2, δ=.9 | α=2, δ=.95 | α=2, δ=.99 | α=5, δ=.9 | α=5, δ=.95 | α=5, δ=.99 | α=10, δ=.9 | α=10, δ=.95 | α=10, δ=.99 |
> |-|-|-|-|-|-|-|-|-|-|
> | -0.5 | 1.6 ± 0.32 | 1.6 ± 0.3 | 1.1 ± 0.22 | 4 ± 0.78 | 4 ± 0.77 | 2.8 ± 0.55 | 7.3 ± 2.1 | 8.2 ± 1.6 | 5.7 ± 1.1 |
> | -0.1 | 1.6 ± 0.32 | 1.6 ± 0.3 | 1.3 ± 0.24 | 4 ± 0.78 | 4 ± 0.77 | 3.3 ± 0.59 | 7.3 ± 2.1 | 8.2 ± 1.6 | 6.5 ± 1.2 |
> | 0 | 1.6 ± 0.32 | 1.6 ± 0.3 | 1.3 ± 0.24 | 4 ± 0.78 | 4 ± 0.77 | 3.3 ± 0.59 | 7.3 ± 2.1 | 8.2 ± 1.5 | 6.7 ± 1.2 |
> | 0.1 | 1.6 ± 0.32 | 1.6 ± 0.3 | 1.3 ± 0.23 | 4 ± 0.78 | 4 ± 0.77 | 3.4 ± 0.59 | 7.3 ± 2.1 | 8.2 ± 1.5 | 6.7 ± 1.2 |
> | 0.5 | 1.6 ± 0.32 | 1.6 ± 0.3 | 1.2 ± 0.29 | 4 ± 0.78 | 4 ± 0.77 | 3.1 ± 0.73 | 7.3 ± 2.1 | 8.2 ± 1.5 | 6.2 ± 1.5 |
>
> Especially from the relative $\Delta t$ tables, it is clear that the task loss and CVaR are not very sensitive to changes in $t$. In particular, these results show that the sensitivity to $t$ is _less_ than that suggested by the tables we presented in our initial rebuttal that did not include the needed constraint on $t$.

---

> ### Author Response · Authors · 2025-08-07
> **Response Part 3/3**
>
> **Theorem 2 and CVaR**
>
> While the result of Theorem 2 does not apply in general to all OCE risk measures, it can be extended to risk measures beyond CVaR. The property of CVaR that we use in the proof of Theorem 2 is the structure of its disutility function $\phi\_{\mathrm{CVaR}^\delta}(x) = \frac{1}{1-\delta}[x]\_+$, and specifically, the term $[x]\_+ = \max\\{x,0\\}$; this term, together with the choice of $t$, ensure that even when the original loss $L_i$ is monotone decreasing, the transformed loss $\tilde{L}\_{i, t}$ will be nondecreasing, enabling the application of CRC.
>
> As such, (a variant of) Theorem 2 applies to other choices of disutility function such as
>
> $$
> \phi(x) = \tilde\phi([x]_+),
> $$
>
> where $\tilde\phi: \mathbb{R} \to \mathbb{R}$ is any other disutility function. Since $[x]_+$ is convex and $\tilde\phi$ is closed convex nondecreasing, $\phi$ remains closed convex nondcreasing. Furthermore, the properties $\tilde\phi(0) = 0$ and $1 \in \tilde\phi(0)$ also imply that $\phi(0) = 0$ and $1 \in \phi(0)$. Therefore, $\phi$ is a valid disutility function.
>
> The transformed loss and upper bound
>
> $$
> \begin{aligned}
> \tilde{L}\_t(\lambda) &= t + \phi([L(\lambda) - t]\_+) \\\\
> \tilde{B}\_t(\lambda) &= t + \phi([B(\lambda) - t]\_+)
> \end{aligned}
> $$
>
> remain nondecreasing in $\lambda$ for $t \geq B(\lambda_{\min})$.
>
> We will be happy to clarify this potential generality of our result in a remark after the theorem.
>
> As for the role of Assumption 5, we'd like to emphasize that this relaxed assumption is what allows us to apply Theorem 2 to the battery storage example where the task loss function may be either monotone increasing or decreasing in $\lambda$ depending on the data point $(X,Y)$. We will highlight this point in our camera-ready revision.
>
> **Gradient computation**
>
> Our assumption that the $f(X_i)_j$ are unique can indeed be relaxed to the case where the probability of ties among $f(X_i)_j$ is null, in which case $\lambda(\theta)$ is differentiable almost everywhere.
>
> In this case, at the non-differentiable $\theta$, it is unclear whether a subgradient exists, because $f$ may not be convex.
>
> Also, could the reviewer please clarify what is meant by:
>
> > how the optimization is performed (by some SGD algorithm)
>
> Thank you for taking the time to read our response.

---

### Note · Authors · 2025-08-16

We would like to thank the reviewers again for their constructive feedback. In this set of final remarks, we summarize the main updates to our original submission:

1. **Analyzed sensitivity to hyperparameter $t$**: We provided new tables demonstrating the sensitivity of the CVaR risk control bound to the hyperparameter $t$. We provided sensitivity results for both absolute and relative changes in $t$, showing that in general, the risk control bound is not very sensitive to changes in $t$.

2. **Clarified gradient computation for conformal risk training**: We have significantly improved the clarity of Theorem 3, which provides sufficient conditions for the conformal risk training gradient to exist. Specifically, we addressed a minor proof mistake pointed out by Reviewer mggh; we clarified the necessity of unique model outputs (in response to Reviewer UbDa); we more clearly listed out our assumptions; and we more clearly prove how Theorem 3 applies to our actual neural network models.

3. **Analyzed sensitivity to calibration set size**: In response to reviewer A5AC, we added tables which show empirical sensitivity of our risk bound to the size $N$ of the calibration set. We also discussed with reviewer UbDa how our Conformal OCE Risk Control method is more sample efficient compared to RCPS/LTT approaches. Our risk bound converges to the target level $\alpha$ at a rate of $O(1/N)$, compared with the $O(1/\sqrt{n})$ rate of RCPS/LTT.

4. **Clarified how Conformal Risk Training generalizes empirical risk minimization (ERM)**: In response to reviewer UbDa, we clarified how using a standard cross-entropy loss with Conformal Risk Training reduces to the standard ERM procedure.

5. **Clarified the importance of Theorem 2**: Theorem 2 allows the application of Conformal CVaR Control to a random loss function that is always monotone, but whose direction of monotonicity may be random. As we explained to reviewer UbDa, this is precisely the setting of our battery storage control example. Furthermore, in discussion with reviewer UbDa, we showed how Theorem 2 can indeed be generalized to a somewhat broader class of risk measures beyond CVaR.

In conclusion, we are grateful to the reviewers for recognizing the importance of our contribution to the uncertainty quantification and risk control literature to enable more reliable deployment of AI in high-stakes settings.

---

### Decision · Program_Chairs · 2025-09-17

**Decision:**

Accept (poster)

**Comment:**

This paper makes two contributions: (i) extending CRC to Optimized Certainty-Equivalent (OCE) risks, and (ii) proposing a method for differentiating through CRC-based optimization objectives for model training.  The reviewers are unanimously in favor of acceptance.  Most of the reviewers' critical comments / questions were about details such as hyperparameters and limitations that also apply to CRC (use of calibration sets and restriction to monotonic losses).